# Connectivity Shapes Implicit Regularization in Matrix Factorization Models for Matrix Completion

**Zhiwei Bai[1], Jiajie Zhao[1], Yaoyu Zhang[1]***

[1] School of Mathematical Sciences, Institute of Natural Sciences, MOE-LSC,
Shanghai Jiao Tong University, Shanghai 200240, P.R. China.
{bai299, zjj0216, zhyy.sjtu}@sjtu.edu.cn.

## Abstract

Matrix factorization models have been extensively studied as a valuable test-bed for understanding the implicit biases of overparameterized models. Although both low nuclear norm and low rank regularization have been studied for these models, a unified understanding of when, how, and why they achieve different implicit regularization effects remains elusive. In this work, we systematically investigate the implicit regularization of matrix factorization for solving matrix completion problems. We empirically discover that the connectivity of observed data plays a crucial role in the implicit bias, with a transition from low nuclear norm to low rank as data shifts from disconnected to connected with increased observations. We identify a hierarchy of intrinsic invariant manifolds in the loss landscape that guide the training trajectory to evolve from low-rank to higher-rank solutions. Based on this finding, we theoretically characterize the training trajectory as following the hierarchical invariant manifold traversal process, generalizing the characterization of Li et al. (2020) to include the disconnected case. Furthermore, we establish conditions that guarantee minimum nuclear norm, closely aligning with our experimental findings, and we provide a dynamics characterization condition for ensuring minimum rank. Our work reveals the intricate interplay between data connectivity, training dynamics, and implicit regularization in matrix factorization models.

## 1 Introduction

Overparameterized models have the capacity to easily fit data with random labels (Zhang et al., 2017, 2021). However, in real-world applications, models with more parameters than training samples still generalize well. This has led researchers to hypothesize that overparameterized models undergo implicit regularization, favoring certain functions as outputs. Overparameterized matrix factorization models, $f_\theta = AB$ with $\theta = (A, B), A, B \in \mathbb{R}^{d \times d}$, have served as a simplified test-bed for studying this implicit regularization. In the context of matrix completion problems like the Netflix challenge, these models aim to find a low-rank completion of a partially observed matrix $M \in \mathbb{R}^{d \times d}$. Prior works have offered seemingly conflicting perspectives on the implicit regularization at play, with some claiming it promotes low nuclear norm (Gunasekar et al., 2017) and others arguing for low rank (Arora et al., 2019; Li et al., 2020; Razin and Cohen, 2020). However, a unified understanding of when, how, and why they achieve different implicit regularization effects remains elusive.

Unlike previous works that focus on either low rank or low nuclear norm regularization,, we systematically investigate the training dynamics and implicit regularization of matrix factorization for matrix completion. Through extensive experiments, we found that a certain connectivity property of observed data plays a key role in the implicit regularization effects. Data connectivity, in the context of this paper, refers to the way observed entries in the matrix are linked through shared rows

---

*Corresponding author: zhyy.sjtu@sjtu.edu.cn.

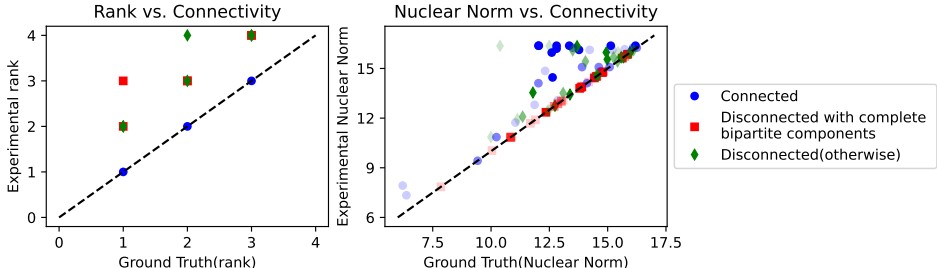

Figure 1: **The connectivity of observed data affects the implicit regularization.** The ground truth matrix $M^* \in \mathbb{R}^{4 \times 4}$ has rank ranging from 1 to 3. The sample size $n$ covers settings where $n$ is equal to, smaller than, and larger than the $2rd - r^2$ threshold required for exact reconstruction. Darker scatter points indicate a greater number of samples, while lighter points indicate fewer samples. The positions of observed entries are randomly chosen, and the experiment is repeated 10 times for each sample size. (Please refer to Appendix B for additional experiments and detailed methodology.)

or columns. A set of observations is considered connected if there's a path between any two observed entries via other observed entries in the same rows or columns. This concept plays a crucial role in determining the behavior of matrix factorization models, as we will demonstrate throughout this paper. As shown in Fig. 1, we sample observations randomly from a ground truth matrix $M^* \in \mathbb{R}^{d \times d}$ with rank$(M^*) < d$ and train models $f_\theta = AB$, $A, B \in \mathbb{R}^{d \times d}$ from small random initialization without any rank constraints. For each observation set, we calculate the solutions with the minimum nuclear norm and minimum rank, which serve as the ground truth benchmarks. These are then compared with the completion matrix obtained by the model. From Fig. 1, we observe that:

(i) **Low rank bias in connected case:** When the observed entries are connected, the model consistently learns the lowest-rank solution.

(ii) **Low nuclear norm bias in certain disconnected case:** When the observed entries are disconnected, the model generally does not find the minimum nuclear norm or lowest-rank solution. However, in the special case where each connected component is a complete bipartite subgraph, the model consistently finds the minimum nuclear norm solution.

To understand how data connectivity modulates the implicit bias, we analyze the loss landscape and optimization dynamics. We find a hierarchy of intrinsic invariant manifolds $\Omega_k$ of different ranks in the loss landscape. These manifolds constrain the optimization trajectory, causing the model to learn by incrementally ascending through higher ranks. In the disconnected case, additional sub-$\Omega_k$ invariant manifolds emerge within the $\Omega_k$ invariant manifold, preventing the model from reaching the global lowest-rank solution. However, we prove that the minimum nuclear norm solution is guaranteed in the disconnected with complete bipartite subgraph case.

The contributions of our work are summarized as follows:

(i) We systematically investigate the influence of data connectivity on the implicit regularization. Our empirical findings indicate that the connectivity of observed data plays a key role in the implicit bias, leading to a transition from favoring solutions with a low nuclear norm to those with a low rank as the data becomes more connected with an increase in observations (refer to Sec. 4).

(ii) We characterize the training dynamics of matrix factorization theoretically, showing that the optimization trajectory follows a *Hierarchical Invariant Manifold Traversal (HIMT)* process. This generalizes the characterization of Li et al. (2020), whose proposed *Greedy Low-Rank Learning(GLRL)* algorithm equivalence only corresponding to the connected case (refer to Sec. 5 and Sec. 6.1).

(iii) Regarding the minimum nuclear norm regularization, we establish conditions that provide guarantees closely aligned with our empirical findings, which complement the results of Gunasekar et al. (2017). For the minimum rank regularization, we present a dynamic characterization condition that assures the attainment of the minimum rank solution (refer to Sec. 6.2).

## 2 Related works

**Norm minimization and rank minimization.** Extensive research has been conducted on the implicit regularization of matrix factorization models, focusing on norm minimization and rank minimization. For norm minimization, Gunasekar et al. (2017) proved that gradient flow with infinitesimal initialization converges to the minimum nuclear norm solution in the special case of commutative observations. Ji and Telgarsky (2019); Gunasekar et al. (2018) studied norm minimization regularization in deep linear networks. For rank minimization, numerous works have shown that matrix factorization models favor low-rank solutions. Arora et al. (2019); Gidel et al. (2019); Gissin et al. (2019); Razin and Cohen (2020); Jiang et al. (2023); Belabbas (2020) investigated how infinitesimal initialization of gradient flow encourages low rank in specific settings. Li et al. (2020) showed that under certain assumptions, matrix factorization dynamics are equivalent to a greedy low-rank learning heuristic. Li et al. (2018); Stöger and Soltanolkotabi (2021); Jin et al. (2023) established low-rank recovery guarantees for matrix sensing problems under the Restricted Isometry Property (RIP) condition. Zhang et al. (2022, 2023) studied a broader class of model rank minimization for nonlinear models, of which the matrix factorization model is a special case.

**Nonlinear dynamics.** The initialization scale can significantly influence the implicit regularization of neural networks. Large initialization typically leads to linear dynamics (Jacot et al., 2018) and poor generalization (Chizat et al., 2019), while small initialization induces nonlinear dynamics (Luo et al., 2021). In this work, we focus on the case of infinitesimal initialization, which corresponds to highly nonlinear dynamics. An important characteristic of nonlinear neural network dynamics is the phenomenon of condensation (Luo et al., 2021; Zhou et al., 2022), where the network's effective complexity is small. The low-rank $\Omega_k$ invariant manifolds we propose are essentially a manifestation of condensation. Zhang et al. (2021, 2022); Bai et al. (2022); Fukumizu et al. (2019); Simsek et al. (2021) established the embedding principle of the loss landscape of neural networks and empirically demonstrated that the training process traverses critical points embedded from smaller subnetworks. Jacot et al. (2021) conjectured a saddle to saddle dynamics for deep linear networks, which is conceptually analogous to the dynamics characterization in this work.

## 3 Preliminaries

**Matrix completion problem.** This study focuses on the matrix completion problem, which involves estimating missing entries within a partially observed matrix. Given an incomplete matrix $M \in \mathbb{R}^{d \times d}$, the goal is to predict the entirety of $M$ based on its observed elements. The set of observed entries is represented as $S = \{(i_k, j_k), M_{i_k, j_k}\}_{k=1}^{n}$, where $(i_k, j_k)$ indicates the row and column indices, and $M_{i_k, j_k}$ is the corresponding value assumed non-zero in the matrix. The set of observed indices is defined as $S_{\boldsymbol{x}} = \{(i_k, j_k)\}_{k=1}^{n}$. Entries that are not observed, denoted by $\star$, are considered missing or unknown. The positions of observed elements in the matrix $M$ are defined by a binary observation matrix $P$, where $P_{ij} = 1$ indicates that $M_{ij}$ is observed, and $P_{ij} = 0$ indicates that $M_{ij}$ is unobserved.

**Matrix factorization model.** Matrix factorization is a prevalent approach for addressing the matrix completion problem. It reconstructs the matrix $W \in \mathbb{R}^{d \times d}$ through the product $W = AB$, where $A \in \mathbb{R}^{d \times r}$ and $B \in \mathbb{R}^{r \times d}$. This work studies the overparameterized scenario with $r = d$, aiming to understand the implicit regularization effect in the absence of explicit rank restrictions, paralleling prior research (Gunasekar et al., 2017; Arora et al., 2019; Li et al., 2020; Jin et al., 2023). In this work, we focus on the asymmetric factorization, which can be represented as a parametric model:

$$\boldsymbol{f}_{\boldsymbol{\theta}} = AB, \quad A, B \in \mathbb{R}^{d \times d}. \tag{1}$$

The matrix factorization model parameters are denoted by $\boldsymbol{\theta} = (A, B)$, identified with its vectorized form $\text{vec}(\boldsymbol{\theta}) \in \mathbb{R}^{2d^2}$. The augmented matrix is $W_{\text{aug}}^{\top} = \begin{bmatrix} A^{\top} & B \end{bmatrix}^{\top} \in \mathbb{R}^{d \times 2d}$, and $\text{row}(A)$ and $\text{col}(B)$ denote the row and column spaces of $A$ and $B$, respectively. The augmented matrix $W_{\text{aug}}$ plays a crucial role in our subsequent analysis, particularly in characterizing the intrinsic invariant manifolds $\Omega_k$ of the optimization process. Specifically, it allows us to establish the relationship $\text{rank}(A) = \text{rank}(B^{\top}) = \text{rank}(W_{\text{aug}})$, which is important to understanding the invariance property under gradient flow.

**Loss function.** The learning process for the parameters $\boldsymbol{\theta} = (\boldsymbol{A}, \boldsymbol{B})$ involves minimizing a loss function that measures the difference between observed and estimated entries. In this work, we focus on the mean squared error, and the empirical risk is thus formulated as

$$R_S(\boldsymbol{\theta}) = \frac{1}{n}\|(\boldsymbol{AB} - \boldsymbol{M})_{S_{\boldsymbol{x}}}\|_F^2 := \frac{1}{n}\sum_{k=1}^{n}(\boldsymbol{a}_{i_k} \cdot \boldsymbol{b}_{\cdot,j_k} - \boldsymbol{M}_{i_k,j_k})^2, \tag{2}$$

where $\boldsymbol{a}_i$ and $\boldsymbol{b}_{\cdot,j}$ represent the $i$-th row and $j$-th column of matrix $\boldsymbol{A}$ and $\boldsymbol{B}$, respectively. The residual matrix $\delta\boldsymbol{M} = (\boldsymbol{AB} - \boldsymbol{M})_{S_{\boldsymbol{x}}}$ has elements $\delta\boldsymbol{M}_{ij} = (\boldsymbol{AB})_{ij} - \boldsymbol{M}_{ij}$ for $(i,j) \in S_{\boldsymbol{x}}$ and $\delta\boldsymbol{M}_{ij} = 0$ for $(i,j) \notin S_{\boldsymbol{x}}$. The training dynamics follow the gradient flow of $R_S(\boldsymbol{\theta})$:

$$\frac{\mathrm{d}\boldsymbol{\theta}}{\mathrm{d}t} = -\nabla_{\boldsymbol{\theta}}R_S(\boldsymbol{\theta}), \quad \boldsymbol{\theta}(0) = \boldsymbol{\theta}_0. \tag{3}$$

In all experiments, $\boldsymbol{\theta}_0 \sim N(0, \sigma^2)$ is initialized from a Gaussian distribution with mean 0 and small variance $\sigma^2$. We use gradient descent with a small learning rate to approximate the gradient flow dynamics (Please refer to Appendix B.1 for the detailed experiment setup).

## 4 Connectivity affects implicit regularization

In this section, we define connectivity and present experimental results on implicit regularization for connected and disconnected observational data.

**Definition 1** (**Associated Observation Graph**). *Given a incomplete matrix $\boldsymbol{M}$ to be completed and its observation matrix $\boldsymbol{P}$, the associated observation graph $G_{\boldsymbol{M}}$ is the bipartite graph with adjacency matrix $\begin{bmatrix} \boldsymbol{0} & \boldsymbol{P}^\top \\ \boldsymbol{P} & \boldsymbol{0} \end{bmatrix}$, with isolated vertices removed.*

**Definition 2** (**Connectivity**). *Given a incomplete matrix $\boldsymbol{M}$ to be completed, it is considered connected if its associated observation graph $G_{\boldsymbol{M}}$ is connected; otherwise, it is disconnected. The connected components of $\boldsymbol{M}$ are defined as the connected components of $G_{\boldsymbol{M}}$.*

The connectivity of the graph, as defined above, reflects the connectivity of the observed data. Appendix A Sec. A.2 provides a detailed discussion on the equivalent definition of connectivity.

In the case of disconnectivity, there is a special case where each connected component has full observations, characterized by disconnectivity with complete bipartite components.

**Definition 3** (**Disconnectivity with Complete Bipartite Components**). *A incomplete matrix $\boldsymbol{M}$ is considered disconnected with complete bipartite components if its associated observation graph $G_{\boldsymbol{M}}$ is disconnected and each connected component forms a complete bipartite subgraph.*

We present examples to demonstrate how connectivity influences the characteristics of the learned solutions. Consider three matrices to be completed, each obtained by adding one more observation to the previous matrix: $\boldsymbol{M}_1$ (disconnected), $\boldsymbol{M}_2$ (disconnected with complete bipartite components), and $\boldsymbol{M}_3$ (connected). Fig. A1 of Appendix B illustrates the associated graphs $G_{\boldsymbol{M}}$.

$$\boldsymbol{M}_1 = \begin{bmatrix} 1 & 2 & \star \\ 3 & \star & \star \\ \star & \star & 5 \end{bmatrix}, \boldsymbol{M}_2 = \begin{bmatrix} 1 & 2 & \star \\ 3 & 4 & \star \\ \star & \star & 5 \end{bmatrix}, \boldsymbol{M}_3 = \begin{bmatrix} 1 & 2 & \star \\ 3 & 4 & \star \\ 6 & \star & 5 \end{bmatrix}. \tag{4}$$

Figs. 2(a-b) compare the learned matrices with the ground truth (GT) solutions having the smallest nuclear norm and rank. For disconnected $\boldsymbol{M}_1$ (blue bars), the learned solution achieves neither the smallest nuclear norm nor rank. For disconnected $\boldsymbol{M}_2$ with complete bipartite components (green bars), the learned matrix has the smallest nuclear norm but not rank. For connected $\boldsymbol{M}_3$ (red bars), the lowest rank-2 solution is not unique; the model identifies a particular lowest rank-2 solution, but it does not correspond to the one with the minimum nuclear norm.

To thoroughly study all possible cases, we examine all sampling patterns of the $3 \times 3$ matrix completion. Fig. 2(c) shows that the model consistently learns the lowest-rank solution for connected sampling patterns but fails to do so for disconnected patterns. Fig. 2(d) further verifies the impact of connectivity on low-rank matrix recovery by comparing the reconstruction error for 100 randomly sampled rank-1 matrices using two connected sampling patterns (red and blue dots) and one disconnected sampling

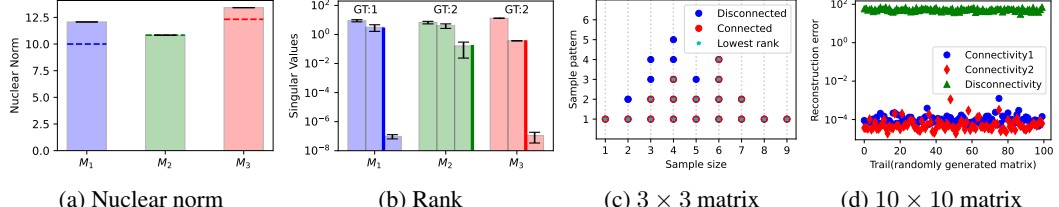

| (a) Nuclear norm | (b) Rank | (c) $3 \times 3$ matrix | (d) $10 \times 10$ matrix |

Figure 2: (a) Nuclear norms of the learned solutions for $M_1$, $M_2$, and $M_3$. Dashed lines represent theoretically computed smallest nuclear norms. (b) Singular values of the learned matrices for $M_1, M_2, M_3$. Each set of three bars represents the singular values of a matrix. The thick vertical lines partition significantly nonzero singular values, which serves as the empirical rank. The text (GT) shows the ground truth minimum rank. Mean and standard deviation are recorded over 100 repetitions. (c) All equivalent sampling patterns of the $3 \times 3$ matrix completion problem (see Appendix B for details). Cyan stars marked the case learning the lowest-rank solution. (d) Reconstruction error of the solutions for a $10 \times 10$ matrix reconstruction problem with $M^*$ randomly sampled at rank $r = 1$ and sample size set to the minimum reconstruction setting $n = 2rd - r^2$.

pattern (green dots). The model consistently achieves small reconstruction errors under connected sampling patterns, while the error is significantly larger for the disconnected pattern.

These empirical results demonstrate an implicit preference for low rank induced by connectivity and a preference for low nuclear norm in a particular kind of disconnection. In the following section, we will investigate the training dynamics under both connected and disconnected scenarios.

## 5 Training dynamics in connected and disconnected cases

### 5.1 Connected case

This section empirically demonstrates the detailed dynamics of connected observed data. Fig. 3(a) shows the connected target matrix $M$ with a single unknown element denoted by $\star$. The rank of $M$ is at least three and equals three if and only if $\star = 1.2$.

**Learning lowest-rank solution.** We initialize $A$ and $B$ with different scales and record the singular values of the learned matrix. As depicted in Fig. 3(b), when starting with larger initialization, the learned solutions are almost always rank-4. Conversely, as the initialization scale decreases, the first three singular values of the learned solution are consistently maintained in magnitude, but the fourth singular value keeps decreasing, resulting in the model learning the lowest rank-3 solution.

**Traversing progressive optima at each rank.** For a small random initialization (Gaussian distribution with mean 0 and variance $10^{-16}$), the loss curves exhibit a steady, stepwise decline (Fig. 3(c)). The flat periods correspond to small gradient norms, indicating potential saddle points (Fig. 3(d)). We compare the matrices learned at these saddle points with the optimal approximation of each rank and plot their difference in Fig. 3(d), which is very small. These findings suggest that the model starts near $\mathbf{0}$ (rank-0) and progressively finds optimal approximations within rank-1, rank-2, and higher-rank manifolds until reaching a global minimum.

**Alignment of the row space of $A$ and the column space of $B$.** Starting with small initialization, we track the rank (number of significantly non-zero singular values) of $W = AB$, $A$, $B$, and the augmented matrix $W_{\mathrm{aug}}$ during the training process. We observe that the rank gradually increases, with singular values growing rapidly one after another (Fig. 3(e-h)). Throughout the entire process, we consistently find that $\mathrm{rank}(A) = \mathrm{rank}(B^\top) = \mathrm{rank}(W_{\mathrm{aug}})$, which implies that the row space of $A$ and the column space of $B$ remain aligned at all times. This alignment corresponds to a special structure that we refer to as the "Hierarchical Intrinsic Invariant Manifold" in Sec. 6.1, which plays a crucial role in the overall dynamics of the system.

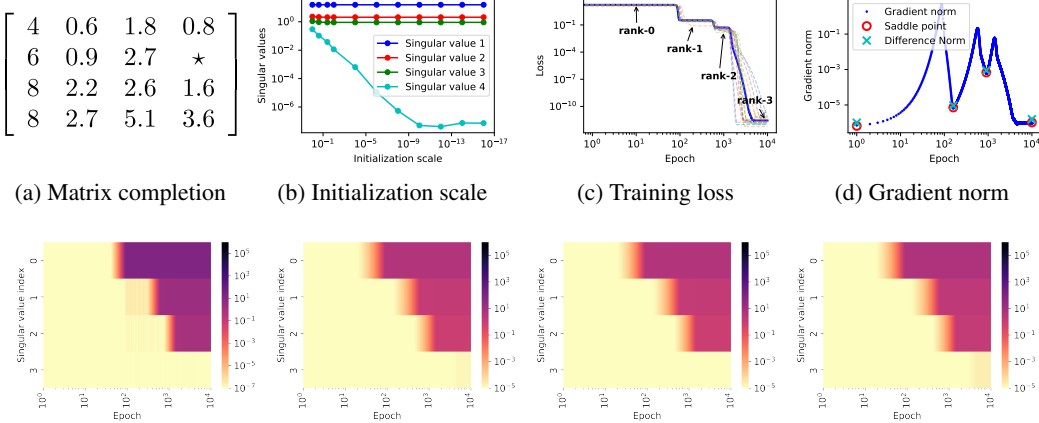

(a) Matrix completion    (b) Initialization scale    (c) Training loss    (d) Gradient norm

(e) Singular values of $W$    (f) Singular values of $A$    (g) Singular values of $B$    (h) Singular values of $W_{\text{aug}}$

Figure 3: (a) The matrix $M$ to be completed, with the $\star$ position unknown. (b) The four singular values of the learned solution at different initialization scale (Gaussian distribution, mean 0, variance from $10^0$ to $10^{-16}$). (c) Training loss for 16 connected sampling patterns in a $4 \times 4$ matrix, each covering 1 element and observing the remaining 15 in a fixed rank-3 matrix. (d) Evolution of the $l_2$-norm of the gradients throughout the training process. The cyan crosses represent the difference between the matrix corresponding to the saddle point and the optimal approximation at each rank. (e-h) Evolution of singular values for matrices $W, A, B$, and $W_{\text{aug}}$ during training.

The dynamics of increasing ranks step by step aligns with the description of *Greedy Low Rank Learning (GLRL)* (Li et al., 2020). However, we will show next that when the observed data are disconnected, the learning process is not equivalent to GLRL.

## 5.2 Disconnected case

In this section, we present a typical experiment in the disconnected situation. As depicted in Fig. 4(a), the target matrix $M$ contains four unknown elements denoted by $\star$ and is disconnected. The rank of $M$ is at least one, and there are infinitely many rank-1 solutions.

**Alignment of the row space of $A$ and the column space of $B$.** As shown in Fig. 4(b-e), the learning process in the disconnected case is similar to the previous experiment: the model naturally evolves from low-rank to high-rank, with each step increasing a singular value and satisfying $\text{rank}(A) = \text{rank}(B^\top) = \text{rank}(W_{\text{aug}})$. Fig. 4(f) illustrates that as the initialization scale decreases, the model tends to learn symmetric solutions. However, unlike the connected case, the output does not approach a particular solution as the initialization decreases. For this specific disconnected $M$, we will show that every symmetric solution learned is a minimal nuclear norm solution(see Sec. 6.2 Thm. 4). For fewer observations, the experimental phenomena are similar (see Appendix B Fig. B5).

**Lowest-rank solution is not learned.** Despite the adaptive learning behavior, the final learned solution has rank 2, as evidenced by the two significantly non-zero singular values in Fig. 4(b-d). Examining the dynamics (3), we find that they decouple into two independent systems: one for the 1st and 3rd rows of $A$ and columns of $B$, and another for the 2nd row of $A$ and column of $B$. Fig. 4(g) shows that the model first learns the surrounding elements $1, 3, 3, 9$ (rank-1 saddle point), then learns the middle element $5$ in the next stage. The decoupling of dynamics is equivalent to the definition of disconnection (see Appendix A Prop. A.4 for proof). In Fig. 4(e), we fixed a rank-1 matrix and explored all nine disconnected sampling patterns with 5 observations. For each pattern, we conducted experiments with small initializations. The loss curves consistently indicate that in disconnected cases, the model learns a sub-optimal solution in the rank-1 manifold, ultimately resulting in a rank-2 solution. This demonstrates that regardless of the specific disconnected sampling pattern, the model fails to achieve the optimal low-rank solution.

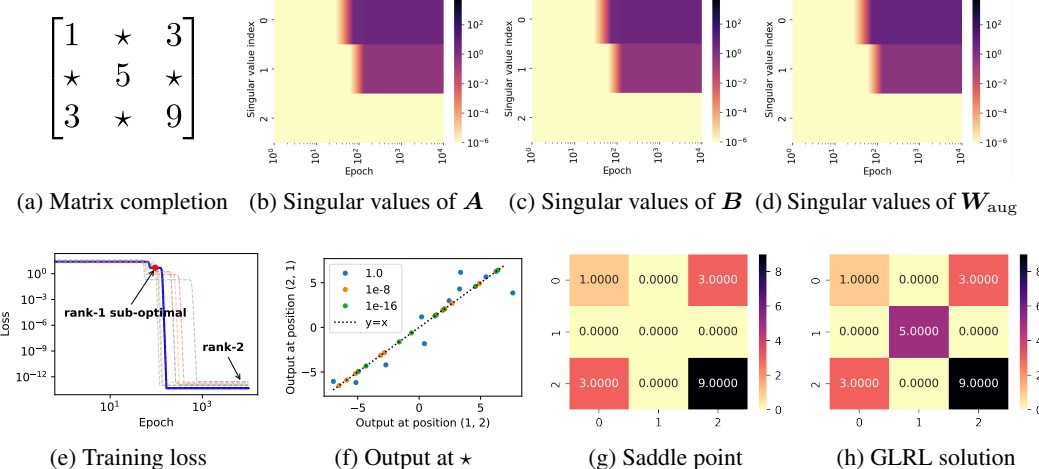

Figure 4: (a) The matrix to be completed, with unknown entries marked by $\star$. (b-d) Evolution of singular values for $\boldsymbol{A}$, $\boldsymbol{B}$, and $\boldsymbol{W}_{\mathrm{aug}}$ during training. (e) Training loss for 9 disconnected sampling patterns in a $3 \times 3$ matrix, each covering 4 elements and observing the remaining 5 in a fixed rank-1 matrix. (f) Learned values at symmetric positions $(1, 2)$ and $(2, 1)$ under varying initialization scales (zero mean, varying variance). Each point represents one of ten random experiments per variance; labels show initialization variance. Other symmetric positions exhibit similar behavior. (g) Learned output at the saddle point corresponding to the red dot in (e). (h) Final learned solution of the GLRL algorithm (Li et al., 2020).

**Not equivalent to GLRL in disconnected case.** We compare the GLRL algorithm (Li et al., 2020) with the matrix factorization model for solving the same matrix completion problem (Fig. 4). Li et al. (2020) claim that the matrix factorization dynamics is mathematically equivalent to the GLRL algorithm under reasonable assumptions. While GLRL learns the same rank-1 saddle point shown in Fig. 4(g) in the first stage, it then fills unobserved elements with 0, resulting in a unique rank-2 solution (Fig. 4(h)). In contrast, the matrix factorization model learns symmetric solutions with some degree of freedom depending on the random seed (Fig. 4(f)). The key difference is that the first critical point (Fig. 4(g)) reached by the trajectory is a sub-optimal and not a second-order stationary point of the rank-1 manifold as assumed by Li et al. (2020). Therefore, the equivalence assumption between GLRL and matrix factorization does not hold in the disconnected case.

## 6 Theoretical analysis of training dynamics and implicit regularization

### 6.1 Characterization of training dynamics

Matrix factorization models exhibit a distinctive adaptive learning behavior, progressively evolving from low rank to high rank. Understanding this phenomenon is rooted in grasping the global dynamics of matrix factorization models, where the role of intrinsic invariant manifolds becomes critical.

**Proposition 1 (Hierarchical Intrinsic Invariant Manifold (HIIM)).** *(see Appendix A Prop. A.1 for Proof) Let $\boldsymbol{f_\theta} = \boldsymbol{AB}$ be a matrix factorization model and $\{\boldsymbol{\alpha}_1, \cdots, \boldsymbol{\alpha}_k\}$ be $k$ linearly independent vectors. Define the manifold $\boldsymbol{\Omega}_k$ as $\boldsymbol{\Omega}_k := \boldsymbol{\Omega}_k(\boldsymbol{\alpha}_1, \cdots, \boldsymbol{\alpha}_k) = \{\boldsymbol{\theta} = (\boldsymbol{A}, \boldsymbol{B}) \mid \mathrm{row}(\boldsymbol{A}) = \mathrm{col}(\boldsymbol{B}) = \mathrm{span}\{\boldsymbol{\alpha}_1, \cdots, \boldsymbol{\alpha}_k\}\}$. The manifold $\boldsymbol{\Omega}_k$ possesses the following properties:*

*(i) **Invariance under Gradient Flow:** Given data $S$ and the gradient flow dynamics $\dot{\boldsymbol{\theta}} = -\nabla R_S(\boldsymbol{\theta})$, if the initial point $\boldsymbol{\theta}_0 \in \boldsymbol{\Omega}_k$, then $\boldsymbol{\theta}(t) \in \boldsymbol{\Omega}_k$ for all $t \geq 0$.*

*(ii) **Intrinsic Property:** $\boldsymbol{\Omega}_k$ is a data-independent invariant manifold, meaning that for any data $S$, $\boldsymbol{\Omega}_k$ remains invariant under the gradient flow dynamics.*

*(iii) **Hierarchical Structure:** The manifolds $\boldsymbol{\Omega}_k$ form a hierarchy: $\boldsymbol{\Omega}_0 \subsetneq \boldsymbol{\Omega}_1 \subsetneq \cdots \subsetneq \boldsymbol{\Omega}_{k-1} \subsetneq \boldsymbol{\Omega}_k$.*

Figs. 3(f-h) and Figs. 4(b-d) show that the training process with small initialization consistently satisfies $\text{rank}(\boldsymbol{A}) = \text{rank}(\boldsymbol{B}^{\top}) = \text{rank}(\boldsymbol{W}_{\text{aug}})$, aligning with the $\boldsymbol{\Omega}_k$ invariant manifold. Since a non-zero initialization in practice, the training trajectory is close to the $\boldsymbol{\Omega}_k$ invariant manifold, approaches a critical point, and transitions to the next level invariant manifold without getting trapped.

In both connected and disconnected scenarios, we observe a step-by-step hierarchical $\boldsymbol{\Omega}_k$ invariant manifold traversal. In the connected case, at each level we observe that the model reaches an optimal solution (Fig. 3). However, in the disconnected case, we can prove that each connected component induces a sub-$\boldsymbol{\Omega}_k$ invariant manifold, leading to the experimentally observed sub-optimal solution (see Fig. 4).

**Proposition 2** (**Intrinsic Sub-$\boldsymbol{\Omega}_k$ Invariant Manifold**). *(see Appendix A Prop. A.2 for Proof) Let $f_{\boldsymbol{\theta}} = \boldsymbol{AB}$ be a matrix factorization model, $\boldsymbol{M}$ be an incomplete matrix and $\boldsymbol{\Omega}_k$ be an invariant manifold defined in Prop. 1. If $\boldsymbol{M}$ is disconnected with $m$ connected components, then there exist $m$ sub-$\boldsymbol{\Omega}_k$ manifolds $\boldsymbol{\omega}_k$ such that $\boldsymbol{\omega}_k \subsetneqq \boldsymbol{\Omega}_k$, each possessing the following properties:*

*(i) **Invariance under Gradient Flow:** Given data $S$ and the gradient flow dynamics $\dot{\boldsymbol{\theta}} = -\nabla R_S(\boldsymbol{\theta})$, if the initial point $\boldsymbol{\theta}_0 \in \boldsymbol{\omega}_k$, then $\boldsymbol{\theta}(t) \in \boldsymbol{\omega}_k$ for all $t \geq 0$.*

*(ii) **Intrinsic Property:** $\boldsymbol{\omega}_k$ is a data-value-independent invariant manifold, meaning that for a fixed sampling pattern in $\boldsymbol{M}$ and any observed values $S$, $\boldsymbol{\omega}_k$ remains invariant under the gradient flow.*

*(iii) **Strict Subset Relation:** The output set $\{f_{\boldsymbol{\theta}} \mid \boldsymbol{\theta} \in \boldsymbol{\omega}_k\}$ is a proper subset of $\{f_{\boldsymbol{\theta}} \mid \boldsymbol{\theta} \in \boldsymbol{\Omega}_k\}$, namely, $\{f_{\boldsymbol{\theta}} \mid \boldsymbol{\theta} \in \boldsymbol{\omega}_k\} \subsetneqq \{f_{\boldsymbol{\theta}} \mid \boldsymbol{\theta} \in \boldsymbol{\Omega}_k\}$.*

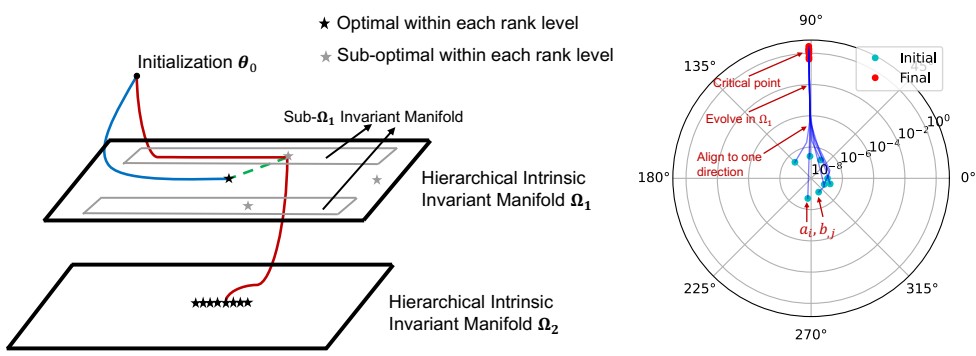

(a) Illustration of training trajectories in disconnected case       (b) Alignment of row($\boldsymbol{A}$) and col($\boldsymbol{B}$)

Figure 5: (a) Illustrated trajectories for the experiment in Fig. 4. The blue line represents the trajectory converging to the lowest-rank solution, and the red line represents the actual trajectory experienced by the model. (b) The parameter trajectory escaping from a second-order stationary point to reach the next critical point for the experiment in Fig. 3. The 8 scatter points represent the 4 row vectors of matrix $\boldsymbol{A}$ and the 4 column vectors of matrix $\boldsymbol{B}$. For ease of visualization, we randomly project them onto two dimensions and plot them in polar coordinates.

Fig. 5(a) illustrates the trajectory of the experiment in Fig. 4. In the disconnected case, sub-$\boldsymbol{\Omega}_k$ invariant manifolds exist and attract the dynamics, leading the model to learn sub-optimal solutions on the entire $\boldsymbol{\Omega}_k$ invariant manifold. In fact, we can prove that these sub-optimal solutions are necessarily strict saddle points. This loss landscape result extends Theorem 5.10 from Li et al. (2020), which established the findings for the specific case of symmetric matrix factorization models (see Appendix A Sec. A.3 for a detailed discussion).

**Theorem 1** (**Loss Landscape**). *(see Appendix A Thm. A.3 for Proof) Given any data $S$, the critical points of $R_S(\boldsymbol{\theta})$ are either strict saddle points or global minima.*

Gradient descent easily escapes saddle points (Lee et al., 2016, 2019). Fig. 5(b) shows that when the model escapes a saddle point, the parameters initially appear chaotic but align in one direction after some time, consistent with the "condensation" phenomenon in neural networks (Luo et al., 2021; Zhou et al., 2022). For matrix factorization models, by meticulously analyzing the Hessian matrix structure (see Appendix A.5), we find that this alignment corresponds to an $\boldsymbol{\Omega}_1$ invariant manifold,

resulting in a rank increase of one at a time. Under reasonable assumptions, we prove that the training trajectory follows the $\mathbf{\Omega}_k$ invariant manifold step by step.

**Assumption 1** (**Unique Top Singular Value**). *Let $\delta \boldsymbol{M} = (\boldsymbol{A}_c \boldsymbol{B}_c - \boldsymbol{M})_{S_x}$ be the residual matrix at the critical point $\boldsymbol{\theta}_c = (\boldsymbol{A}_c, \boldsymbol{B}_c)$. Assume that the largest singular value of $\delta \boldsymbol{M}$ is unique.*

**Assumption 2** (**Second-order Stationary Point**). *Let $\mathbf{\Omega}$ be an $\mathbf{\Omega}_k$ invariant manifold or sub-$\mathbf{\Omega}_k$ invariant manifold defined in Prop. 1 or 2. Assume $\boldsymbol{\theta}_c$ is a second-order stationary point within $\mathbf{\Omega}$, i.e., $\nabla R_S(\boldsymbol{\theta}_c) = 0$ and $\boldsymbol{\theta}^\top \nabla^2 R_S(\boldsymbol{\theta}_c)\boldsymbol{\theta} \geq 0$ for all $\boldsymbol{\theta} \in \mathbf{\Omega}$.*

**Theorem 2** (**Transition to the Next Rank-level Invariant Manifold**). *(see Appendix A Thm. A.4 for proof) Consider the dynamics $\dot{\boldsymbol{\theta}} = -\nabla R_S(\boldsymbol{\theta})$. Let $\varphi(\boldsymbol{\theta}_0, t)$ denote the value of $\boldsymbol{\theta}(t)$ when $\boldsymbol{\theta}(0) = \boldsymbol{\theta}_0$. Let $\mathbf{\Omega}$ be an $\mathbf{\Omega}_k$ or sub-$\mathbf{\Omega}_k$ invariant manifold. Let $\boldsymbol{\theta}_c \in \mathbf{\Omega}$ be a critical point satisfying Assump. 1 and 2. Then, for randomly selected $\boldsymbol{\theta}_0$, with probability 1 with respect to $\boldsymbol{\theta}_0$, the limit*

$$\tilde{\varphi}(\boldsymbol{\theta}_c, t) := \lim_{\alpha \to 0} \varphi\left(\boldsymbol{\theta}_c + \alpha\boldsymbol{\theta}_0, t + \frac{1}{\lambda_1}\log\frac{1}{\alpha}\right) \tag{5}$$

*exists and falls into an invariant manifold $\mathbf{\Omega}_{k+1}$. Here $\lambda_1$ is the top eigenvalue of $-\nabla^2 R_S(\boldsymbol{\theta}_c)$.*

*Proof sketch.* The main idea is to analyze the local dynamics near the critical point $\boldsymbol{\theta}_c$. The nonlinear dynamics can be approximated linearly in the vicinity of $\boldsymbol{\theta}_c$: $\frac{\mathrm{d}\boldsymbol{\theta}}{\mathrm{d}t} \approx \boldsymbol{H}(\boldsymbol{\theta}_0 - \boldsymbol{\theta}_c)$, where $\boldsymbol{H} = -\nabla^2 R_S(\boldsymbol{\theta}_c)$ is the negative Hessian matrix. For exact linear approximation, the solution is: $\boldsymbol{\theta}(t) = e^{t\boldsymbol{H}}(\boldsymbol{\theta}_0 - \boldsymbol{\theta}_c) + \boldsymbol{\theta}_c$. Let $\lambda_1 > \lambda_2 > ... > \lambda_s$ be the eigenvalues of $\boldsymbol{H}$, with corresponding eigenvectors $\boldsymbol{q}_{ij}$. We can express $\boldsymbol{\theta}(t)$ as: $\boldsymbol{\theta}(t) = \sum_{i=1}^{s}\sum_{j=1}^{l_i} e^{\lambda_i t}\langle\boldsymbol{\theta}_0 - \boldsymbol{\theta}_c, \boldsymbol{q}_{ij}\rangle\boldsymbol{q}_{ij} + \boldsymbol{\theta}_c$. For sufficiently large $t_0$, the dynamics follows a dominant eigenvalue dynamics: $\boldsymbol{\theta}(t_0) = \sum_{j=1}^{l_1} e^{\lambda_1 t_0}\langle\boldsymbol{\theta}_0 - \boldsymbol{\theta}_c, \boldsymbol{q}_{1j}\rangle\boldsymbol{q}_{1j} + O(e^{\lambda_2 t_0})$. Through detailed analysis of the eigenvalues and eigenvectors of the Hessian matrix (please refer to Lems A.2-A.4 of Appendix A), we show that if the largest singular value of residual matrix $\delta \boldsymbol{M}$ at $\boldsymbol{\theta}_c$ is unique and $\boldsymbol{\theta}_c$ is a second-order stationary point within $\mathbf{\Omega}$, the first principal component $\sum_{j=1}^{l_1} e^{\lambda_1 t_0}\langle\boldsymbol{\theta}_0 - \boldsymbol{\theta}_c, \boldsymbol{q}_{1j}\rangle\boldsymbol{q}_{1j}$ will happen to be an $\mathbf{\Omega}_1$ invariant manifold. Consequently, escaping $\boldsymbol{\theta}_c$ increases the rank by 1, entering $\mathbf{\Omega}_{k+1}$. $\square$

**Remark.** *Assump. 1 ensures that upon departing from a critical point $\boldsymbol{\theta}_c$, the trajectory is constrained to escape along a single dominant eigendirection corresponding to the largest singular value. This assumption holds for randomly generated matrix with probability 1, making it a reasonable condition in most practical scenarios. In Sec A.7 of Appendix A, we provide an special example to illustrate the situation where Assump. 1 does not hold.*

**Remark.** *To ensure the escape direction falls within the $\mathbf{\Omega}_{k+1}$ invariant manifold, the Hessian's top eigenvectors must satisfy $rank(\boldsymbol{A}) = rank(\boldsymbol{B}^\top) = rank(\boldsymbol{W}_{aug})$. The condition that $\boldsymbol{\theta}_c$ is a second-order stationary point within $\mathbf{\Omega}$ in Assump. 2 guarantees this Hessian structure. Our Assump. 2 is more general than conditions proposed by Li et al. (2020), as it remains valid across both connected and disconnected configurations. Empirical findings (Figs. 3 and 4) indicate that this assumption consistently holds in practical scenarios.*

Thm. 2 provides a characterization of the escape trajectory. It shows that as the point approaches a second-order stationary point $\boldsymbol{\theta}_c \in \mathbf{\Omega}_k$, the trajectory generically converges to a well-defined limit within $\mathbf{\Omega}_{k+1}$. Since the origin $\boldsymbol{0}$ is always a second-order stationary point of $\mathbf{\Omega}_0$, the theorem implies that the trajectory escaping from a small initialization will be close to $\mathbf{\Omega}_1$. This iterative process gives rise to the phenomenon of *Hierarchical Invariant Manifold Traversal (HIMT)*, which involves a sequential progression through these $\mathbf{\Omega}_k$ manifolds.

## 6.2 Implicit regularization analysis

Rank minimization is a challenging non-convex optimization problem. Li et al. (2018); Jin et al. (2023) proved that the Restricted Isometry Property (RIP) condition ensures a minimal rank solution. However, the RIP condition is often too stringent for practical matrix completion. For instance, the matrix $\boldsymbol{M}_3$ in Eq. (4) does not satisfy the RIP criteria, yet the model still finds the minimum rank solution. Our empirical findings (Figs. 1, 2, 3) suggest that a more lenient condition, specifically the connectivity of the observed data, frequently leads to convergence towards the minimal rank solution. Proving this result directly, however, would necessitate a comprehensive examination of

the convergence characteristics within each $\boldsymbol{\Omega}_k$ invariant manifold, which is an endeavor we leave for future work. Despite this, our insights into the system's dynamics, i.e., hierarchical invariant manifold traversal, allow us to assert that if a trajectory successfully navigates through the optimal on each rank-level invariant manifold $\boldsymbol{\Omega}_k$, a solution of minimal rank can be achieved naturally.

**Theorem 3** (**Minimum Rank**). *(see Appendix A Thm. A.5 for proof) Consider the dynamics $\dot{\boldsymbol{\theta}} = -\nabla R_S(\boldsymbol{\theta})$, where $\boldsymbol{\theta}(t) = (\boldsymbol{A}(t), \boldsymbol{B}(t))$, and denote $\boldsymbol{W}_t = \boldsymbol{A}(t)\boldsymbol{B}(t)$. Assume $\boldsymbol{W}_t$ achieves an optimal within each invariant manifold $\boldsymbol{\Omega}_k$. For a full rank initialization $\boldsymbol{W}_0$, if the limit $\widehat{\boldsymbol{W}} = \lim_{\alpha \to 0} \boldsymbol{W}_\infty(\alpha \boldsymbol{W}_0)$ exists and is a global optimum with $\widehat{\boldsymbol{W}}_{ij} = \boldsymbol{M}_{ij}$ for all $(i,j) \in S_{\boldsymbol{x}}$, then*

$$\widehat{\boldsymbol{W}} \in \operatorname{argmin}_{\boldsymbol{W}} \operatorname{rank}(\boldsymbol{W}) \quad s.t. \quad \boldsymbol{W}_{ij} = \boldsymbol{M}_{ij}, \forall (i,j) \in S_{\boldsymbol{x}}. \tag{6}$$

For a disconnected matrix $\boldsymbol{M}$, our theoretical results (Prop. 2) and experiments (Fig. 4) confirm the existence of sub-$\boldsymbol{\Omega}_k$ invariant manifolds. These manifolds attract the training trajectory, leading to sub-optimal solutions and preventing convergence to the lowest-rank solution.

However, in a specific disconnected case, such as disconnection with complete bipartite components, as illustrated in Figs. 1 and 2, the minimum nuclear norm may still serve as a characterization. Gunasekar et al. (2017) proved a special case: if the observations are commutative, then the symmetric model will learn the minimum nuclear norm solution. Intriguingly, for the example $\boldsymbol{M}_2$ in Eq. (4), even though the observations are not commutative, the model still learns a minimum nuclear norm solution. In fact, we can prove the following result, which aligns well with practical experiments.

**Theorem 4** (**Minimum Nuclear Norm Guarantee**). *(see Appendix A Thm. A.6 for proof) Consider the dynamics $\dot{\boldsymbol{\theta}} = -\nabla R_S(\boldsymbol{\theta})$, where $\boldsymbol{\theta}(t) = (\boldsymbol{A}(t), \boldsymbol{B}(t))$, and let $\boldsymbol{W}_t = \boldsymbol{A}(t)\boldsymbol{B}(t)$. If the observation graph associated with the incomplete matrix $\boldsymbol{M}$ is disconnected with complete bipartite components, and if for a full rank initialization $\boldsymbol{W}_0$, the limit $\widehat{\boldsymbol{W}} = \lim_{\alpha \to 0} \boldsymbol{W}_\infty(\alpha \boldsymbol{W}_0)$ exists and is a global optimum with $\widehat{\boldsymbol{W}}_{ij} = \boldsymbol{M}_{ij}$ for all $(i,j) \in S_{\boldsymbol{x}}$, then*

$$\widehat{\boldsymbol{W}} \in \operatorname{argmin}_{\boldsymbol{W}} \|\boldsymbol{W}\|_* \quad s.t. \quad \boldsymbol{W}_{ij} = \boldsymbol{M}_{ij}, \forall (i,j) \in S_{\boldsymbol{x}}. \tag{7}$$

# 7 Conclusion and future work

This study presents a comprehensive experimental and theoretical investigation of matrix factorization models. The primary objective was to develop a cohesive framework for understanding the conditions, mechanisms, and reasons behind the diverse implicit regularization effects exhibited by matrix factorization models. A key finding of this research is the pivotal role of the connectivity of observed data in shaping the implicit regularization behavior. To elucidate this phenomenon, we identified the significance of hierarchical invariant manifold traversal within the training dynamics.

Our experiments (Figs. 1, 2, 3) provide strong evidence that connected observed data leads to minimum-rank solutions, as the model learns the optimal of the $\boldsymbol{\Omega}_k$ invariant manifold. However, further investigation is needed to uncover the underlying mechanisms by which connectivity facilitates optimal attainment across different $\boldsymbol{\Omega}_k$ invariant manifolds. Additionally, the trade-offs between initialization scale and training efficiency warrant further research, as certain cases may require extremely small initialization, potentially impacting training speed (see Appendix B Sec. B.4).

Generalizing the insights gained from matrix factorization models to other architectures is also an important avenue for future work. Our preliminary experiments indicate that the learning phenomenon from low rank to high rank persists in deep multi-layer matrix factorization and the query-key factorization model in Transformer attention mechanisms (see Appendix B Figs. B9, B11). These findings suggest that the hierarchical invariant manifold traversal process uncovered in our study may have broader implications and merit further exploration.

## Acknowledgments and Disclosure of Funding

This work is sponsored by the National Natural Science Foundation of China Grant No. 12101402, the National Key R&D Program of China Grant No. 2022YFA1008200, the Lingang Laboratory Grant No. LG-QS-202202-08, Shanghai Municipal of Science and Technology Major Project No. 2021SHZDZX0102.

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

# A  Proofs of Theoretical Results

In this section, we give all proofs for our theoretical results mentioned in the main text.

## A.1  Hierarchical Intrinsic Invariant Manifold and Sub Invariant Manifold

**Proposition A.1** (**Hierarchical Intrinsic Invariant Manifold (HIIM)**)**.** *Let $f_{\boldsymbol{\theta}} = \boldsymbol{A}\boldsymbol{B}$ be a matrix factorization model and $\{\boldsymbol{\alpha}_1, \cdots, \boldsymbol{\alpha}_k\}$ be $k$ linearly independent vectors. Define the manifold $\boldsymbol{\Omega}_k$ as $\boldsymbol{\Omega}_k := \boldsymbol{\Omega}_k(\boldsymbol{\alpha}_1, \cdots, \boldsymbol{\alpha}_k) = \{\boldsymbol{\theta} = (\boldsymbol{A}, \boldsymbol{B}) \mid \mathrm{row}(\boldsymbol{A}) = \mathrm{col}(\boldsymbol{B}) = \mathrm{span}\{\boldsymbol{\alpha}_1, \cdots, \boldsymbol{\alpha}_k\}\}$. The manifold $\boldsymbol{\Omega}_k$ possesses the following properties:*

*(i) **Invariance under Gradient Flow:** Given data $S$ and the gradient flow dynamics $\dot{\boldsymbol{\theta}} = -\nabla R_S(\boldsymbol{\theta})$, if the initial point $\boldsymbol{\theta}_0 \in \boldsymbol{\Omega}_k$, then $\boldsymbol{\theta}(t) \in \boldsymbol{\Omega}_k$ for all $t \geq 0$.*

*(ii) **Intrinsic Property:** $\boldsymbol{\Omega}_k$ is a data-independent invariant manifold, meaning that for any data $S$, $\boldsymbol{\Omega}_k$ remains invariant under the gradient flow dynamics.*

*(iii) **Hierarchical Structure:** The manifolds $\boldsymbol{\Omega}_k$ form a hierarchy: $\boldsymbol{\Omega}_0 \subsetneqq \boldsymbol{\Omega}_1 \subsetneqq \cdots \subsetneqq \boldsymbol{\Omega}_{k-1} \subsetneqq \boldsymbol{\Omega}_k$.*

*Proof.* (i) Invariance under Gradient Flow.

By definition, $\boldsymbol{\Omega}_k := \boldsymbol{\Omega}_k(\boldsymbol{\alpha}_1, \cdots, \boldsymbol{\alpha}_k) = \{\boldsymbol{\theta} = (\boldsymbol{A}, \boldsymbol{B}) \mid \mathrm{row}(\boldsymbol{A}) = \mathrm{col}(\boldsymbol{B}) = \mathrm{span}\{\boldsymbol{\alpha}_1, \cdots, \boldsymbol{\alpha}_k\}\}$. Consider the gradient flow dynamics in (8):

$$\begin{cases} \dot{\boldsymbol{a}}_i = -\dfrac{2}{n} \sum_{j \in I_i} (\boldsymbol{a}_i \cdot \boldsymbol{b}_{\cdot,j} - M_{ij}) \boldsymbol{b}_{\cdot,j}^\top, \\ \dot{\boldsymbol{b}}_{\cdot,j} = -\dfrac{2}{n} \sum_{i \in I_j} (\boldsymbol{a}_i \cdot \boldsymbol{b}_{\cdot,j} - M_{ij}) \boldsymbol{a}_i^\top, \end{cases} \tag{8}$$

where $I_i = \{j | \exists i : (i,j) \in S_{\boldsymbol{x}}\}$, $I_j = \{i | \exists j : (i,j) \in S_{\boldsymbol{x}}\}$, $\boldsymbol{a}_i$ and $\boldsymbol{b}_{\cdot,j}$ represent the $i$-th row and $j$-th column of $\boldsymbol{A}$ and $\boldsymbol{B}$, respectively.

For any $(i,j) \in S_{\boldsymbol{x}}$, the evolution of $\boldsymbol{a}_i$ is coupled with $\boldsymbol{b}_{\cdot,j}$ for $j \in I_i$. The condition $\mathrm{row}(\boldsymbol{A}) = \mathrm{col}(\boldsymbol{B}) = \mathrm{span}\{\boldsymbol{\alpha}_1, \cdots, \boldsymbol{\alpha}_k\}$ ensures the existence of $k$ linearly independent vectors $\boldsymbol{\alpha}_1, \cdots, \boldsymbol{\alpha}_k \in \mathbb{R}^d$ such that $\boldsymbol{a}_i, \boldsymbol{b}_{\cdot,j} \in \mathrm{span}\{\boldsymbol{\alpha}_1, \cdots, \boldsymbol{\alpha}_k\}$ for all $1 \leq i, j \leq d$.

Consequently, if $\boldsymbol{a}_i$ and $\boldsymbol{b}_{\cdot,j}$ are initially in $\mathrm{span}\{\boldsymbol{\alpha}_1, \boldsymbol{\alpha}_2, \cdots, \boldsymbol{\alpha}_k\}$, they will continue to evolve within this subspace under the gradient flow dynamics. Additionally, for $(i,j) \notin S_{\boldsymbol{x}}$, the gradients for the corresponding $\boldsymbol{a}_i$ and $\boldsymbol{b}_{\cdot,j}$ will be zero, provided their initial values are zero, maintaining this state throughout the evolution.

(ii) Intrinsic Property.

As demonstrated in part (i), $\boldsymbol{\Omega}_k$ is invariant under gradient flow dynamics for any dataset $S$, confirming its status as a data-independent invariant manifold.

(iii) Hierarchical Structure.

Th invariant manifold $\boldsymbol{\Omega}_k$ encompasses matrices of rank up to $k$, including those of lower ranks. Consequently, the manifolds exhibit the following hierarchical nesting:

$$\boldsymbol{\Omega}_0 \subsetneqq \boldsymbol{\Omega}_1 \subsetneqq \cdots \subsetneqq \boldsymbol{\Omega}_{k-1} \subsetneqq \boldsymbol{\Omega}_k.$$

$\square$

**Proposition A.2** (**Intrinsic Sub-$\boldsymbol{\Omega}_k$ Invariant Manifold**)**.** *Let $f_{\boldsymbol{\theta}} = \boldsymbol{A}\boldsymbol{B}$ be a matrix factorization model, $\boldsymbol{M}$ be an incomplete matrix and $\boldsymbol{\Omega}_k$ be an invariant manifold defined in Prop. 1. If $\boldsymbol{M}$ is disconnected with $m$ connected components, then there exist $m$ sub-$\boldsymbol{\Omega}_k$ manifolds $\boldsymbol{\omega}_k$ such that $\boldsymbol{\omega}_k \subsetneqq \boldsymbol{\Omega}_k$, each possessing the following properties:*

*(i) **Invariance under Gradient Flow:** Given data $S$ and the gradient flow dynamics $\dot{\boldsymbol{\theta}} = -\nabla R_S(\boldsymbol{\theta})$, if the initial point $\boldsymbol{\theta}_0 \in \boldsymbol{\omega}_k$, then $\boldsymbol{\theta}(t) \in \boldsymbol{\omega}_k$ for all $t \geq 0$.*

*(ii) **Intrinsic Property:** $\boldsymbol{\omega}_k$ is a data-value-independent invariant manifold, meaning that for a fixed sampling pattern in $\boldsymbol{M}$ and any observed values $S$, $\boldsymbol{\omega}_k$ remains invariant under the gradient flow.*

*(iii) **Strict Subset Relation:** The output set $\{f_\theta \mid \theta \in \omega_k\}$ is a proper subset of $\{f_\theta \mid \theta \in \Omega_k\}$, namely, $\{f_\theta \mid \theta \in \omega_k\} \subsetneqq \{f_\theta \mid \theta \in \Omega_k\}$.*

*Proof.* Existence.

Let us consider an incomplete matrix $\boldsymbol{M}$ whose associated observational graph is divided into $m$ connected components, denoted by $L_1, L_2, \ldots, L_m$. For each component $L_p$, we define $S_{\boldsymbol{x}}^{L_p}$ as the subset of observed indices within $L_p$, where $1 \le p \le m$ and $S_{\boldsymbol{x}}$ is the set of all observed indices.

For each $L_p$, we can identify row indices $R_p$ and column indices $C_p$ corresponding to the observed entries in $L_p$ as follows:

$$R_p = \{i | \exists j : (i,j) \in S_{\boldsymbol{x}}^{L_p}\}, \quad C_p = \{j | \exists i : (i,j) \in S_{\boldsymbol{x}}^{L_p}\}.$$

Here, $R_p$ includes the row indices and $C_p$ includes the column indices of the entries observed in $L_p$.

Define $\boldsymbol{A}^{L_p}$ and $\boldsymbol{B}^{L_p}$ as the submatrices of $\boldsymbol{A}$ and $\boldsymbol{B}$ corresponding to $R_p$ and $C_p$, respectively, and let $\boldsymbol{A}_r^{L_p}$ and $\boldsymbol{B}_r^{L_p}$ be the remaining rows not in $R_p$ and $C_p$.

Let $\boldsymbol{\Omega}_k := \boldsymbol{\Omega}_k(\boldsymbol{\alpha}_1, \cdots, \boldsymbol{\alpha}_k) = \{\boldsymbol{\theta} = (\boldsymbol{A}, \boldsymbol{B}) \mid \mathrm{row}(\boldsymbol{A}) = \mathrm{col}(\boldsymbol{B}) = \mathrm{span}\{\boldsymbol{\alpha}_1, \cdots, \boldsymbol{\alpha}_k\}\}$ be the given $\boldsymbol{\Omega}_k$ invariant manifold.

The sub-$\boldsymbol{\Omega}_k$ invariant manifold associated with the connected component $L_p$ can be defined as

$$\boldsymbol{\omega}_k^{L_p} := \{(\boldsymbol{\theta} = (\boldsymbol{A}, \boldsymbol{B})) \mid \mathrm{row}(\boldsymbol{A}^{L_p}) = \mathrm{col}((\boldsymbol{B}^{L_p})) = \mathrm{span}\{\boldsymbol{\alpha}_1, \cdots, \boldsymbol{\alpha}_k\}, \boldsymbol{A}_r^{L_p} = \boldsymbol{B}_r^{L_p} = \boldsymbol{0}\}. \tag{9}$$

It is easy to check $\boldsymbol{\omega}_k^{L_p}$ is a proper subset of $\boldsymbol{\Omega}_k$.

(i) Invariance under Gradient Flow.

The condition $\mathrm{row}(\boldsymbol{A}^{L_p}) = \mathrm{col}((\boldsymbol{B}^{L_p})) = \mathrm{span}\{\boldsymbol{\alpha}_1, \cdots, \boldsymbol{\alpha}_k\}$ along with $\boldsymbol{A}_r^{L_p} = \boldsymbol{B}_r^{L_p} = \boldsymbol{0}$ guarantees that $\boldsymbol{a}_i, \boldsymbol{b}_{\cdot,j} \in \mathrm{span}\{\boldsymbol{\alpha}_1, \ldots, \boldsymbol{\alpha}_k\}$ for all $(i,j) \in S_{\boldsymbol{x}}^{L_p}$, and $\boldsymbol{a}_i, \boldsymbol{b}_{\cdot,j} = \boldsymbol{0}$ for all $(i,j) \notin S_{\boldsymbol{x}}^{L_p}$.

In other words, the sub-$\boldsymbol{\Omega}_k$ invariant manifold $\boldsymbol{\omega}_k^{L_p}$ is the set of all pairs $(\boldsymbol{A}, \boldsymbol{B})$ where, for each observed position $(i,j)$ in the connected component $L_p$, the vectors $\boldsymbol{a}_i$ and $\boldsymbol{b}_{\cdot,j}$ lie within the span of $\{\boldsymbol{\alpha}_1, \cdots, \boldsymbol{\alpha}_k\}$, and for any position not in $S_{\boldsymbol{x}}^{L_p}$, the vectors are zero.

Considering the dynamics expressed in equation (8), it is evident that the evolution of $\boldsymbol{a}_i$ is influenced by $\boldsymbol{b}_{\cdot,j}$ for $(i,j) \in S_{\boldsymbol{x}}^{L_p}$. Hence, if $\boldsymbol{a}_i$ and $\boldsymbol{b}_{\cdot,j}$ are initially in the span of $\{\boldsymbol{\alpha}_1, \boldsymbol{\alpha}_2, \cdots, \boldsymbol{\alpha}_k\}$, they will continue to evolve within this span under the gradient flow dynamics. Moreover, for positions $(i,j) \notin S_{\boldsymbol{x}}^{L_p}$, we consider the following scenarios:

- For $(i,j) \notin S_{\boldsymbol{x}}$, since the matrix entry $\boldsymbol{M}_{ij}$ does not contribute to the loss $R_S(\boldsymbol{\theta})$, the gradients for corresponding $\boldsymbol{a}_i$ and $\boldsymbol{b}_{\cdot,j}$ will perpetually be zero. Thus, if their initial values are zero, they will remain zero throughout the evolution.

- For $(i,j) \in S_{\boldsymbol{x}}$ but not in $S_{\boldsymbol{x}}^{L_p}$, the dynamics corresponding to different connected components are decoupled. Therefore, if the initial values for $\boldsymbol{a}_i$ and $\boldsymbol{b}_{\cdot,j}$ are zero, they will stay zero during the evolution.

(ii) Intrinsic Property.

As established in (i), the manifold $\boldsymbol{\omega}_k^{L_p}$ is invariant under gradient flow for any data $S$ with a fixed sampling pattern, qualifying it as a data-value-independent invariant manifold.

(iii) Strict Subset Relation.

The output set $\{f_\theta \mid \theta \in \Omega_k\}$ encompasses all matrices of rank $k$, whereas $\{f_\theta \mid \theta \in \omega_k^{L_p}\}$ is limited to rank-$k$ matrices with specific row and column indices confined to $R_p$ and $C_p$. Consequently, $\{f_\theta \mid \theta \in \omega_k\}$ forms a strict subset of $\{f_\theta \mid \theta \in \Omega_k\}$, as stated by $\{f_\theta \mid \theta \in \omega_k\} \subsetneqq \{f_\theta \mid \theta \in \Omega_k\}$. $\qquad\square$

## A.2 Connectivity

**Definition A.1** (**Associated Observation Graph**). *Given a incomplete matrix $M$ to be completed and its observation matrix $P$, the associated observation graph $G_M$ is the bipartite graph with adjacency matrix $\begin{bmatrix} 0 & P^\top \\ P & 0 \end{bmatrix}$, with isolated vertices removed.*

**Definition A.2** (**Connectivity**). *A matrix $M$ to be completed is considered connected if its associated observation graph $G_M$ is connected, otherwise, we call it disconnected. The connected components of $M$ are defined as the connected components of this graph.*

**Definition A.3** (**Disconnectivity with Complete Bipartite Components**). *A matrix $M$ to be completed is considered disconnected with complete bipartite components if its associated observation graph $G_M$ is disconnected and each connected component forms a complete bipartite subgraph.*

**Remark.** *In the bipartite graph representation of the observed data, isolated vertices correspond to entire rows or columns of the matrix $M$ that are not observed. These rows or columns do not contribute to the loss calculation and have no influence on the dynamics of the matrix factorization under infinitesimal initialization. Consequently, when analyzing the connectivity of the observed data and its impact on the learning dynamics, these isolated vertices can be safely disregarded.*

**Remark.** *The disconnectivity of the bipartite graph representing the observed data is equivalent to the reducibility of the adjacency matrix $\begin{bmatrix} 0 & P^\top \\ P & 0 \end{bmatrix}$, where $P$ is the binary observation matrix indicating the positions of the observed entries in $M$.*

*In the context of matrix completion problems, such as the Netflix problem, connectivity has a practical interpretation. Connected components in the bipartite graph indicate groups of users and movies that are linked by the users' viewing history. Users within the same connected component are related through the movies they have watched in common. Due to this practical significance, we prefer to use the term "connectivity" instead of "reducibility" when discussing the structure of the observed data in matrix completion problems.*

**Definition A.4** (**Connectivity of Observed Data**). *Given a matrix $M$ to be completed, an undirected simple graph $G$ can be induced from it: the nodes of the graph are the observed elements in the matrix, and two nodes are adjacent if and only if they are in the same row or column of the matrix $M$. A matrix $M$ to be completed is considered connected if its induced graph $G$ is connected, otherwise, we call it disconnected.*

**Lemma A.1.** *For any simple graph $G$, if we remove all isolated vertices from $G$ to obtain a new graph $G'$, then $G'$ is connected if and only if the line graph of $G'$, denoted as $L(G')$, is connected.*

*Proof.* $\implies$ Assume $G'$ is connected. Consider any two nodes in $L(G')$, which correspond to two edges in $G'$, say $e_1$ and $e_2$. Since $G'$ is connected, there exists a path connecting the endpoints of $e_1$ and $e_2$. This path corresponds to a sequence of edges in $G'$, which in turn corresponds to a path connecting the nodes representing $e_1$ and $e_2$ in $L(G')$. Therefore, $L(G')$ is connected.

$\impliedby$ Conversely, assume $L(G')$ is connected. Consider any two vertices $v_1$ and $v_2$ in $G'$. Since $G'$ has no isolated vertices, each of $v_1$ and $v_2$ is incident to at least one edge. Let these edges be $e_1$ and $e_2$, respectively. Since $L(G')$ is connected, there exists a path connecting the nodes representing $e_1$ and $e_2$ in $L(G')$. This path corresponds to a sequence of edges in $G'$, which in turn corresponds to a path connecting $v_1$ and $v_2$ in $G'$. Therefore, $G'$ is connected.

In conclusion, we have proven that for any simple graph $G$, if we remove all isolated vertices from $G$ to obtain a new graph $G'$, then $G'$ is connected if and only if the line graph of $G'$, denoted as $L(G')$, is connected. $\square$

**Proposition A.3.** *Given a incomplete matrix $M$, the connectivity of $M$ defined in Def. A.2 and Def. A.4 is equivalent.*

*Proof.* By definition, each edge of a bipartite graph corresponds to an observed data item, and two edges in a bipartite graph are adjacent if and only if the two corresponding observed data items are in the same row or column. Therefore, the connectivity of the observed data is equivalent to the connectivity of the edges of the bipartite graph, which is, in turn, equivalent to the connectivity of the line graph of the bipartite graph.

According to Lem. A.1, for any graph $G$, if we remove all isolated vertices from $G$ to obtain a new graph $G'$, then $G'$ is connected if and only if the line graph of $G'$, denoted as $L(G')$, is connected.

In the context of the bipartite graph representation of the observed data, removing isolated vertices corresponds to removing rows and columns that contain no observed entries. Thus, the connectivity of the bipartite graph after removing isolated vertices is equivalent to the connectivity of the observed data as defined in Def. A.4.

Consequently, the connectivity of the observed data as defined in Def. A.2 (based on the line graph of the bipartite graph) is equivalent to the connectivity of the observed data as defined in Def. A.4 (based on the connectivity of observed data). $\qquad\square$

**Definition A.5** (**Decoupling of Dynamics**). *Given an incomplete matrix $\boldsymbol{M}$, consider the gradient flow dynamics of matrix factorization models, $\forall 1 \leq i, j \leq d$,*

$$
\begin{cases}
\dot{\boldsymbol{a}}_i = -\dfrac{2}{n} \sum_{j \in I_i} (\boldsymbol{a}_i \cdot \boldsymbol{b}_{\cdot,j} - \boldsymbol{M}_{ij}) \boldsymbol{b}_{\cdot,j}^\top, \\[2mm]
\dot{\boldsymbol{b}}_{\cdot,j} = -\dfrac{2}{n} \sum_{i \in I_j} (\boldsymbol{a}_i \cdot \boldsymbol{b}_{\cdot,j} - \boldsymbol{M}_{ij}) \boldsymbol{a}_i^\top.
\end{cases}
\tag{10}
$$

*The dynamics are said to be decoupled if there exist disjoint subsets of indices $R_1, R_2, \ldots, R_k \subseteq \{1, 2, \ldots, d\}$ for the rows of $\boldsymbol{A}$ and $C_1, C_2, \ldots, C_k \subseteq \{1, 2, \ldots, d\}$ for the columns of $\boldsymbol{B}$, such that for each $l \in \{1, 2, \ldots, k\}$, the dynamics of $\{\boldsymbol{a}_i : i \in R_l\}$ and $\{\boldsymbol{b}_{\cdot,j} : j \in C_l\}$ form an independent system of equations. In other words, the dynamics can be divided into $k(k > 1)$ independent subsystems, each involving a subset of rows of $\boldsymbol{A}$ and a subset of columns of $\boldsymbol{B}$. If such a division is not possible, the dynamics are said to be coupled.*

**Proposition A.4.** *Given an incomplete matrix $\boldsymbol{M}$, if it is disconnected as defined by Def. A.2, then the dynamics are decoupled as defined by Def. A.5; if it is connected as defined by Def. A.2, then the dynamics are coupled as defined by Def. A.5.*

*Proof.* Consider a matrix $\boldsymbol{M}$ to be completed, with its associated observation graph comprising $m$ connected components, denoted as $L_1, L_2, \cdots, L_m$. Let $S_{\boldsymbol{x}}^{L_p} \subseteq S_{\boldsymbol{x}}$ represent the subset of observed indices corresponding to the connected component $L_p$, where $1 \leq p \leq m$ and $S_{\boldsymbol{x}}$ denotes the complete set of observed indices. If the incomplete matrix $\boldsymbol{M}$ is disconnected, then for each connected component $L_p$, the subset $S_{\boldsymbol{x}}^{L_p}$ can be partitioned into two subsets $R_p$ and $C_p$, $1 \leq p \leq m$, such that

$$
R_p = \{i | \exists j : (i, j) \in S_{\boldsymbol{x}}^{L_p}\}, \quad C_p = \{j | \exists i : (i, j) \in S_{\boldsymbol{x}}^{L_p}\}.
\tag{11}
$$

In other words, $R_p$ contains the row indices and $C_p$ contains the column indices of the observed entries in the connected component $L_p$.

It can be easily verified that the dynamics are decoupled in this case, as the subsets $\{R_p, C_p\}_{p=1}^m$ satisfy the conditions in Def. A.5. Each connected component $L_p$ corresponds to an independent subsystem involving the rows of $\boldsymbol{A}$ indexed by $R_p$ and the columns of $\boldsymbol{B}$ indexed by $C_p$.

If $\boldsymbol{M}$ is connected, then its associated observation graph consists of a single connected component, and the entire dynamics are coupled. $\qquad\square$

**Examples of connectivity and disconnectivity.** Consider three matrices to be completed, each obtained by adding one more observation to the previous matrix: $\boldsymbol{M}_1$ (disconnected), $\boldsymbol{M}_2$ (disconnected with complete bipartite components), and $\boldsymbol{M}_3$ (connected).

$$
\boldsymbol{M}_1 = \begin{bmatrix} 1 & 2 & \star \\ 3 & \star & \star \\ \star & \star & 5 \end{bmatrix}, \boldsymbol{M}_2 = \begin{bmatrix} 1 & 2 & \star \\ 3 & 4 & \star \\ \star & \star & 5 \end{bmatrix}, \boldsymbol{M}_3 = \begin{bmatrix} 1 & 2 & \star \\ 3 & 4 & \star \\ 6 & \star & 5 \end{bmatrix}
\tag{12}
$$

The observation matrix P is:

$$
\boldsymbol{P}_1 = \begin{bmatrix} 1 & 1 & 0 \\ 1 & 0 & 0 \\ 0 & 0 & 1 \end{bmatrix}, \boldsymbol{P}_2 = \begin{bmatrix} 1 & 1 & 0 \\ 1 & 1 & 0 \\ 0 & 0 & 1 \end{bmatrix}, \boldsymbol{P}_3 = \begin{bmatrix} 1 & 1 & 0 \\ 1 & 1 & 0 \\ 1 & 0 & 1 \end{bmatrix}
\tag{13}
$$

And the adjacency matrix is:

$$A_1 = \begin{bmatrix} \mathbf{0} & P_1^\top \\ P_1 & \mathbf{0} \end{bmatrix}, A_2 = \begin{bmatrix} \mathbf{0} & P_2^\top \\ P_2 & \mathbf{0} \end{bmatrix}, A_3 = \begin{bmatrix} \mathbf{0} & P_3^\top \\ P_3 & \mathbf{0} \end{bmatrix} \tag{14}$$

Given the adjacency matrix $A$, we can obtain a graph $G_M$. Fig. A1 illustrates the associated graphs $G_M$, from which we can see that $M_1$ is disconnected, with its associated observation graph consisting of two connected components. $M_2$ is also disconnected, but each connected component of its associated observation graph forms a complete bipartite subgraph. In contrast, $M_3$ is connected, and its associated observation graph consists of a single connected component.

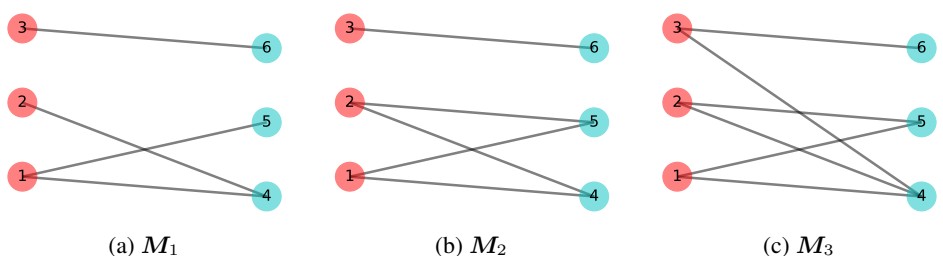

(a) $M_1$      (b) $M_2$      (c) $M_3$

Figure A1: **The associated observation graphs $G_M$ of the incomplete matrices $M_1$, $M_2$, and $M_3$ in Eq. 4.** $M_1$ is disconnected, with its associated observation graph consisting of two connected components. $M_2$ is also disconnected, but each connected component of its associated observation graph forms a complete bipartite subgraph. In contrast, $M_3$ is connected, and its associated observation graph consists of a single connected component.

### A.3 Loss Landscape

In this paper, we focus on the problem of asymmetric matrix factorization. Previous literature (Gunasekar et al., 2017; Li et al., 2018, 2020; Jin et al., 2023) has predominantly concentrated on symmetric matrix factorization problems. Although asymmetric matrix factorization models can be transformed into symmetric cases, studying symmetric matrix factorization does not necessarily cover all aspects of the asymmetric scenarios.

Generally, an asymmetric matrix factorization model $W = AB$ can be transformed into a symmetric situation by setting

$$U = \begin{bmatrix} A \\ B^\top \end{bmatrix} \in \mathbb{R}^{2d \times d}.$$

We then consider the model $W' = UU^\top$, which corresponds to the following matrix completion problem:

$$\begin{bmatrix} AA^\top & AB \\ B^\top A^\top & B^\top B \end{bmatrix}.$$

We define the loss as

$$\mathcal{L}'\left(\begin{bmatrix} A & B \\ C & D \end{bmatrix}\right) = \frac{1}{2}\mathcal{L}(B) + \frac{1}{2}\mathcal{L}(C^\top).$$

Li et al. (2020) established the following results:

**Theorem A.1** (Theorem 5.10 in Li et al. (2020)). *Let $f : \mathbb{R}^{d \times d} \to \mathbb{R}$ be a convex $\mathcal{C}^2$-smooth function. (1). All stationary points of $\mathcal{L} : \mathbb{R}^{d \times d} \to \mathbb{R}, \mathcal{L}(U) = \frac{1}{2}f\left(UU^\top\right)$ are either strict saddles or global minimizers; (2). For any random initialization, $GF(1)$ converges to strict saddles of $\mathcal{L}(U)$ with probability 0.*

The proof of this theorem relies heavily on Theorem A.2 of Du and Lee (2018), which requires the parameter matrix $U \in \mathbb{R}^{d \times k}$ to satisfy the condition that $k \geq d$. In the case of symmetric matrix factorization, where $U \in \mathbb{R}^{d \times d}$, this condition is naturally met. However, for asymmetric matrix factorization, where $U = \begin{bmatrix} A \\ B^\top \end{bmatrix} \in \mathbb{R}^{2d \times d}$, this condition is not satisfied, and thus the proof of Theorem A.1 is only applicable to the symmetric case of matrix factorization.

**Theorem A.2** (Theorem 3.1 in Du and Lee (2018))**.** *Let* $f : \mathbb{R}^{d \times d} \to \mathbb{R}$ *be a* $\mathcal{C}^2$ *convex function. Then* $\mathcal{L} : \mathbb{R}^{d \times k} \to \mathbb{R}, \mathcal{L}(\boldsymbol{U}) = f\left(\boldsymbol{U}\boldsymbol{U}^\top\right), k \geq d$ *satisfies that (1). Every local minimizer of* $\mathcal{L}$ *is also a global minimizer; (2). All saddles are strict. Here saddles denote those stationary points whose hessian are not positive semi-definite (thus including local maximizers).*

Below we give a direct proof of the loss landscape of an asymmetric matrix factorization model.

**Theorem A.3** (**Loss Landscape**)**.** *For any data $S$, the critical points of $R_S(\boldsymbol{\theta})$ are either strict saddle points or global minima.*

*Proof.* We start by recalling the definition of the loss function:

$$R_S(\boldsymbol{\theta}) = \mathcal{L}(\boldsymbol{A}, \boldsymbol{B}) = \frac{1}{2}\|\boldsymbol{AB} - \boldsymbol{M}\|_{S_x}^2 = \frac{1}{2}\sum_{(i,j) \in S_x}((\boldsymbol{AB})_{ij} - \boldsymbol{M}_{ij})^2.$$

Let $\boldsymbol{\theta} = (\boldsymbol{A}, \boldsymbol{B})$ denote a critical point. We define a new matrix, $\delta\boldsymbol{M}$, as the difference between the product of $\boldsymbol{A}$ and $\boldsymbol{B}$ and the matrix $\boldsymbol{M}$, with this difference being computed only over the indices in the set $S_x$. More formally, $\delta\boldsymbol{M} = (\boldsymbol{AB} - \boldsymbol{M})_{S_x}$, where the elements of $\delta\boldsymbol{M}$ are given by:

- For $(i,j) \in S_x$, we have $\delta\boldsymbol{M}_{ij} = (\boldsymbol{AB})_{ij} - \boldsymbol{M}_{ij}$.

- For $(i,j) \notin S_x$, we have $\delta\boldsymbol{M}_{ij} = 0$.

This definition of $\delta\boldsymbol{M}$ ensures that we only consider the differences in the entries that belong to the set $S_x$, while all other entries are set to zero.

Consider the function:

$$\begin{aligned}
\mathcal{L}(\boldsymbol{A} + \boldsymbol{\varepsilon}, \boldsymbol{B} + \boldsymbol{\eta}) &= \frac{1}{2}\|\delta\boldsymbol{M} + \boldsymbol{\varepsilon}\boldsymbol{B} + \boldsymbol{A}\boldsymbol{\eta} + \boldsymbol{\varepsilon}\boldsymbol{\eta}\|_{S_x}^2 \\
&= \frac{1}{2}\|\delta\boldsymbol{M}\|_{S_x}^2 + \langle\delta\boldsymbol{M}, \boldsymbol{\varepsilon}\boldsymbol{B} + \boldsymbol{A}\boldsymbol{\eta}\rangle_{S_x} + \frac{1}{2}\|\boldsymbol{\varepsilon}\boldsymbol{B} + \boldsymbol{A}\boldsymbol{\eta}\|_{S_x}^2 \qquad (15) \\
&\quad + \langle\delta\boldsymbol{M}, \boldsymbol{\varepsilon}\boldsymbol{\eta}\rangle_{S_x} + o(\|\boldsymbol{\varepsilon}\|^2, \|\boldsymbol{\eta}\|^2),
\end{aligned}$$

where the inner product of two matrices $\boldsymbol{A}, \boldsymbol{B}$ is defined as $\langle\boldsymbol{A}, \boldsymbol{B}\rangle := \text{Tr}(\boldsymbol{A}\boldsymbol{B}^\top)$.

At the critical point, the first order term $\langle\delta\boldsymbol{M}, \boldsymbol{\varepsilon}\boldsymbol{B} + \boldsymbol{A}\boldsymbol{\eta}\rangle_{S_x}$ equals 0. The Hessian operator, representing the second order term, is given by:

$$h_{\boldsymbol{A},\boldsymbol{B}}(\boldsymbol{\varepsilon}, \boldsymbol{\eta}) = \frac{1}{2}\|\boldsymbol{\varepsilon}\boldsymbol{B} + \boldsymbol{A}\boldsymbol{\eta}\|_{S_x}^2 + \langle\delta\boldsymbol{M}, \boldsymbol{\varepsilon}\boldsymbol{\eta}\rangle_{S_x}.$$

Our goal is to demonstrate that if $\delta\boldsymbol{M} \neq \boldsymbol{0}$, there always exists $\boldsymbol{\varepsilon}, \boldsymbol{\eta}$ such that $h_{\boldsymbol{A},\boldsymbol{B}}(\boldsymbol{\varepsilon}, \boldsymbol{\eta}) < 0$. To this end, we consider the ranks of matrices $\boldsymbol{A}$ and $\boldsymbol{B}$ in two cases:

(i) $\text{rank}(\boldsymbol{A}) < d$ or $\text{rank}(\boldsymbol{B}) < d$:

Without loss of generality, we assume $\delta\boldsymbol{M}_{ij} := \delta\boldsymbol{M}_{ij} \neq 0$ for some $(i,j) \in S_x$, and $\text{rank}\boldsymbol{A} < d$. Under these conditions, there exists a non-zero vector $\boldsymbol{v}$ such that $\boldsymbol{A}\boldsymbol{v} = 0$.

We set $\boldsymbol{\eta}_{,j}^* = \boldsymbol{v}$ and $\boldsymbol{\eta}_{,s}^* = 0$ for $s \neq j$, where $\boldsymbol{\eta}_{,j}^*$ denotes the $j$-th column of the matrix $\boldsymbol{\eta}$. Let $\boldsymbol{\varepsilon}_i^* = \boldsymbol{w}^\top \in \mathbb{R}^d$ and $\boldsymbol{\varepsilon}_s = 0$ for $s \neq i$, where $\boldsymbol{\varepsilon}_i^*$ denotes the $i$-th row of the matrix $\boldsymbol{\varepsilon}$.

We then have:

$$\begin{aligned}
h_{\boldsymbol{A},\boldsymbol{B}}(\boldsymbol{\varepsilon}^*, \boldsymbol{\eta}^*) &= \frac{1}{2}\|\boldsymbol{\varepsilon}^*\boldsymbol{B}\|_{S_x}^2 + \delta\boldsymbol{M}_{ij}\boldsymbol{w}^\top\boldsymbol{v} \\
&\leq \frac{1}{2}\|\boldsymbol{w}^\top\boldsymbol{B}\|_F^2 + \delta\boldsymbol{M}_{ij}\boldsymbol{w}^\top\boldsymbol{v} \\
&= \frac{1}{2}\boldsymbol{w}^\top\boldsymbol{B}\boldsymbol{B}^\top\boldsymbol{w} + \delta\boldsymbol{M}_{ij}\boldsymbol{w}^\top\boldsymbol{v}.
\end{aligned}$$

We define $g(\boldsymbol{w}, \boldsymbol{v}) = \frac{1}{2}\boldsymbol{w}^\top \boldsymbol{B}\boldsymbol{B}^\top \boldsymbol{w} + \delta \boldsymbol{M}_{ij}\boldsymbol{w}^\top \boldsymbol{v}$ and consider:

$$
\begin{aligned}
g(-\alpha\delta\boldsymbol{M}_{ij}\boldsymbol{v}, \boldsymbol{v}) &= \frac{1}{2}\alpha^2\delta\boldsymbol{M}_{ij}^2\boldsymbol{v}^\top \boldsymbol{B}\boldsymbol{B}^\top\boldsymbol{v} - \alpha\delta\boldsymbol{M}_{ij}^2\boldsymbol{v}^\top\boldsymbol{v} \\
&= \frac{1}{2}\alpha^2\delta\boldsymbol{M}_{ij}^2(\boldsymbol{v}^\top\boldsymbol{B}\boldsymbol{B}^\top\boldsymbol{v} - 2\frac{1}{\alpha}\boldsymbol{v}^\top\boldsymbol{v}).
\end{aligned}
$$

For $0 < \alpha < \frac{2}{\lambda_{\boldsymbol{B},\max}}$, where $\lambda_{\boldsymbol{B},\max}$ represents the top eigenvalue of $\boldsymbol{B}\boldsymbol{B}^\top$, we find $g(-\alpha\delta\boldsymbol{M}_{ij}\boldsymbol{v}, \boldsymbol{v}) < 0$.

Therefore, when $\boldsymbol{w} = -\alpha\delta\boldsymbol{M}_{ij}\boldsymbol{v}$, we obtain $h_{\boldsymbol{A},\boldsymbol{B}}(\boldsymbol{\varepsilon}^*, \boldsymbol{\eta}^*) < 0$. This immediately implies the critical point $\boldsymbol{\theta} = (\boldsymbol{A}, \boldsymbol{B})$ is a strict saddle point.

(ii) $\text{rank}(\boldsymbol{A}) = \text{rank}(\boldsymbol{B}) = d$:

Let $\boldsymbol{\varepsilon} = \alpha\delta\boldsymbol{M}\boldsymbol{B}^{-1}$ and $\boldsymbol{\eta} = 0$. In this scenario, the first order term $\langle\delta\boldsymbol{M}, \boldsymbol{\varepsilon}\boldsymbol{B} + \boldsymbol{A}\boldsymbol{\eta}\rangle_{S_{\boldsymbol{x}}}$ in Eq. (15) simplifies to $\alpha\|\delta\boldsymbol{M}\|_{S_{\boldsymbol{x}}}^2$. At a critical point, this quantity equals zero, it implies that $\delta\boldsymbol{M} = 0$. This in turn implies that the critical point $\boldsymbol{\theta} = (\boldsymbol{A}, \boldsymbol{B})$ is a global minimum.

This concludes the proof, establishing that the critical points of $R_S(\boldsymbol{\theta})$ are either strict saddle points or global minima. $\qquad\square$

### A.4  Escaping from Top Eigendirection

In this section, we focus on the dynamics of escaping from a critical point. According to Prop. A.3, the loss landscape consists solely of strict saddle points and a global minimum. Consequently, gradient-based methods can readily escape from a critical point that is not a global minimum.

In the following, we will demonstrate that the escaping dynamics near a critical point can be approximated by a linearized version of these dynamics. For this, consider the following:

$$\dot{\boldsymbol{\theta}} = -\nabla R_S(\boldsymbol{\theta}). \tag{16}$$

Assume $\boldsymbol{\theta}_c$ is a saddle point for which $\nabla R_S(\boldsymbol{\theta}_c) = 0$. We can apply a first-order Taylor expansion to the right-hand side of Eq. (16), yielding:

$$-\nabla R_S(\boldsymbol{\theta}) = -\nabla R_S(\boldsymbol{\theta}_c) - \nabla^2 R_S(\boldsymbol{\theta}_c)(\boldsymbol{\theta} - \boldsymbol{\theta}_c) + \mathcal{O}(\|\boldsymbol{\theta} - \boldsymbol{\theta}_c\|^2), \tag{17}$$

where $\nabla^2 R_S(\boldsymbol{\theta}_c)$ represents the Hessian matrix. Given that $\nabla R_S(\boldsymbol{\theta}_c) = 0$, the gradient flow dynamics around $\boldsymbol{\theta}_c$ can be approximated as:

$$\dot{\boldsymbol{\theta}} = \boldsymbol{H}(\boldsymbol{\theta} - \boldsymbol{\theta}_c), \tag{18}$$

where $\boldsymbol{H} := -\nabla^2 R_S(\boldsymbol{\theta}_c)$. Eq. (18) is a classic linear ordinary differential equation, with the solution:

$$\boldsymbol{\theta}(t) = \mathrm{e}^{t\boldsymbol{H}}(\boldsymbol{\theta}_0 - \boldsymbol{\theta}_c) + \boldsymbol{\theta}_c. \tag{19}$$

The dynamics near a critical point can be approximated by a linearized version. Hence, in the vicinity of a critical point, we can analyze the linearized dynamics to understand the escape mechanism. In the following, we will show that during this escape process, the dynamics follow a pattern referred to as *dominant eigenvalue dynamics*.

Eq. (19) elucidates that the dynamics near the critical point $\boldsymbol{\theta}_c$ are predominantly dictated by the properties of $\boldsymbol{H}$, a real symmetric matrix in $\mathbb{R}^{2d^2 \times 2d^2}$. Its eigendecomposition is given by:

$$\boldsymbol{H} := -\nabla^2 R_S(\boldsymbol{\theta}_c) = \boldsymbol{Q}\boldsymbol{\Lambda}\boldsymbol{Q}^\top, \tag{20}$$

where $\boldsymbol{\Lambda}$ is a diagonal matrix and $\boldsymbol{Q}$ is an orthogonal matrix. Let $\lambda_1 > \lambda_2 > \cdots > \lambda_s \in \mathbb{R}$ denote the eigenvalues of $\boldsymbol{H}$, and let $\boldsymbol{q}_{i1}, \boldsymbol{q}_{i2}, \cdots, \boldsymbol{q}_{il_i}$ represent the eigenvectors corresponding to $\lambda_i$.

Given that $\lambda_1 > \lambda_2$, the ratio $e^{\lambda_1 t}/e^{\lambda_i t}$ for $i > 1$ grows exponentially fast. Consequently, near $\boldsymbol{\theta}_c$, the evolution of the system is primarily driven by the eigenvectors $\boldsymbol{q}_{11}, \boldsymbol{q}_{12}, \cdots, \boldsymbol{q}_{1l_1}$ associated with the largest eigenvalue $\lambda_1$.

This following proposition formalizes the intuitive idea that in the vicinity of a saddle point, the dynamics primarily follow the direction associated with the largest eigenvalue. This leading eigendirection becomes increasingly dominant as time evolves, allowing for an escape from the saddle point and facilitating a specific structured transition.

Let's consider $\boldsymbol{\theta}_c$ to be a saddle point. Consider:

$$\dot{\boldsymbol{\theta}} = -\nabla^2 R_S(\boldsymbol{\theta}_c)(\boldsymbol{\theta} - \boldsymbol{\theta}_c), \quad \boldsymbol{\theta}(0) = \boldsymbol{\theta}_0. \tag{21}$$

Here, $\boldsymbol{\theta}_0$ and $\boldsymbol{\theta}_c$ are in close enough proximity for the linearized dynamics to be valid over a sufficiently long period. We can then establish the following proposition:

**Proposition A.5** (**Escape from Saddle Points Following a Dominant Eigenvalue Dynamics**). *Consider the linearized dynamics given by $\dot{\boldsymbol{\theta}} = -\nabla^2 R_S(\boldsymbol{\theta}_c)(\boldsymbol{\theta} - \boldsymbol{\theta}_c)$, we denote $\boldsymbol{H} := -\nabla^2 R_S(\boldsymbol{\theta}_c)$. Let $\lambda_1 \in \mathbb{R}$ be the largest eigenvalue of $\boldsymbol{H}$, with corresponding eigenvectors $\boldsymbol{q}_{11}, \boldsymbol{q}_{12}, \cdots, \boldsymbol{q}_{1l_1}$. Denote $c_j = \langle \boldsymbol{\theta}_0 - \boldsymbol{\theta}_c, \boldsymbol{q}_{1j} \rangle, \forall 1 \leq j \leq l_1$. Assume there exists $j$ such that $c_j \neq 0$, then, given any $\varepsilon > 0$, there exists a $t_0 > 0$ such that for all $t \geq t_0$, the following holds:*

$$\left\| \frac{\boldsymbol{\theta}(t) - \boldsymbol{\theta}_c}{\mathrm{e}^{\lambda_1 t} \sum_{j=1}^{l_1} c_j \boldsymbol{q}_{1j}} - \mathbf{1} \right\| < \varepsilon. \tag{22}$$

This means that, as time progresses, the direction of the parameter evolution increasingly aligns with the dominant eigenvectors of $\boldsymbol{H}$.

This proposition is a consequence of the fact that the solution of the differential equation is given by $\boldsymbol{\theta}(t) = e^{\boldsymbol{H}t}\boldsymbol{\theta}(0)$, and as $t$ tends to infinity, the term corresponding to the dominant eigenvalue in the matrix exponential $e^{\boldsymbol{H}t}$ becomes dominant. Therefore, near the saddle point, we have $\boldsymbol{\theta}(t) \approx \mathrm{e}^{\lambda_1 t} \sum_{j=1}^{l_1} c_j \boldsymbol{q}_{1j} + \boldsymbol{\theta}_c$.

*Proof.* The solution of the ordinary differential equation (21) is $\boldsymbol{\theta}(t) = \mathrm{e}^{t\boldsymbol{H}}(\boldsymbol{\theta}_0 - \boldsymbol{\theta}_c) + \boldsymbol{\theta}_c$. Here, $\boldsymbol{H} = -\nabla^2 R_S(\boldsymbol{\theta}_c)$ is a real symmetric matrix, which can be diagonalized. Let $\lambda_1 > \lambda_2 > \cdots > \lambda_s$ be the eigenvalues of $\boldsymbol{H}$, and let $\boldsymbol{q}_{i1}, \boldsymbol{q}_{i2}, \cdots, \boldsymbol{q}_{il_i}$ be the eigenvectors corresponding to $\lambda_i$. We can then express $\boldsymbol{\theta}(t)$ as:

$$\boldsymbol{\theta}(t) = \sum_{i=1}^{s} \sum_{j=1}^{l_i} \mathrm{e}^{\lambda_i t} \langle \boldsymbol{\theta}_0 - \boldsymbol{\theta}_c, \boldsymbol{q}_{ij} \rangle \boldsymbol{q}_{ij} + \boldsymbol{\theta}_c. \tag{23}$$

Next, we analyze the norm of the relative difference between $\boldsymbol{\theta}(t)$ and a term dominating its growth:

$$\left\| \frac{\boldsymbol{\theta}(t) - \boldsymbol{\theta}_c}{\sum_{j=1}^{l_1} \mathrm{e}^{\lambda_1 t} \langle \boldsymbol{\theta}_0 - \boldsymbol{\theta}_c, \boldsymbol{q}_{1j} \rangle \boldsymbol{q}_{1j}} - \mathbf{1} \right\| = \left\| \frac{\sum_{i=2}^{s} \sum_{j=1}^{l_i} \mathrm{e}^{\lambda_i t} \langle \boldsymbol{\theta}_0 - \boldsymbol{\theta}_c, \boldsymbol{q}_{ij} \rangle \boldsymbol{q}_{ij}}{\sum_{j=1}^{l_1} \mathrm{e}^{\lambda_1 t} \langle \boldsymbol{\theta}_0 - \boldsymbol{\theta}_c, \boldsymbol{q}_{1j} \rangle \boldsymbol{q}_{1j}} \right\|$$

$$\leq \sum_{i=2}^{s} \sum_{j=1}^{l_i} \mathrm{e}^{-(\lambda_1 - \lambda_i)t} \left\| \frac{\langle \boldsymbol{\theta}_0 - \boldsymbol{\theta}_c, \boldsymbol{q}_{ij} \rangle \boldsymbol{q}_{ij}}{\sum_{j=1}^{l_1} \langle \boldsymbol{\theta}_0 - \boldsymbol{\theta}_c, \boldsymbol{q}_{1j} \rangle \boldsymbol{q}_{1j}} \right\| \tag{24}$$

$$\leq \mathrm{e}^{-(\lambda_1 - \lambda_2)t} \sum_{i=2}^{s} \sum_{j=1}^{l_i} \left\| \frac{\langle \boldsymbol{\theta}_0 - \boldsymbol{\theta}_c, \boldsymbol{q}_{ij} \rangle \boldsymbol{q}_{ij}}{\sum_{j=1}^{l_1} \langle \boldsymbol{\theta}_0 - \boldsymbol{\theta}_c, \boldsymbol{q}_{1j} \rangle \boldsymbol{q}_{1j}} \right\|.$$

We define $C = \sum_{i=2}^{s} \sum_{j=1}^{l_i} \left\| \dfrac{\langle \boldsymbol{\theta}_0 - \boldsymbol{\theta}_c, \boldsymbol{q}_{ij} \rangle \boldsymbol{q}_{ij}}{\sum_{j=1}^{l_1} \langle \boldsymbol{\theta}_0 - \boldsymbol{\theta}_c, \boldsymbol{q}_{1j} \rangle \boldsymbol{q}_{1j}} \right\|$. By choosing $t_0 = \dfrac{\log \frac{C}{\varepsilon}}{\lambda_1 - \lambda_2}$, we ensure that for all $t > t_0$, the following condition is met:

$$\left\| \frac{\boldsymbol{\theta}(t) - \boldsymbol{\theta}_c}{\sum_{j=1}^{l_1} \mathrm{e}^{\lambda_1 t} \langle \boldsymbol{\theta}_0 - \boldsymbol{\theta}_c, \boldsymbol{q}_{1j} \rangle \boldsymbol{q}_{1j}} - \mathbf{1} \right\| < \varepsilon. \tag{25}$$

$\square$

Prop. A.5 describes that under the linearized dynamics, the parameters will escape from the saddle point along a specific direction. However, when considering the original nonlinear dynamics $\dot{\boldsymbol{\theta}}(t) = -\nabla R_S(\boldsymbol{\theta})$, we encounter a trade-off: we should choose $t_0$ sufficiently large so that the trajectory can align well with the dominant eigendirection while escaping the saddle point, but if $t_0$ is too large, the linearization approximation will fail as $\boldsymbol{\theta}(t_0)$ moves away from $\boldsymbol{\theta}_c$. Li et al. (2020) (Theorem 5.3) proved a general dynamical result through careful analysis and error control: assuming the eigenvector corresponding to the maximum eigenvalue is unique and the initialization is sufficiently close to the saddle point, there always exists a suitable $t_0$ such that the linear dynamics can align with the dominant eigendirection before the linearization breaks down. We can generalize this result to the case where the eigenvector corresponding to the largest eigenvalue is not unique:

**Proposition A.6.** *Consider the dynamics given by $\dot{\boldsymbol{\theta}}(t) = -\nabla R_S(\boldsymbol{\theta})$, we use $\varphi(\boldsymbol{\theta}_0, t)$ to denote the value of $\boldsymbol{\theta}(t)$ in the case of $\boldsymbol{\theta}(0) = \boldsymbol{\theta}_0$. At a critical point $\boldsymbol{\theta}_c$, we denote the negative Hessian as $\boldsymbol{H} := -\nabla^2 R_S(\boldsymbol{\theta}_c)$. Let $\lambda_1 \in \mathbb{R}$ be the largest eigenvalue of $\boldsymbol{H}$, with corresponding eigenvectors $\boldsymbol{q}_{11}, \boldsymbol{q}_{12}, \cdots, \boldsymbol{q}_{1l_1}$. Denote $c_j = \langle \boldsymbol{\theta}_0 - \boldsymbol{\theta}_c, \boldsymbol{q}_{1j} \rangle, \forall 1 \le j \le l_1$, and $\boldsymbol{v}_1 = \sum_{j=1}^{l_1} c_j \boldsymbol{q}_{1j}$. Assume there exists $j$ such that $c_j \ne 0$. Let $\boldsymbol{z}_\alpha(t) := \varphi\left(\boldsymbol{\theta}_c + \alpha \boldsymbol{v}_1, t + \frac{1}{\lambda_1} \log \frac{1}{\alpha}\right)$ for every $\alpha > 0$, then $\boldsymbol{z}(t) := \lim_{\alpha \to 0} \boldsymbol{z}_\alpha(t)$ exists and is also a solution of the given dynamics, i.e., $\boldsymbol{z}(t) = \varphi(\boldsymbol{z}(0), t)$. Furthermore, $\forall t \in \mathbb{R}$, there exists a constant $C > 0$ such that*

$$\left\| \varphi\left(\boldsymbol{\theta}_c + \alpha \boldsymbol{\theta}_0, t + \frac{1}{\lambda_1} \log \frac{1}{\alpha}\right) - \boldsymbol{z}(t) \right\|_2 \le C \alpha^{\frac{\lambda_1 - \lambda_2}{2\lambda_1 - \lambda_2}}$$

*for every sufficiently small $\alpha$, where $\lambda_1 - \lambda_2 > 0$ is the eigenvalue gap.*

*Proof.* By Theorem 5.3 in Section 5.1 of Li et al. (2020), we know that if the eigenspace corresponding to $\lambda_1$ is one-dimensional, i.e., $l_1 = 1$, then the escaping direction will be the top eigenvector direction, and the convergence rate is $O(\alpha^{\frac{\lambda_1 - \lambda_2}{2\lambda_1 - \lambda_2}})$. Therefore, Proposition A.6 holds in this case.

Now, if the eigenspace corresponding to $\lambda_1$ is not one-dimensional, we denote $c_j = \langle \boldsymbol{\theta}_0 - \boldsymbol{\theta}_c, \boldsymbol{q}_{1j} \rangle, \forall 1 \le j \le l_1$, and $\boldsymbol{v}_1 = \sum_{j=1}^{l_1} c_j \boldsymbol{q}_{1j}$ will be the escaping direction. Following the same technique as in Li et al. (2020), we can easily verify that the convergence rate remains $O(\alpha^{\frac{\lambda_1 - \lambda_2}{2\lambda_1 - \lambda_2}})$. Therefore, Proposition A.6 holds in this case as well.

$\square$

### A.5 Eigenvalues and Eigenvectors of Hessian

Suppose the dominant directions fulfill specific conditions, such as any combination $c_1 \boldsymbol{q}_{11} + c_2 \boldsymbol{q}_{12} + \cdots + c_{l_1} \boldsymbol{q}_{1l_1}$, leading to rank 1 model parameters $(\boldsymbol{A}, \boldsymbol{B})$. In such scenarios, we may observe a phenomenon where the rank of the matrix increases incrementally.

Firstly, we analyze the eigenvector structure of the Hessian matrix at the critical point $\boldsymbol{\theta}_c = (\boldsymbol{A}_c, \boldsymbol{B}_c)$ to understand why the parameter will enter the rank-1 invariant manifold.

**Computation of the Hessian Matrix at a Critical Point.** To compute the Hessian matrix, we first consider the gradient:

$$R_S(\boldsymbol{\theta}) = \mathbb{E}_S \ell\left(f(\boldsymbol{x}, \boldsymbol{\theta}), f^*(\boldsymbol{x})\right),$$

$$\nabla_{\boldsymbol{\theta}} R_S(\boldsymbol{\theta}) = \mathbb{E}_S \nabla \ell\left(\boldsymbol{f}(\boldsymbol{x}, \boldsymbol{\theta}), \boldsymbol{f}^*(\boldsymbol{x})\right)^\top \nabla_{\boldsymbol{\theta}} \boldsymbol{f}_{\boldsymbol{\theta}}(\boldsymbol{x}),$$

$$= \sum_{i=1}^{d^2} \mathbb{E}_S \partial_i \ell\left(\boldsymbol{f}_{\boldsymbol{\theta}}, \boldsymbol{f}^*\right) \nabla_{\boldsymbol{\theta}} \left(\boldsymbol{f}_{\boldsymbol{\theta}}\right)_i,$$

$$= \sum_{i=1}^{d^2} \mathbb{E}_S (\boldsymbol{f}_{\boldsymbol{\theta}} - \boldsymbol{f}^*)_i \nabla_{\boldsymbol{\theta}} \left(\boldsymbol{f}_{\boldsymbol{\theta}}\right)_i,$$

where $\partial_i \ell\left(\boldsymbol{f}_{\boldsymbol{\theta}}, \boldsymbol{f}^*\right)$ is the $i$-th element of $\nabla \ell\left(\boldsymbol{f}(\boldsymbol{x}, \boldsymbol{\theta}), \boldsymbol{f}^*(\boldsymbol{x})\right)$, and $\left(\boldsymbol{f}_{\boldsymbol{\theta}}\right)_i$ is the $i$-th element of the vectorization of $\boldsymbol{f}_{\boldsymbol{\theta}}$.

For the Hessian matrix $\boldsymbol{H}_S(\boldsymbol{\theta})$, we have

$$\boldsymbol{H}(\boldsymbol{\theta}) := \nabla_{\boldsymbol{\theta}} \nabla_{\boldsymbol{\theta}} R_S(\boldsymbol{\theta}) = \sum_{i=1}^{d^2} \mathbb{E}_S \nabla_{\boldsymbol{\theta}} \left(\partial_i \ell\left(\boldsymbol{f}_{\boldsymbol{\theta}}, \boldsymbol{f}^*\right)\right) \nabla_{\boldsymbol{\theta}} \left(\boldsymbol{f}_{\boldsymbol{\theta}}\right)_i + \sum_{i=1}^{d^2} \mathbb{E}_S \partial_i \ell\left(\boldsymbol{f}_{\boldsymbol{\theta}}, \boldsymbol{f}^*\right) \nabla_{\boldsymbol{\theta}} \nabla_{\boldsymbol{\theta}} \left((\boldsymbol{f}_{\boldsymbol{\theta}})_i\right)$$

$$= \sum_{i,j=1}^{d^2} \mathbb{E}_S \partial_{ij} \ell\left(\boldsymbol{f}_{\boldsymbol{\theta}}, \boldsymbol{f}^*\right) \nabla_{\boldsymbol{\theta}} \left(\boldsymbol{f}_{\boldsymbol{\theta}}\right)_i \left(\nabla_{\boldsymbol{\theta}} \left(\boldsymbol{f}_{\boldsymbol{\theta}}\right)_j\right)^\top + \sum_{i=1}^{d^2} \mathbb{E}_S \partial_i \ell\left(\boldsymbol{f}_{\boldsymbol{\theta}}, \boldsymbol{f}^*\right) \nabla_{\boldsymbol{\theta}} \nabla_{\boldsymbol{\theta}} \left((\boldsymbol{f}_{\boldsymbol{\theta}})_i\right),$$

$$= \sum_{i,j=1}^{d^2} \nabla_{\boldsymbol{\theta}} \left(\boldsymbol{f}_{\boldsymbol{\theta}}\right)_i \left(\nabla_{\boldsymbol{\theta}} \left(\boldsymbol{f}_{\boldsymbol{\theta}}\right)_j\right)^\top + \sum_{i=1}^{d^2} \mathbb{E}_S (\boldsymbol{f}_{\boldsymbol{\theta}} - \boldsymbol{f}^*)_i \nabla_{\boldsymbol{\theta}} \nabla_{\boldsymbol{\theta}} \left((\boldsymbol{f}_{\boldsymbol{\theta}})_i\right),$$

where $\partial_{ij} \ell\left(\boldsymbol{f}_{\boldsymbol{\theta}}, \boldsymbol{f}^*\right)$ is the $(i, j)$-th element of $\nabla \nabla \ell\left(\boldsymbol{f}(\boldsymbol{x}, \boldsymbol{\theta}), \boldsymbol{f}^*(\boldsymbol{x})\right)$.

We define matrices $\boldsymbol{H}^{(1)}(\boldsymbol{\theta})$ and $\boldsymbol{H}^{(2)}(\boldsymbol{\theta})$ as follows:

$$\boldsymbol{H}^{(1)}(\boldsymbol{\theta}) := \sum_{i,j=1}^{d^2} \nabla_{\boldsymbol{\theta}} \left(\boldsymbol{f}_{\boldsymbol{\theta}}\right)_i \left(\nabla_{\boldsymbol{\theta}} \left(\boldsymbol{f}_{\boldsymbol{\theta}}\right)_j\right)^\top,$$

$$\boldsymbol{H}^{(2)}(\boldsymbol{\theta}) := \sum_{i=1}^{d^2} \mathbb{E}_S (\boldsymbol{f}_{\boldsymbol{\theta}} - \boldsymbol{f}^*)_i \nabla_{\boldsymbol{\theta}} \nabla_{\boldsymbol{\theta}} \left((\boldsymbol{f}_{\boldsymbol{\theta}})_i\right),$$

We further denote that $\boldsymbol{H}(\boldsymbol{\theta}) := \boldsymbol{H}^{(1)}(\boldsymbol{\theta}) + \boldsymbol{H}^{(2)}(\boldsymbol{\theta})$.

For matrix factorization model, the eigenvectors of $\boldsymbol{H}^{(2)}$ has a special structure, as characterized by Lem. A.2.

**Lemma A.2 (Data-Independent Interleaved Structure of Eigenvectors of $\boldsymbol{H}^{(2)}$).** *Let $\boldsymbol{\theta}_c = (\boldsymbol{A}_c, \boldsymbol{B}_c)$ be any critical point of the matrix factorization model. If $\lambda$ is an eigenvalue of $\boldsymbol{H}^{(2)}(\boldsymbol{\theta}_c) \in \mathbb{R}^{2d^2 \times 2d^2}$, then there exist at least $d$ eigenvectors associated with $\lambda$. These $d$ eigenvectors take the form $\boldsymbol{v} \otimes \boldsymbol{e}_1, \boldsymbol{v} \otimes \boldsymbol{e}_2, \cdots, \boldsymbol{v} \otimes \boldsymbol{e}_d \in \mathbb{R}^{2d^2}$, where $\boldsymbol{v} \in \mathbb{R}^{2d}$ is a vector to be determined and $\boldsymbol{e}_i$ is the unit vector representing the $i$-th column of the identity matrix $\boldsymbol{I}_d \in \mathbb{R}^{d \times d}$.*

*Proof.* Let's denote the residual matrix at the critical point as $\delta \boldsymbol{M} = (\boldsymbol{A}_c \boldsymbol{B}_c - \boldsymbol{M})_{S_{\boldsymbol{x}}}$, where $(\boldsymbol{A}_c, \boldsymbol{B}_c)$ is a critical point. For the vectorized parameter $\boldsymbol{\theta}_c$, by direct calculation the matrix $\boldsymbol{H}^{(2)} := -\nabla^2 R_S(\boldsymbol{\theta}_c)$ can be formulated as a block matrix, with the diagonal blocks being $0$. The specific format is as follows:

$$\boldsymbol{H}^{(2)} = \begin{bmatrix} \boldsymbol{0} & -\delta \boldsymbol{M} \otimes \boldsymbol{I}_d \\ -\delta \boldsymbol{M}^\top \otimes \boldsymbol{I}_d & \boldsymbol{0} \end{bmatrix}. \tag{26}$$

Next, we compute the eigenvectors of $\boldsymbol{H}^{(2)}$. Let $\lambda$ be an eigenvalue of $\boldsymbol{H}^{(2)}$. We need to verify that $\boldsymbol{v} \otimes \boldsymbol{e}_1, \boldsymbol{v} \otimes \boldsymbol{e}_2, \cdots, \boldsymbol{v} \otimes \boldsymbol{e}_d \in \mathbb{R}^{2d^2}$ are the eigenvectors of $\boldsymbol{H}^{(2)}$ corresponding to $\lambda$, for a particular

$\boldsymbol{v} \in \mathbb{R}^{2d}$ yet to be determined. That is, we need to ensure that for all $1 \leq i \leq d$, the equation $(\boldsymbol{H}^{(2)} - \lambda \boldsymbol{I}_{2d^2})(\boldsymbol{v} \otimes \boldsymbol{e}_i) = \boldsymbol{0}$ has a non-zero solution for $\boldsymbol{v}$. Notice that

$$
\begin{aligned}
(\boldsymbol{H}^{(2)} - \lambda \boldsymbol{I}_{2d^2})(\boldsymbol{v} \otimes \boldsymbol{e}_i) &= \begin{bmatrix} -\lambda \boldsymbol{I}_d \otimes \boldsymbol{I}_d & -\boldsymbol{\delta M} \otimes \boldsymbol{I}_d \\ -\boldsymbol{\delta M}^\top \otimes \boldsymbol{I}_d & -\lambda \boldsymbol{I}_d \otimes \boldsymbol{I}_d \end{bmatrix} (\boldsymbol{v} \otimes \boldsymbol{e}_i) \\
&= \left( \begin{bmatrix} -\lambda \boldsymbol{I}_d & -\boldsymbol{\delta M} \\ -\boldsymbol{\delta M}^\top & -\lambda \boldsymbol{I}_d \end{bmatrix} \otimes \boldsymbol{I}_d \right)(\boldsymbol{v} \otimes \boldsymbol{e}_i) \quad (27) \\
&= \left( \begin{bmatrix} -\lambda \boldsymbol{I}_d & -\boldsymbol{\delta M} \\ -\boldsymbol{\delta M}^\top & -\lambda \boldsymbol{I}_d \end{bmatrix} \boldsymbol{v} \right) \otimes \boldsymbol{e}_i.
\end{aligned}
$$

Since $\lambda$ is an eigenvalue of $\boldsymbol{H}^{(2)}$, the determinant of the matrix $\boldsymbol{H}^{(2)} - \lambda \boldsymbol{I}_{2d^2}$ equals zero. Hence

$$
\det\left( \begin{bmatrix} -\lambda \boldsymbol{I}_d & -\boldsymbol{\delta M} \\ -\boldsymbol{\delta M}^\top & -\lambda \boldsymbol{I}_d \end{bmatrix} \right)^{2d} = \det\left( \begin{bmatrix} -\lambda \boldsymbol{I}_d & -\boldsymbol{\delta M} \\ -\boldsymbol{\delta M}^\top & -\lambda \boldsymbol{I}_d \end{bmatrix} \otimes \boldsymbol{I}_d \right) = 0. \quad (28)
$$

Consequently, from Eq. (27), we conclude that there always exists a non-zero vector $\boldsymbol{v} \in \mathbb{R}^{2d}$ such that $(\boldsymbol{H}^{(2)} - \lambda \boldsymbol{I}_{2d^2})(\boldsymbol{v} \otimes \boldsymbol{e}_i) = 0$. Since $\boldsymbol{v} \neq \boldsymbol{0}$, it is evident that $\boldsymbol{v} \otimes \boldsymbol{e}_1, \boldsymbol{v} \otimes \boldsymbol{e}_2, \cdots, \boldsymbol{v} \otimes \boldsymbol{e}_d \in \mathbb{R}^{2d^2}$ are linearly independent, and thus they represent $d$ eigenvectors corresponding to $\lambda$. $\square$

**Proposition A.7** (**Eigenvectors Structure of $H$ at the Origin**). *Consider the dynamics given by Eq. (21), where we denote $\boldsymbol{H} := -\nabla^2 R_S(\boldsymbol{0})$. If $\lambda$ is an eigenvalue of $\boldsymbol{H} \in \mathbb{R}^{2d^2 \times 2d^2}$, then there exist at least $d$ eigenvectors associated with $\lambda$ in $\boldsymbol{H}$. These $d$ eigenvectors take the form $\boldsymbol{v} \otimes \boldsymbol{e}_1, \boldsymbol{v} \otimes \boldsymbol{e}_2, \cdots, \boldsymbol{v} \otimes \boldsymbol{e}_d \in \mathbb{R}^{2d^2}$, where $\boldsymbol{v} \in \mathbb{R}^{2d}$ is a vector to be determined and $\boldsymbol{e}_i$ is the unit vector representing the $i$-th column of the identity matrix $\boldsymbol{I}_d \in \mathbb{R}^{d \times d}$.*

*Proof.* In the matrix factorization model, at the origin the gradient $\nabla_{\boldsymbol{\theta}}(\boldsymbol{f_\theta}) = \boldsymbol{0}$ and thus $\boldsymbol{H}^{(1)}(\boldsymbol{0}) = 0$ and the Hessian matrix reduces to $\boldsymbol{H}^{(2)}(\boldsymbol{0})$, making $\boldsymbol{H} = -\nabla^2 R_S(\boldsymbol{0}) = -\boldsymbol{H}^{(2)}(\boldsymbol{0})$.

Let's denote the residual matrix at the origin as $\boldsymbol{\delta M} = (\boldsymbol{A}_c \boldsymbol{B}_c - \boldsymbol{M})_{S_x}$, where at the origin $(\boldsymbol{A}_c, \boldsymbol{B}_c) = (\boldsymbol{0}, \boldsymbol{0})$. For the vectorized parameter $\boldsymbol{\theta}_c$, the matrix $\boldsymbol{H} := -\nabla^2 R_S(\boldsymbol{0})$ can be formulated as a block matrix, with the diagonal blocks being 0. The specific format is as follows:

$$
\boldsymbol{H} = \boldsymbol{H}^{(2)} = \begin{bmatrix} \boldsymbol{0} & -\boldsymbol{\delta M} \otimes \boldsymbol{I}_d \\ -\boldsymbol{\delta M}^\top \otimes \boldsymbol{I}_d & \boldsymbol{0} \end{bmatrix}. \quad (29)
$$

By Lem. A.2, the proof is completed. $\square$

**Lemma A.3** (**Eigenvectors Structure of $H$ at Second-order Stationary Point**). *Let $\Omega$ denote an $\Omega_k$ invariant manifold or sub-$\Omega_k$ invariant manifold defined in Prop. A.1 and A.2, and consider a second-order stationary point $\boldsymbol{\theta}_c$ within $\Omega$, i.e., $\nabla R_S(\boldsymbol{\theta}_c) = 0$ and $\boldsymbol{\theta}^\top \nabla^2 R_S(\boldsymbol{\theta}_c) \boldsymbol{\theta} \geq 0$ for all $\boldsymbol{\theta} \in \Omega$. Then, the eigenvectors corresponding to the negative eigenvalues of the Hessian matrix $\boldsymbol{H}(\boldsymbol{\theta}_c)$ are contained within the span of the eigenvectors corresponding to the negative eigenvalues of $\boldsymbol{H}^{(2)}(\boldsymbol{\theta}_c)$.*

*Proof.* Recall the definitions of $\boldsymbol{H}^{(1)}(\boldsymbol{\theta})$ and $\boldsymbol{H}^{(2)}(\boldsymbol{\theta})$ given by:

$$
\boldsymbol{H}^{(1)}(\boldsymbol{\theta}) := \sum_{i,j=1}^{d^2} \nabla_{\boldsymbol{\theta}}(\boldsymbol{f_\theta})_i \left( \nabla_{\boldsymbol{\theta}}(\boldsymbol{f_\theta})_j \right)^\top,
$$

$$
\boldsymbol{H}^{(2)}(\boldsymbol{\theta}) := \sum_{i=1}^{d^2} \mathbb{E}_S \left[ (\boldsymbol{f_\theta} - \boldsymbol{f}^*)_i \nabla_{\boldsymbol{\theta}}^2 (\boldsymbol{f_\theta})_i \right].
$$

The Hessian matrix $\boldsymbol{H}(\boldsymbol{\theta})$ at $\boldsymbol{\theta}$ is $\boldsymbol{H}(\boldsymbol{\theta}) := \boldsymbol{H}^{(1)}(\boldsymbol{\theta}) + \boldsymbol{H}^{(2)}(\boldsymbol{\theta})$.

The manifold $\Omega$ is an affine subspace with orthogonal complement denoted by $\Omega^\perp$. Let $\boldsymbol{H}$ have the following block representation in the bases of $\Omega$ and $\Omega^\perp$:

$$
\boldsymbol{H} = \begin{bmatrix} \boldsymbol{H}_{11} & \boldsymbol{H}_{12} \\ \boldsymbol{H}_{21} & \boldsymbol{H}_{22} \end{bmatrix}. \quad (30)
$$

Since $\boldsymbol{\Omega}$ is an invariant subspace under the gradient flow, we have $\boldsymbol{H}\boldsymbol{\theta} \in \boldsymbol{\Omega}$ for all $\boldsymbol{\theta} \in \boldsymbol{\Omega}$, which implies that $\boldsymbol{H}_{12} = \boldsymbol{0}$. Since $\boldsymbol{H}$ is symmetry, we have $\boldsymbol{H}_{21} = \boldsymbol{0}$.

Let $\lambda < 0$ be a negative eigenvalue of $\boldsymbol{H} := \boldsymbol{H}(\boldsymbol{\theta}_c)$ with $\boldsymbol{v}$ as the corresponding eigenvector. Since $\boldsymbol{H}_{11}$ is positive semi-definite, $\boldsymbol{v}$ must lie in $\boldsymbol{\Omega}^\perp$.

At a critical point $\boldsymbol{\theta}_c = (\boldsymbol{A}_c, \boldsymbol{B}_c)$, by direct calculation, the gradient $\nabla_{\boldsymbol{\theta}} \boldsymbol{f}_{\boldsymbol{\theta}_c}$ can be structured as:

$$\nabla_{\boldsymbol{\theta}} \boldsymbol{f}_{\boldsymbol{\theta}_c} = \begin{bmatrix} \boldsymbol{B}_c & & & \\ & \boldsymbol{B}_c & & \\ & & \ddots & \\ a_{11}\boldsymbol{I} & a_{21}\boldsymbol{I} & \cdots & a_{d1}\boldsymbol{I} \\ a_{12}\boldsymbol{I} & a_{22}\boldsymbol{I} & \cdots & a_{d2}\boldsymbol{I} \\ \vdots & \vdots & \ddots & \vdots \\ a_{1d}\boldsymbol{I} & a_{2d}\boldsymbol{I} & \cdots & a_{dd}\boldsymbol{I} \end{bmatrix} = \begin{bmatrix} \boldsymbol{I} \otimes \boldsymbol{B}_c \\ \boldsymbol{A}_c^\top \otimes \boldsymbol{I} \end{bmatrix}_{2d^2 \times d^2}, \tag{31}$$

where $\otimes$ denotes the Kronecker product.

Note that $\nabla_{\boldsymbol{\theta}} (\boldsymbol{f}_{\boldsymbol{\theta}^*})_j$ is the $j$-th column of matrix $\nabla_{\boldsymbol{\theta}} \boldsymbol{f}_{\boldsymbol{\theta}_c}$ and it falls precisely within the defined $\boldsymbol{\Omega}_k$ invariant manifold or sub-$\boldsymbol{\Omega}_k$ invariant manifold $\boldsymbol{\Omega}$. Therefore, we have:

$$\left( \nabla_{\boldsymbol{\theta}} (\boldsymbol{f}_{\boldsymbol{\theta}_c})_j \right)^\top \boldsymbol{v} = 0 \quad \forall 1 \le j \le d^2, \tag{32}$$

which implies that $\boldsymbol{v}$ is orthogonal to the image of $\nabla_{\boldsymbol{\theta}} (\boldsymbol{f}_{\boldsymbol{\theta}_c})$, placing it in the null space of $\boldsymbol{H}^{(1)}(\boldsymbol{\theta}_c)$.

As a result, we have:

$$\boldsymbol{H}(\boldsymbol{\theta}_c)\boldsymbol{v} = \left( \boldsymbol{H}^{(1)}(\boldsymbol{\theta}_c) + \boldsymbol{H}^{(2)}(\boldsymbol{\theta}_c) \right) \boldsymbol{v} = \boldsymbol{H}^{(2)}(\boldsymbol{\theta}_c)\boldsymbol{v} = \lambda\boldsymbol{v}.$$

Thus, the eigenvector $\boldsymbol{v}$ of the Hessian $\boldsymbol{H}(\boldsymbol{\theta}_c)$, corresponding to the negative eigenvalue $\lambda$, is also an eigenvector of $\boldsymbol{H}^{(2)}(\boldsymbol{\theta}_c)$, confirming that $\boldsymbol{v}$ is within the span of the eigenvectors of $\boldsymbol{H}^{(2)}(\boldsymbol{\theta}_c)$. $\qquad \square$

### A.6 Transition to the Next Rank-level Invariant Manifold

**Proposition A.8.** *The linear combination of the eigenvectors of $\boldsymbol{H}^{(2)}$: $c_1(\boldsymbol{v} \otimes \boldsymbol{e}_1) + c_2(\boldsymbol{v} \otimes \boldsymbol{e}_2) + \cdots + c_d(\boldsymbol{v} \otimes \boldsymbol{e}_d)$ falls within the invariant manifold $\boldsymbol{\Omega}_1(\boldsymbol{c})$, where $\boldsymbol{c} = (c_1, c_2, \cdots, c_d)^\top$.*

*Proof.* Notice that

$$\begin{aligned} & c_1(\boldsymbol{v} \otimes \boldsymbol{e}_1) + c_2(\boldsymbol{v} \otimes \boldsymbol{e}_2) + \cdots + c_d(\boldsymbol{v} \otimes \boldsymbol{e}_d) \\ & = \boldsymbol{v} \otimes [c_1\boldsymbol{e}_1 + c_2\boldsymbol{e}_2 + \cdots + c_d\boldsymbol{e}_d] \\ & = \boldsymbol{v} \otimes \boldsymbol{c}. \end{aligned} \tag{33}$$

By Definition A.1, the data-independent invariant manifold generated by $\boldsymbol{c}$ is $\boldsymbol{\Omega}_1(\boldsymbol{c}) = \{(\boldsymbol{A}, \boldsymbol{B}) | \boldsymbol{a}_i, \boldsymbol{b}_j \in \text{span}\{\boldsymbol{c}\}, \forall 1 \le i, j \le d\}$. If $\boldsymbol{\theta} = (\boldsymbol{A}, \boldsymbol{B}) \in \boldsymbol{\Omega}_1(\boldsymbol{c})$, then $\boldsymbol{A}, \boldsymbol{B}$ must take the form

$$\boldsymbol{A} = \begin{bmatrix} \beta_1\boldsymbol{c} \\ \beta_2\boldsymbol{c} \\ \vdots \\ \beta_d\boldsymbol{c} \end{bmatrix}, \quad \boldsymbol{B} = \begin{bmatrix} \beta_{d+1}\boldsymbol{c} \\ \beta_{d+2}\boldsymbol{c} \\ \vdots \\ \beta_{2d}\boldsymbol{c} \end{bmatrix}, \tag{34}$$

for some $\boldsymbol{\beta} = [\beta_1, \beta_2, \cdots, \beta_{2d}]^\top \in \mathbb{R}^{2d}$, and the vectorized parameter $\boldsymbol{\theta} = \text{vec}((\boldsymbol{A}, \boldsymbol{B})) \in \mathbb{R}^{2d}$ takes the form $\boldsymbol{\beta} \otimes \boldsymbol{c}$. Let $\boldsymbol{\beta} = \boldsymbol{v}$, and the proof is complete. $\qquad \square$

**Lemma A.4.** *Suppose $\boldsymbol{\alpha}_1, \boldsymbol{\alpha}_2, \cdots, \boldsymbol{\alpha}_{k+1} \in \mathbb{R}^d$ are linearly independent, the data-independent invariant manifold exhibits the property $\boldsymbol{\Omega}_k(\boldsymbol{\alpha}_1, \boldsymbol{\alpha}_2, \cdots, \boldsymbol{\alpha}_k) + \boldsymbol{\Omega}_1(\boldsymbol{\alpha}_{k+1}) = \boldsymbol{\Omega}_{k+1}(\boldsymbol{\alpha}_1, \boldsymbol{\alpha}_2, \cdots, \boldsymbol{\alpha}_{k+1})$.*

*Proof.* Assume that $\boldsymbol{\theta} = (\boldsymbol{A}, \boldsymbol{B}) \in \boldsymbol{\Omega}_{k+1}(\boldsymbol{\alpha}_1, \boldsymbol{\alpha}_2, \cdots, \boldsymbol{\alpha}_{k+1})$. Then $\boldsymbol{A}, \boldsymbol{B}$ should adopt the form:

$$A = \sum_{i=1}^{k} \beta_i \boldsymbol{\alpha}_i^\top + \beta_{k+1}\boldsymbol{\alpha}_{k+1}^\top, B = \sum_{i=1}^{k} \gamma_i \boldsymbol{\alpha}_i^\top + \gamma_{k+1}\boldsymbol{\alpha}_{k+1}^\top. \tag{35}$$

Denote $\boldsymbol{c}_i = \begin{bmatrix} \boldsymbol{\beta}_i \\ \boldsymbol{\gamma}_i \end{bmatrix} \in \mathbb{R}^{2d}$, then the vectorized parameters $\boldsymbol{\theta} := \text{vec}(\boldsymbol{\theta})$ can be expressed as:

$$\boldsymbol{\theta} = \sum_{i=1}^{k} \boldsymbol{c}_i \otimes \boldsymbol{\alpha}_i + \boldsymbol{c}_{k+1} \otimes \boldsymbol{\alpha}_{k+1}. \tag{36}$$

Since we know that $\sum_{i=1}^{k} \boldsymbol{c}_i \otimes \boldsymbol{\alpha}_i \in \boldsymbol{\Omega}_k(\boldsymbol{\alpha}_1, \boldsymbol{\alpha}_2, \cdots, \boldsymbol{\alpha}_k)$ and $\boldsymbol{c}_{k+1} \otimes \boldsymbol{\alpha}_{k+1} \in \boldsymbol{\Omega}_1(\boldsymbol{\alpha})$, it is straightforward to validate that $\boldsymbol{\Omega}_{k+1}(\boldsymbol{\alpha}_1, \boldsymbol{\alpha}_2, \cdots, \boldsymbol{\alpha}_{k+1}) = \boldsymbol{\Omega}_k(\boldsymbol{\alpha}_1, \boldsymbol{\alpha}_2, \cdots, \boldsymbol{\alpha}_k) + \boldsymbol{\Omega}_1(\boldsymbol{\alpha}_{k+1})$. $\qquad \square$

**Assumption A.1 (Unique Top Eigenvalue).** *Let $\delta \boldsymbol{M} = (\boldsymbol{A}_c \boldsymbol{B}_c - \boldsymbol{M})_{S_x}$ be the residual matrix at the critical point $\boldsymbol{\theta}_c = (\boldsymbol{A}_c, \boldsymbol{B}_c)$. Assume that the top eigenvalue of the matrix $\begin{bmatrix} \boldsymbol{0} & -\delta \boldsymbol{M} \\ -\delta \boldsymbol{M}^\top & \boldsymbol{0} \end{bmatrix}$ is unique.*

**Assumption A.2 (Second-order Stationary Point).** *Let $\boldsymbol{\Omega}$ be an $\boldsymbol{\Omega}_k$ invariant manifold or sub-$\boldsymbol{\Omega}_k$ invariant manifold defined in Prop. A.1 or A.2. Assume $\boldsymbol{\theta}_c$ is a second-order stationary point within $\boldsymbol{\Omega}$, i.e., $\nabla R_S(\boldsymbol{\theta}_c) = 0$ and $\boldsymbol{\theta}^\top \nabla^2 R_S(\boldsymbol{\theta}_c)\boldsymbol{\theta} \geq 0$ for all $\boldsymbol{\theta} \in \boldsymbol{\Omega}$.*

**Theorem A.4 (Transition to the Next Rank-level Invariant Manifold).** *Consider the dynamics $\dot{\boldsymbol{\theta}} = -\nabla R_S(\boldsymbol{\theta})$. Let $\varphi(\boldsymbol{\theta}_0, t)$ denote the value of $\boldsymbol{\theta}(t)$ when $\boldsymbol{\theta}(0) = \boldsymbol{\theta}_0$. Let $\boldsymbol{\Omega}$ be a $\boldsymbol{\Omega}_k$ invariant manifold or sub-$\boldsymbol{\Omega}_k$ invariant manifold. Let $\boldsymbol{\theta}_c \in \boldsymbol{\Omega}$ be a critical point satisfying Assump. A.1 and A.2. Then, for randomly selected $\boldsymbol{\theta}_0$, with probability 1 with respect to $\boldsymbol{\theta}_0$, the limit*

$$\tilde{\varphi}(\boldsymbol{\theta}_c, t) := \lim_{\alpha \to 0} \varphi\left(\boldsymbol{\theta}_c + \alpha\boldsymbol{\theta}_0, t + \frac{1}{\lambda_1} \log \frac{1}{\alpha}\right) \tag{37}$$

*exists and falls into an invariant manifold $\boldsymbol{\Omega}_{k+1}$. Here $\lambda_1$ is the top eigenvalue of negative Hessian $-\nabla^2 R_S(\boldsymbol{\theta}_c)$.*

*Proof.* At the critical point $\boldsymbol{\theta}_c$, we denote the negative Hessian as $\boldsymbol{H} := -\nabla^2 R_S(\boldsymbol{\theta}_c)$. Let $\lambda_1 \in \mathbb{R}$ be the largest eigenvalue of $\boldsymbol{H}$, with corresponding eigenvectors $\boldsymbol{q}_{11}, \boldsymbol{q}_{12}, \cdots, \boldsymbol{q}_{1l_1}$.

Denote $c_j = \langle \boldsymbol{\theta}_0 - \boldsymbol{\theta}_c, \boldsymbol{q}_{1j} \rangle, \forall 1 \leq j \leq l_1$, and $\boldsymbol{v}_1 = \sum_{j=1}^{l_1} c_j \boldsymbol{q}_{1j}$. For a randomly selected $\boldsymbol{\theta}_0$, with probability 1, there exists at least one $j$ such that $c_j \neq 0$.

Consider the path $\boldsymbol{z}_\alpha(t) := \varphi\left(\boldsymbol{\theta}_c + \alpha\boldsymbol{v}_1, t + \frac{1}{\lambda_1} \log \frac{1}{\alpha}\right)$ for every $\alpha > 0$. By Prop. A.6, the limit $\boldsymbol{z}(t) := \lim_{\alpha \to 0} \boldsymbol{z}_\alpha(t)$ exists and satisfies the dynamics $\boldsymbol{z}(t) = \varphi(\boldsymbol{z}(0), t)$.

Furthermore, $\forall t \in \mathbb{R}$, there exists a constant $C > 0$ such that

$$\left\| \varphi\left(\boldsymbol{\theta}_c + \alpha\boldsymbol{\theta}_0, t + \frac{1}{\lambda_1} \log \frac{1}{\alpha}\right) - \boldsymbol{z}(t) \right\|_2 \leq C\alpha^{\frac{\lambda_1 - \lambda_2}{2\lambda_1 - \lambda_2}}$$

for every sufficiently small $\alpha$, where $\lambda_1 - \lambda_2 > 0$ is the eigenvalue gap.

This implies that the limit $\lim_{\alpha \to 0} \varphi\left(\boldsymbol{\theta}_c + \alpha\boldsymbol{\theta}_0, t + \frac{1}{\lambda_1} \log \frac{1}{\alpha}\right)$ exists and

$$\tilde{\varphi}(\boldsymbol{\theta}_c, t) := \lim_{\alpha \to 0} \varphi\left(\boldsymbol{\theta}_c + \alpha\boldsymbol{\theta}_0, t + \frac{1}{\lambda_1} \log \frac{1}{\alpha}\right) = \lim_{\alpha \to 0} \varphi\left(\boldsymbol{\theta}_c + \alpha\boldsymbol{v}_1, t + \frac{1}{\lambda_1} \log \frac{1}{\alpha}\right).$$

Assuming $\boldsymbol{\theta}_c \in \boldsymbol{\Omega}_k$ and satisfies Assumps. A.1 and A.2, we aim to show the existence of a rank-$(k+1)$ invariant manifold $\boldsymbol{\Omega}_{k+1}$ containing $\boldsymbol{\theta}_c + \alpha\boldsymbol{v}_1$.

Define the following matrices:

$$\boldsymbol{H}^{(1)}(\boldsymbol{\theta}_c) := \sum_{i,j=1}^{d^2} \nabla_{\boldsymbol{\theta}_c} \left(\boldsymbol{f}_{\boldsymbol{\theta}_c}\right)_i \left(\nabla_{\boldsymbol{\theta}_c} \left(\boldsymbol{f}_{\boldsymbol{\theta}_c}\right)_j\right)^\top,$$

$$\boldsymbol{H}^{(2)}(\boldsymbol{\theta}_c) := \sum_{i=1}^{d^2} \mathbb{E}_S \left[(\boldsymbol{f}_{\boldsymbol{\theta}_c} - \boldsymbol{f}^*)_i \nabla_{\boldsymbol{\theta}_c}^2 \left(\boldsymbol{f}_{\boldsymbol{\theta}_c}\right)_i\right].$$

The Hessian matrix $\boldsymbol{H}(\boldsymbol{\theta}_c)$ at $\boldsymbol{\theta}_c$ can be expressed as $\boldsymbol{H}(\boldsymbol{\theta}_c) := \boldsymbol{H}^{(1)}(\boldsymbol{\theta}_c) + \boldsymbol{H}^{(2)}(\boldsymbol{\theta}_c)$, and we have $\boldsymbol{H} = -\boldsymbol{H}(\boldsymbol{\theta}_c)$.

Assump. A.1 and Lem. A.2 imply that there exist exactly $d$ eigenvectors associated with the top eigenvalue $\lambda_1$ of $-\boldsymbol{H}^{(2)}(\boldsymbol{\theta}_c)$. These eigenvectors are of the form $\boldsymbol{v} \otimes \boldsymbol{e}_i$ for $i = 1, \ldots, d$, where $\boldsymbol{v} \in \mathbb{R}^{2d}$ is a vector to be determined and $\boldsymbol{e}_i$ is the $i$-th standard basis vector in $\mathbb{R}^d$. By Assump. A.2 and Lem. A.3, the eigenvectors corresponding to $\lambda_1$ of $\boldsymbol{H}$ are contained within the span of the eigenvectors associated with the negative eigenvalues of $\boldsymbol{H}^{(2)}(\boldsymbol{\theta}_c)$.

Prop. A.8 ensures that the escaping direction $\boldsymbol{v}_1$ lies within a rank-1 invariant manifold $\boldsymbol{\Omega}_1$. Lem. A.4 then guarantees the existence of an invariant manifold $\boldsymbol{\Omega}_{k+1}$ that includes $\boldsymbol{\theta}_c + \alpha \boldsymbol{v}_1$. Since $\boldsymbol{\Omega}_{k+1}$ is invariant under the gradient flow, the trajectory $\varphi\left(\boldsymbol{\theta}_c + \alpha \boldsymbol{v}_1, t + \frac{1}{\lambda_1} \log \frac{1}{\alpha}\right)$ remains within $\boldsymbol{\Omega}_{k+1}$.

Finally, since $\boldsymbol{\Omega}_{k+1}$ is a closed subspace, the limit $\tilde{\varphi}(\boldsymbol{\theta}_c, t)$ lies in $\bar{\boldsymbol{\Omega}}_{k+1}$, concluding the proof. $\qquad\square$

## A.7 Example of Coincident Top Eigenvalues

Consider the $2 \times 2$ matrix completion problem: $\boldsymbol{M} = \begin{bmatrix} 2 & \star \\ \star & 2 \end{bmatrix}$. In this case, the two numbers on the diagonal are identical, which causes the maximum singular value of the residual matrix at the origin to be non-unique, violating Assump. 1. Consequently, the training process will jump directly from the rank 0 to the rank 2 invariant manifold, thereby missing the lowest rank solution of rank 1. This behavior is demonstrated in Fig. A2, which shows experimental results for this scenario.

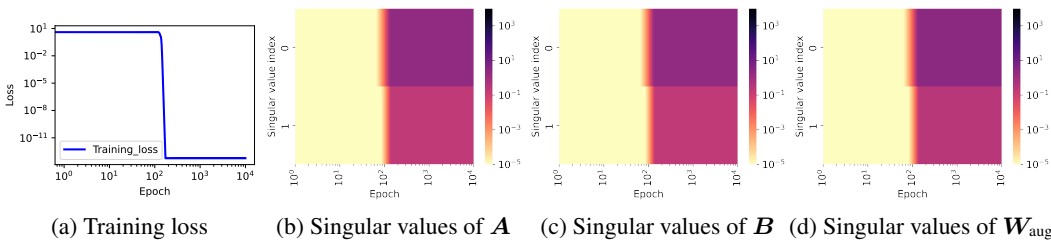

(a) Training loss      (b) Singular values of $\boldsymbol{A}$     (c) Singular values of $\boldsymbol{B}$    (d) Singular values of $\boldsymbol{W}_{\text{aug}}$

Figure A2: **Analysis of matrix completion for $\boldsymbol{M}$ with identical diagonal elements.** (a) Training loss under small initialization. (b-d) Singular value evolution for $\boldsymbol{A}, \boldsymbol{B}, \boldsymbol{W}_{\text{aug}}$. Simultaneous growth of singular values results in direct convergence to a rank-2 invariant manifold.

## A.8 Minimum Rank

**Theorem A.5 (Minimum Rank).** *Let $\boldsymbol{\Omega}$ denote an invariant as defined previously. Assume $\boldsymbol{W}_t$ achieves a global minimum within each invariant manifold $\boldsymbol{\Omega}_k$. If the limit $\widehat{\boldsymbol{W}} = \lim_{\alpha \to 0} \boldsymbol{W}_\infty(\alpha I)$ exists and is a global optimum with $\widehat{\boldsymbol{W}}(i, j) = \boldsymbol{M}(i, j)$ for all $(i, j) \in S_{\boldsymbol{x}}$, then*

$$\widehat{\boldsymbol{W}} \in \operatorname{argmin}_{\boldsymbol{W}} \operatorname{rank}(\boldsymbol{W}) \quad s.t. \quad \boldsymbol{W}(i, j) = \boldsymbol{M}(i, j), \forall (i, j) \in S_{\boldsymbol{x}}. \tag{38}$$

*Proof.* Consider the invariant manifold $\boldsymbol{\Omega}_k$, which is defined as follows:

$$\boldsymbol{\Omega}_k := \boldsymbol{\Omega}_k(\boldsymbol{\alpha}_1, \cdots, \boldsymbol{\alpha}_k) = \{(\boldsymbol{A}, \boldsymbol{B}) | \boldsymbol{a}_i, \boldsymbol{b}_{\cdot, j} \in \operatorname{span}\{\boldsymbol{\alpha}_1, \cdots, \boldsymbol{\alpha}_k\}, \forall 1 \le i, j \le d\},$$

where $\boldsymbol{a}_i$ denotes the $i$-th row of $\boldsymbol{A}$, $\boldsymbol{b}_{\cdot, j}$ denotes the $j$-th column of $\boldsymbol{B}$, and $\boldsymbol{\alpha}_1, \ldots, \boldsymbol{\alpha}_k$ are independent vectors that span the invariant subspace associated with $\boldsymbol{\Omega}_k$.

According to Thm. A.4, the training trajectory adheres to a hierarchical traversal across invariant manifolds. For any matrix $\boldsymbol{C}$ with $\operatorname{rank}(\boldsymbol{C}) \le k$, we will show that there always exists $\boldsymbol{\theta} = (\boldsymbol{A}, \boldsymbol{B}) \in \boldsymbol{\Omega}_k$ such that $\boldsymbol{A}\boldsymbol{B} = \boldsymbol{C}$. Therefore, $\boldsymbol{\Omega}_k$ contains all matrices of rank $k$.

In fact, Since $\operatorname{rank}(\boldsymbol{C}) \le k$, we can express $\boldsymbol{C}$ as a sum of $k$ rank-one matrices: $\boldsymbol{C} = \sum_{i=1}^{k} \boldsymbol{u}_i \boldsymbol{v}_i^\top$ where $\boldsymbol{u}_i$ and $\boldsymbol{v}_i$ are column vectors. By the definition of $\boldsymbol{\Omega}_k$, for any $\boldsymbol{\theta} = (\boldsymbol{A}, \boldsymbol{B}) \in \boldsymbol{\Omega}_k$, each

row of $A$ and each column of $B$ can be expressed as a linear combination of $\{\alpha_1, \cdots, \alpha_k\}$: $a_i = \sum_{j=1}^{k} c_{ij}\alpha_j$ $b_{\cdot,j} = \sum_{i=1}^{k} d_{ij}\alpha_i$ where $c_{ij}$ and $d_{ij}$ are scalars. We can write:

$$A = \begin{bmatrix} \sum_{j=1}^{k} c_{1j}\alpha_j \\ \vdots \\ \sum_{j=1}^{k} c_{dj}\alpha_j \end{bmatrix}, B = \begin{bmatrix} \sum_{i=1}^{k} d_{i1}\alpha_i & \cdots & \sum_{i=1}^{k} d_{id}\alpha_i \end{bmatrix}.$$

Now, we can express the product $AB$ as: $AB = \sum_{i=1}^{k}\sum_{j=1}^{k}\left(\sum_{l=1}^{d} c_{li}d_{jl}\right)\alpha_i\alpha_j^\top$ By choosing appropriate values for $c_{ij}$ and $d_{ij}$, we can make $AB = C$. This is possible because the outer products $\alpha_i\alpha_j^\top$ span the same subspace as the rank-one matrices $u_iv_i^\top$ in the expression of $C$.

Therefore, for any matrix $C$ with $\text{rank}(C) \leq k$, there always exists $\theta = (A, B) \in \Omega_k$ such that $AB = C$.

If the output matrix $W_t$ attains optimums within each $\Omega_k$, it suggests that the optimization process is selecting the best approximation from the set of all possible rank-$k$ matrices. Provided that each step in the optimization is optimal, the resulting solution will naturally be the matrix with the lowest feasible rank that satisfies the matrix completion criteria, thereby completing the proof. $\qquad\square$

### A.9 Minimum Nuclear Norm Guarantee

**Lemma A.5** (**Minimal Nuclear Norm Computation**). *Given a matrix $M$ to be completed with observed diagonal entries, i.e., $\text{diag}(M) = v$, the minimal nuclear norm solution among all possible completions is $\|v\|_1$.*

*Proof.* The nuclear norm of a matrix is the dual of the spectral norm $\|\cdot\|_2$, defined as:

$$\|A\|_* = \max_{\|X\|_2 \leq 1} \langle A, X\rangle.$$

Given that $\|\text{diag}(\text{sign}(v))\|_2 \leq 1$, for any matrix $A$ with $\text{diag}(A) = v$, it follows that:

$$\|A\|_* \geq \langle A, \text{diag}(\text{sign}(v))\rangle = \langle v, \text{sign}(v)\rangle = \|v\|_1.$$

Specifically, the nuclear norm of the diagonal matrix with $v$ on its diagonal is $\|\text{diag}(v)\|_* = \|v\|_1$, which establishes that the diagonal matrix with $v$ is indeed a minimizer for the nuclear norm. $\qquad\square$

**Diagonal Observations**

**Proposition A.9** (**Minimum Nuclear Norm Guarantee in Diagonal Case**). *Consider the dynamics $\dot{\theta} = -\nabla R_S(\theta)$, where $\theta(t) = (A(t), B(t))$ and denote $W_t = A(t)B(t)$. If the observation data is diagonal, and if for a full rank initialization $W_0$, the limit $\widehat{W} = \lim_{\alpha\to 0} W_\infty(\alpha W_0)$ exists and is a global optimum with $\widehat{W}_{ij} = M_{ij}$ for all $(i,j) \in S_x$, then*

$$\widehat{W} \in \text{argmin}_W \|W\|_* \quad s.t. \quad W_{ij} = M_{ij}, \forall(i,j) \in S_x. \tag{39}$$

*Proof.* Without loss of generality, assume that $M$ is a diagonal matrix given by:

$$M = \begin{bmatrix} \mu_1 & & & \\ & \mu_2 & & \\ & & \ddots & \\ & & & \mu_d \end{bmatrix}.$$

By Lem. A.5, the minimal nuclear norm among all possible completions is $|\mu_1| + |\mu_2| + \cdots + |\mu_d|$. When the matrix to be completed is diagonal, the evolution of the $i$-th row of $A$ is influenced only by the $i$-th column of $B$. Hence, the dynamics decouple into $d$ independent parts, each equivalent to

learning a scalar $\mu_i$. The learning process thus unfolds in $d$ stages, with each stage passing through a critical point to learn a respective $\mu_i$.

By Lem. A.2, the second term of the Hessian matrix can be expressed as:

$$\boldsymbol{H}^{(2)} = \begin{bmatrix} \boldsymbol{0} & -\boldsymbol{\delta M} \otimes \boldsymbol{I_d} \\ -\boldsymbol{\delta M}^{\top} \otimes \boldsymbol{I_d} & \boldsymbol{0} \end{bmatrix}.$$

According to Lem. A.2, the $d$ eigenvectors of $\boldsymbol{H}^{(2)}$ take the form $\boldsymbol{v} \otimes \boldsymbol{e}_1, \boldsymbol{v} \otimes \boldsymbol{e}_2, \ldots, \boldsymbol{v} \otimes \boldsymbol{e}_d \in \mathbb{R}^{2d^2}$, where $\boldsymbol{v} \in \mathbb{R}^{2d}$ is a vector to be determined and $\boldsymbol{e}_i$ is the unit vector corresponding to the $i$-th column of the identity matrix $\boldsymbol{I_d} \in \mathbb{R}^{d \times d}$.

(i) Suppose $\mu_1 > \mu_2 > \cdots > \mu_d > 0$.

With a infinitesimal initialization, the training dynamics first focus on the element with the largest singular value, then proceed sequentially to blocks with smaller singular values. This pattern of learning is consistent with the concept of "sequential learning" as reported in the literature (Gidel et al., 2019; Gissin et al., 2019; Jiang et al., 2023).

For a diagonal observation matrix, the residual matrix $\delta M$ at any critical point remains a diagonal matrix. While starting to learn $\mu_i$ from a critical point, direct calculation confirms that vector $\boldsymbol{v} = [\boldsymbol{e}_i, \boldsymbol{e}_i]^{\top} \in \mathbb{R}^{2d}$. Escaping from each saddle point $\boldsymbol{\theta}_c$, the trajectory $\boldsymbol{\theta}(t) - \boldsymbol{\theta}_c$ approximates $\sum_{i=1}^{d} c_i(\boldsymbol{v} \otimes \boldsymbol{e}_i)$, which satisfies $\boldsymbol{B}_{,i} = (\boldsymbol{A}^{\top})_{,i}$. Thus, learning a diagonal matrix $\boldsymbol{M}$ using the asymmetric model $\boldsymbol{AB}$ is equivalent to using a symmetric model $\boldsymbol{AA}^{\top}$. The final outcome ensures that $\mathrm{diag}(\boldsymbol{AB}) = \mathrm{diag}(\boldsymbol{AA}^{\top}) = \mathrm{diag}(\boldsymbol{M})$.

The nuclear norm of $\boldsymbol{AA}^{\top}$ equals the sum of its eigenvalues, which is precisely the trace of the matrix, and $\mathrm{tr}(\boldsymbol{AA}^{\top}) = \mathrm{tr}(\boldsymbol{M}) = \mu_1 + \mu_2 + \cdots + \mu_d$. Therefore, the nuclear norm of the learned matrix $\boldsymbol{W} = \boldsymbol{AB} = \boldsymbol{AA}^{\top}$ remains $|\mu_1| + |\mu_2| + \cdots + |\mu_d|$.

(ii) If some $\mu_i < 0$, assume without loss of generality that $|\mu_1| > |\mu_2| > \cdots > |\mu_n| > 0$.

While starting to learn $\mu_i$ from a critical point, direct calculation confirms that $\boldsymbol{v} = [\boldsymbol{e}_i, \mathrm{sign}(\mu_i)\boldsymbol{e}_i]^{\top} \in \mathbb{R}^{2d}$. Escaping from each saddle point $\boldsymbol{\theta}_c$, the trajectory $\boldsymbol{\theta}(t) - \boldsymbol{\theta}_c$ approximates $\sum_{i=1}^{d} c_i(\boldsymbol{v} \otimes \boldsymbol{e}_i)$, satisfying $\boldsymbol{B}_{,i} = \mathrm{sign}(\mu_i)(\boldsymbol{A}^{\top})_{,i}$. Hence, $\boldsymbol{AB} = \boldsymbol{AA}^{\top}\boldsymbol{Q}$, where $\boldsymbol{Q}$ is an orthogonal matrix given by:

$$\boldsymbol{Q} = \begin{bmatrix} \mathrm{sign}(\mu_1) & & & \\ & \mathrm{sign}(\mu_2) & & \\ & & \ddots & \\ & & & \mathrm{sign}(\mu_d) \end{bmatrix}.$$

The final result ensures that $\mathrm{diag}(\boldsymbol{AB}) = \mathrm{diag}(\boldsymbol{AA}^{\top}\boldsymbol{Q}) = \mathrm{diag}(\boldsymbol{M})$, meaning $\mathrm{diag}(\boldsymbol{AA}^{\top}) = \mathrm{diag}(\boldsymbol{QM})$. The nuclear norm of $\boldsymbol{AA}^{\top}$ equals the sum of its eigenvalues, which is the trace of the matrix, and $\mathrm{tr}(\boldsymbol{AA}^{\top}) = \mathrm{tr}(\boldsymbol{QM}) = |\mu_1| + |\mu_2| + \cdots + |\mu_d|$.

Since an orthogonal transformation does not change the nuclear norm of a matrix, the nuclear norm of the final learned matrix $\boldsymbol{W} = \boldsymbol{AB} = \boldsymbol{AA}^{\top}\boldsymbol{Q}$ is still $|\mu_1| + |\mu_2| + \cdots + |\mu_d|$. $\qquad\square$

**Disconnected with Complete Bipartite Components**

**Theorem A.6 (Minimum Nuclear Norm Guarantee).** *Consider the dynamics $\dot{\boldsymbol{\theta}} = -\nabla R_S(\boldsymbol{\theta})$, where $\boldsymbol{\theta}(t) = (\boldsymbol{A}(t), \boldsymbol{B}(t))$, and let $\boldsymbol{W}_t = \boldsymbol{A}(t)\boldsymbol{B}(t)$. If the observation graph associated with the matrix $\boldsymbol{M}$ to be completed is disconnected with complete bipartite components, and if for a full rank initialization $\boldsymbol{W}_0$, the limit $\widehat{\boldsymbol{W}} = \lim_{\alpha \to 0} \boldsymbol{W}_{\infty}(\alpha \boldsymbol{W}_0)$ exists and is a global optimum with $\widehat{\boldsymbol{W}}_{ij} = \boldsymbol{M}_{ij}$ for all $(i, j) \in S_{\boldsymbol{x}}$, then*

$$\widehat{\boldsymbol{W}} \in \mathrm{argmin}_{\boldsymbol{W}} \|\boldsymbol{W}\|_* \quad s.t. \quad \boldsymbol{W}_{ij} = \boldsymbol{M}_{ij}, \forall (i, j) \in S_{\boldsymbol{x}}. \tag{40}$$

*Proof.* Consider a matrix $\boldsymbol{M} \in \mathbb{R}^{d \times d}$ composed of $m$ connected components, with each component forming a complete bipartite subgraph. Since $\boldsymbol{M}$ is disconnected, it can be represented in a block

diagonal form without loss of generality:

$$
\boldsymbol{M} = \begin{bmatrix} \boldsymbol{M}_1 & & & \\ & \boldsymbol{M}_2 & & \\ & & \ddots & \\ & & & \boldsymbol{M}_m \end{bmatrix},
$$

where each block $\boldsymbol{M}_i \in \mathbb{R}^{d_i \times d_i'}$, and $\sum_{i=1}^m d_i = d, \sum_{i=1}^m d_i' = d$, representing the sum of the dimensions of the blocks.

Each block $\boldsymbol{M}_i$ corresponds some singular values of the corresponding Hessian matrix at a critical point. With a infinitesimal initialization, the training dynamics first focus on the block with the largest singular value, then proceed sequentially to blocks with smaller singular values. This pattern of learning is consistent with the concept of "sequential learning" as reported in the literature (Gidel et al., 2019; Gissin et al., 2019; Jiang et al., 2023).

Since each connected component of $\boldsymbol{M}$ forms a complete bipartite subgraph, the block $\boldsymbol{M}_i$ is fully observed. We can do singular value decomposition (SVD) on each sub-block $\boldsymbol{M}_i$ as $\boldsymbol{M}_i = \boldsymbol{U}_i \Sigma_i \boldsymbol{V}_i^\top$, where $\boldsymbol{U}_i$ and $\boldsymbol{V}_i$ are orthogonal matrices, and $\Sigma_i$ is a diagonal matrix with the singular values of $\boldsymbol{M}_i$.

Construct block diagonal matrices $\boldsymbol{U}$ and $\boldsymbol{V}$ as follows:

$$
\boldsymbol{U} = \begin{bmatrix} \boldsymbol{U}_1 & & & \\ & \boldsymbol{U}_2 & & \\ & & \ddots & \\ & & & \boldsymbol{U}_m \end{bmatrix}, \quad \boldsymbol{V} = \begin{bmatrix} \boldsymbol{V}_1 & & & \\ & \boldsymbol{V}_2 & & \\ & & \ddots & \\ & & & \boldsymbol{V}_m \end{bmatrix}.
$$

This leads to the diagonal matrix:

$$
\boldsymbol{U}\boldsymbol{M}\boldsymbol{V}^\top = \begin{bmatrix} \Sigma_1 & & & \\ & \Sigma_2 & & \\ & & \ddots & \\ & & & \Sigma_m \end{bmatrix} = \begin{bmatrix} \mu_1 & & & \\ & \mu_2 & & \\ & & \ddots & \\ & & & \mu_d \end{bmatrix}.
$$

Orthogonal transformations preserve the nuclear norm, so by Lem. A.5, the minimal nuclear norm among all possible completions is the sum of the absolute values of the diagonal entries, i.e., $|\mu_1| + |\mu_2| + \cdots + |\mu_d|$.

Consider an incomplete matrix $\boldsymbol{M}$ whose associated observational graph is divided into $m$ connected components, denoted by $L_1, L_2, \ldots, L_m$. For each component $L_p$, we define $S_{\boldsymbol{x}}^{L_p}$ as the subset of observed indices within $L_p$, where $1 \le p \le m$ and $S_{\boldsymbol{x}}$ is the set of all observed indices.

For each $L_p$, we can identify row indices $R_p$ and column indices $C_p$ corresponding to the observed entries in $L_p$ as follows:

$$
R_p = \{i | \exists j : (i,j) \in S_{\boldsymbol{x}}^{L_p}\}, \quad C_p = \{j | \exists i : (i,j) \in S_{\boldsymbol{x}}^{L_p}\}.
$$

Here, $R_p$ includes the row indices and $C_p$ includes the column indices of the entries observed in $L_p$.

Define $\boldsymbol{A}^{L_p}$ and $\boldsymbol{B}^{L_p}$ as the submatrices of $\boldsymbol{A}$ and $\boldsymbol{B}$ corresponding to $R_p$ and $C_p$, respectively. The evolution of $\boldsymbol{A}^{L_p}$ is influenced only by $\boldsymbol{B}^{L_p}$. Thus, the dynamics decouple into $m$ independent parts, each equivalent to learning a fully observed matrix $\boldsymbol{M}_i$.

Accordingly, we partition $\boldsymbol{A}$ and $\boldsymbol{B}$ into $m$ blocks:

$$
\boldsymbol{A} = \begin{bmatrix} \boldsymbol{A}_1 \\ \vdots \\ \boldsymbol{A}_m \end{bmatrix}, \quad \boldsymbol{B} = \begin{bmatrix} \boldsymbol{B}_1 & \cdots & \boldsymbol{B}_m \end{bmatrix},
$$

where $\boldsymbol{A}_i = \boldsymbol{A}^{L_i}$ and $\boldsymbol{B}_j = \boldsymbol{B}^{L_j}$.

Denote $L_i = \|\boldsymbol{A}_i \boldsymbol{B}_i - \boldsymbol{M}_i\|_2^2$. The overall loss function $L = \sum_{i=1}^m L_i$ can be decomposed into $m$ independent parts. Performing orthogonal transformations $\tilde{\boldsymbol{A}}_i = \boldsymbol{U}_i^\top \boldsymbol{A}_i$ and $\tilde{\boldsymbol{B}}_i = \boldsymbol{B}_i \boldsymbol{V}_i$, we obtain a diagonal loss $\tilde{L}_i = \|\tilde{\boldsymbol{A}}_i \tilde{\boldsymbol{B}}_i - \Sigma_i\|_2^2$ for $1 \le i \le C$.

Since gradient descent is the steepest descent in the $l_2$ norm and orthogonal transformations preserve this norm, the dynamics of optimizing $\tilde{L}_i$ are equivalent to those of optimizing $L_i$.

Without loss of generality, assume $\mu_1 > \mu_2 > \cdots > \mu_d > 0$. Otherwise, as with Prop. A.9, a sign orthogonal transformation $Q$ can be applied without changing the nuclear norm.

By Prop. A.9, the learning result for a diagonal matrix implies $\tilde{B}_i = \tilde{A}_i^\top$, which means $B_i V_i = U_i^\top A_i^\top$.

The final learning result

$$\tilde{A} = \begin{bmatrix} \tilde{A}_1 \\ \vdots \\ \tilde{A}_m \end{bmatrix}, \quad \tilde{B} = \begin{bmatrix} \tilde{B}_1 & \cdots & \tilde{B}_m \end{bmatrix},$$

satisfies $\tilde{B} = \tilde{A}^\top$.

The result ensures that $\mathrm{diag}(\tilde{A}\tilde{B}) = \mathrm{diag}(\tilde{A}\tilde{A}^\top) = \mathrm{diag}(\Sigma)$. The nuclear norm of $\tilde{A}\tilde{A}^\top$ equals the sum of its eigenvalues, which is the trace of the matrix, and $\mathrm{Tr}(\tilde{A}\tilde{A}^\top) = \mathrm{Tr}(\Sigma) = |\mu_1| + |\mu_2| + \cdots + |\mu_d|$.

Since orthogonal transformations do not alter the nuclear norm of a matrix, the nuclear norm of $W = AB$ is also $|\mu_1| + |\mu_2| + \cdots + |\mu_d|$, concluding the proof. $\square$

## B    Experimental Setup and Supplementary Experiments

In this section, we present the supplementary experiments mentioned in the main text and the details of experiments.

### B.1    Experimental Setup

For all our experiments, we employ gradient descent with a carefully chosen small learning rate. A learning rate is deemed suitable when it yields a smooth, monotonically decreasing training trajectory for the loss function, free from any abrupt fluctuations or oscillations. We initialize all model parameters using a Gaussian distribution with a mean of zero and a variance that is detailed for each specific experiment. Because of the small size of the experiment, the experiment can be completed on a single CPU.

The criterion for the sufficiency of training in all cases is a training loss that falls below $10^{-10}$. To ascertain the rank of the matrix produced by the learning process, we utilize a technique of extrapolation with an infinitesimally small initialization. As depicted in Fig. 3(b), if a singular value persistently diminishes in response to decreasing initialization magnitudes, it is then inferred that such a singular value will not contribute to the rank in the context of an infinitesimal initialization.

We have included code in the Supplementary Material that determines the connectivity of a partially observed matrix and provides specific examples illustrating the implicit regularization effects. This code can be used to reproduce our results and explore the relationship between data connectivity and the implicit biases of matrix factorization models in various matrix completion scenarios.

### B.2    Connectivity Experiments

In the connectivity experiments corresponding to Fig. 1, we explore the behavior of randomly generated $4 \times 4$ matrices with intrinsic ranks of 1, 2, and 3. To investigate the impact of sampling density on matrix reconstruction, we sample matrices at three different levels: $2rd - r^2$, which meets the threshold for exact reconstruction, $2rd - r^2 - 1$, which is just below the threshold, and $2rd - r^2 + 1$, which exceeds the threshold.

For each sampling size, we randomly generate 10 sets of sampling positions. We then assess the connectivity of the sampled positions and compute both the rank and the nuclear norm of the solutions obtained through gradient descent. As an illustration, in Fig. B1, panel (a) presents a scenario with connected sampling positions, panel (b) shows disconnected sampling positions, and panel (c) depicts disconnected sampling with each disconnected component forming a complete bipartite graph.

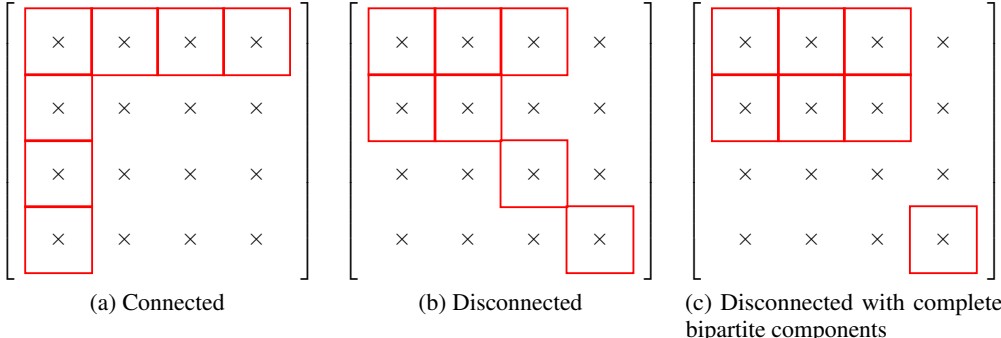

(a) Connected      (b) Disconnected      (c) Disconnected with complete bipartite components

Figure B1: Examples of connected sampling and disconnected sampling patterns in Fig. 1.

In the connectivity experiments depicted in Figs. 2(c-d), we examine the behavior of randomly generated matrices of size $4 \times 4$ and $10 \times 10$ with a rank of 1. The matrices are sampled at a size of $2rd - r^2$, which corresponds to the threshold for exact reconstruction. We evaluate two connected and one disconnected sampling patterns.

Fig. B2(a) displays the first connected sampling pattern, where all entries in the first row and the first column are sampled. Fig. B2(b) illustrates the second connected sampling pattern, which forms a "Z" shape across the matrix. Fig. B2(c) shows the disconnected sampling pattern, where the samples are split into two unconnected blocks, one in the top-left and the other in the bottom-right of the matrix. A similar approach is taken for the $10 \times 10$ matrices.

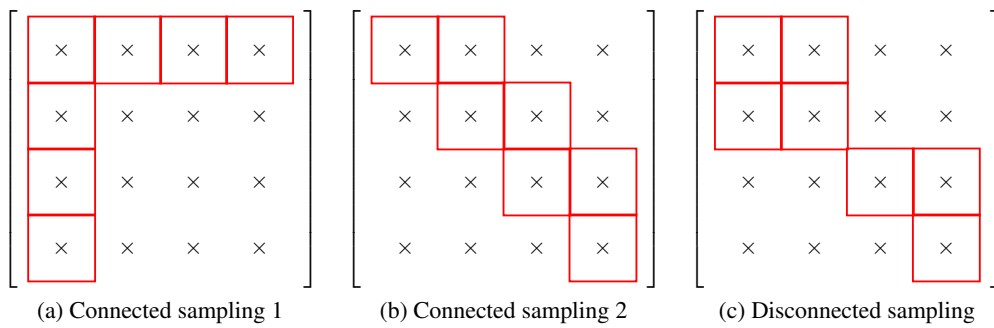

(a) Connected sampling 1      (b) Connected sampling 2      (c) Disconnected sampling

Figure B2: Examples of connected sampling and disconnected sampling patterns in Fig. 2(c).

For Figs. 2 (a-b), we performed 100 random initializations for each initialization scale, recorded the mean and standard deviation, and plotted them on the figure. For a sequence $x_1, x_2, \cdots, x_n$, the mean is calculated by $\bar{x} = \frac{1}{n} \sum_{i=1}^{n} x_i$, and the standard deviation is calculated by:

$$\sigma = \sqrt{\frac{1}{n-1} \sum_{i=1}^{n} (x_i - \bar{x})^2}.$$

Figs. 2(c-d) demonstrate that when the target matrix has a rank of 1 and the number of samples meets the minimum requirement for reconstruction with connected sampling positions, the matrix factorization model is capable of accurately reconstructing the original target matrix.

In scenarios where the target matrix has a higher rank, we have extended our experiments accordingly. For a randomly chosen $4 \times 4$ matrix with rank 2, we selected a sample count less than the threshold of $2rd - r^2 = 12$, specifically 10 samples, while ensuring that the sampling pattern is connected, as shown in Fig. B3. The resulting solution from the matrix completion has a rank of 2, which is the minimal rank that fits the sampled data.

Fig. B4 reveals distinct behaviors of the matrix completion depending on the scale of initialization. With a larger initialization, the third and fourth singular values of the completed matrix remain

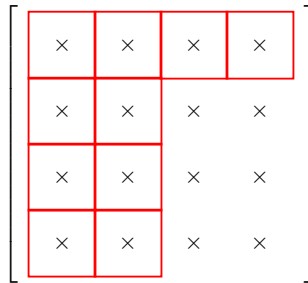

Figure B3: The sample pattern in Fig. B4.

relatively significant, suggesting that the model does not converge to the lowest rank solution. On the other hand, with a smaller initialization, the third and fourth singular values are uniformly small, indicating that the model successfully converges to the lowest rank solution.

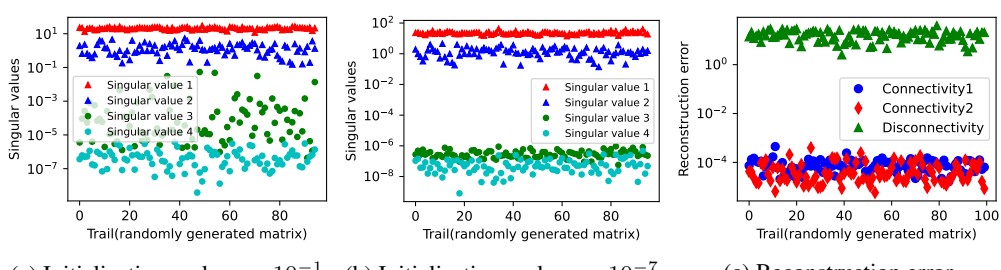

(a) Initialization scale $\sigma = 10^{-1}$    (b) Initialization scale $\sigma = 10^{-7}$    (c) Reconstruction error

Figure B4: For a randomly selected $4 \times 4$ matrix with rank 2, we chose 10 samples, fewer than the threshold $2rd - r^2 = 12$, ensuring connected sampling positions, as shown in Fig. B3. The figures show the experimental results for the four singular values of the matrix learned under Gaussian initialization with mean 0 and standard deviations of $10^{-1}$ (a) and $10^{-7}$ (b), respectively. (c) Reconstruction error of the solutions for a $4 \times 4$ matrix reconstruction problem with $M^*$ randomly sampled at rank $r = 1$ and sample size set to the minimum reconstruction setting $n = 2rd - r^2$. Red and blue scatter points represent two connected sampling patterns, while green points represent a disconnected pattern.

### B.3    Equivalent Sampling Patterns

For a given sample size, there are different sampling models corresponding to connected or disconnected sampling. As shown in Fig. B1, 7 observations are sampled, but different sampling positions affect connectivity or disconnection. To thoroughly study all possible cases, we examine all sampling cases of a $3 \times 3$ matrix completion, as illustrated in Figure 2(c).

For a $3 \times 3$ matrix, the sample size varies from 1 to 9. When using the matrix decomposition model $f_\theta = AB$ for matrix completion, the dynamics obtained by exchanging rows or columns or transposing the matrix to be completed are equivalent. These three operations allow us to divide all sampling patterns equally.

For sample size = 1, there is only one sampling pattern in the equivalent sense, and the observation matrix $P$ is:

$$P_1 = \begin{bmatrix} 1 & 0 & 0 \\ 0 & 0 & 0 \\ 0 & 0 & 0 \end{bmatrix}.$$

where 1 indicates that the position is observed and non-zero, and 0 means that the position is not observed or is 0.

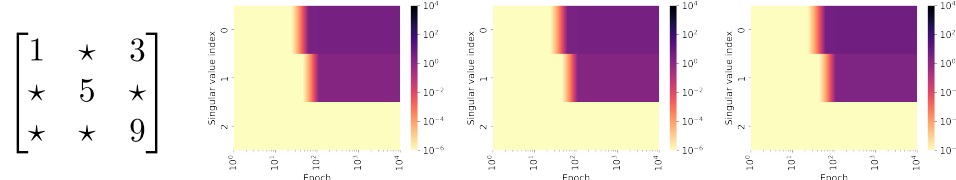

(a) Matrix completion  (b) Singular values of $\boldsymbol{A}$  (c) Singular values of $\boldsymbol{B}$  (d) Singular values of $\boldsymbol{W}_{\mathrm{aug}}$

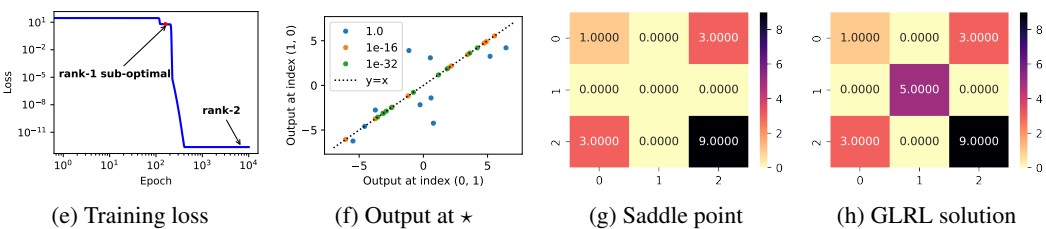

(e) Training loss  (f) Output at $\star$  (g) Saddle point  (h) GLRL solution

Figure B5: (a) The matrix to be completed, with unknown entries marked by $\star$. (b-d) Evolution of singular values for $\boldsymbol{A}$, $\boldsymbol{B}$, and $\boldsymbol{W}_{\mathrm{aug}}$ during training. (e) Training loss for the disconnected sampling pattern. (f) Learned values at symmetric positions $(0,1)$ and $(1,0)$ under varying initialization scales (zero mean, varying variance). Each point represents one of ten random experiments per variance; labels show initialization variance. Other symmetric positions exhibit similar behavior. (g) Learned output at the saddle point corresponding to the red dot in (e). (h) Final learned solution of the GLRL algorithm (Li et al., 2020).

For sample size = 2, there are only 2 sampling patterns in the equivalent sense:

$$\boldsymbol{P}_1 = \begin{bmatrix} 1 & 1 & 0 \\ 0 & 0 & 0 \\ 0 & 0 & 0 \end{bmatrix}, \boldsymbol{P}_2 = \begin{bmatrix} 1 & 0 & 0 \\ 0 & 1 & 0 \\ 0 & 0 & 0 \end{bmatrix}$$

For sample size = 3, there are only 4 sampling patterns in the equivalent sense:

$$\boldsymbol{P}_1 = \begin{bmatrix} 1 & 1 & 1 \\ 0 & 0 & 0 \\ 0 & 0 & 0 \end{bmatrix}, \boldsymbol{P}_2 = \begin{bmatrix} 1 & 1 & 0 \\ 1 & 0 & 0 \\ 0 & 0 & 0 \end{bmatrix}, \boldsymbol{P}_3 = \begin{bmatrix} 1 & 1 & 0 \\ 0 & 0 & 1 \\ 0 & 0 & 0 \end{bmatrix}, \boldsymbol{P}_4 = \begin{bmatrix} 1 & 0 & 0 \\ 0 & 1 & 0 \\ 0 & 0 & 1 \end{bmatrix}$$

For sample size = 4, there are only 5 sampling patterns in the equivalent sense:

$$\boldsymbol{P}_1 = \begin{bmatrix} 1 & 1 & 1 \\ 1 & 0 & 0 \\ 0 & 0 & 0 \end{bmatrix}, \boldsymbol{P}_2 = \begin{bmatrix} 1 & 1 & 0 \\ 1 & 1 & 0 \\ 0 & 0 & 0 \end{bmatrix}, \boldsymbol{P}_3 = \begin{bmatrix} 1 & 1 & 0 \\ 0 & 1 & 1 \\ 0 & 0 & 0 \end{bmatrix}, \boldsymbol{P}_4 = \begin{bmatrix} 1 & 1 & 0 \\ 1 & 0 & 0 \\ 0 & 0 & 1 \end{bmatrix}, \boldsymbol{P}_5 = \begin{bmatrix} 1 & 1 & 0 \\ 0 & 0 & 1 \\ 0 & 0 & 1 \end{bmatrix}$$

For sample size = 5, there are only 3 sampling patterns in the equivalent sense:

$$\boldsymbol{P}_1 = \begin{bmatrix} 1 & 1 & 1 \\ 1 & 1 & 0 \\ 0 & 0 & 0 \end{bmatrix}, \boldsymbol{P}_2 = \begin{bmatrix} 1 & 1 & 1 \\ 1 & 0 & 0 \\ 1 & 0 & 0 \end{bmatrix}, \boldsymbol{P}_3 = \begin{bmatrix} 1 & 1 & 0 \\ 1 & 1 & 0 \\ 0 & 0 & 1 \end{bmatrix}$$

For sample size = 6, there are only 4 sampling patterns in the equivalent sense:

$$\boldsymbol{P}_1 = \begin{bmatrix} 1 & 1 & 1 \\ 1 & 1 & 1 \\ 0 & 0 & 0 \end{bmatrix}, \boldsymbol{P}_2 = \begin{bmatrix} 1 & 1 & 1 \\ 1 & 1 & 0 \\ 0 & 0 & 1 \end{bmatrix}, \boldsymbol{P}_3 = \begin{bmatrix} 1 & 1 & 1 \\ 1 & 1 & 0 \\ 1 & 0 & 0 \end{bmatrix}, \boldsymbol{P}_4 = \begin{bmatrix} 1 & 1 & 0 \\ 0 & 1 & 1 \\ 1 & 0 & 1 \end{bmatrix}$$

For sample size = 7, there are only 2 sampling patterns in the equivalent sense:

$$\boldsymbol{P}_1 = \begin{bmatrix} 1 & 1 & 1 \\ 1 & 1 & 1 \\ 1 & 0 & 0 \end{bmatrix}, \boldsymbol{P}_2 = \begin{bmatrix} 1 & 1 & 1 \\ 1 & 1 & 0 \\ 0 & 1 & 1 \end{bmatrix}$$

For sample size = 8, there is only 1 sampling pattern in the equivalent sense:

$$\boldsymbol{P}_1 = \begin{bmatrix} 1 & 1 & 1 \\ 1 & 1 & 1 \\ 1 & 1 & 0 \end{bmatrix}$$

For sample size = 9, there is only 1 sampling pattern in the equivalent sense:

$$\boldsymbol{P}_1 = \begin{bmatrix} 1 & 1 & 1 \\ 1 & 1 & 1 \\ 1 & 1 & 1 \end{bmatrix}$$

## B.4   Initialization Scale Analysis

Our experimental findings indicate that when the observational data is connected, matrix factorization models often learn the lowest-rank solution starting from a small initialization. However, the required scale of initialization is not constant across different instances. We empirically observed that if the magnitude of the numerical values in the matrix to be completed varies significantly, an extremely small initialization is necessary, which, in some cases, can exceed machine precision.

Consider the following two simple $2 \times 2$ matrix completion problems, with the only difference being that the number 3 in the first row is replaced by 20. When training begins from a small initialization, for $\boldsymbol{M}_4$, the fourth element only needs to learn the value 6 to be a rank-1 solution. However, for $\boldsymbol{M}_5$, the fourth element needs to learn the value 40 to achieve rank-1.

$$\boldsymbol{M}_4 = \begin{bmatrix} 1 & 2 \\ 3 & \times \end{bmatrix}, \qquad \boldsymbol{M}_5 = \begin{bmatrix} 1 & 2 \\ 20 & \times \end{bmatrix}.$$

Fig. B6 illustrates the difficulty in learning these two examples. For $\boldsymbol{M}_4$, an initialization variance of approximately $10^{-7}$ is sufficient to learn the lowest-rank solution. In contrast, for $\boldsymbol{M}_5$, an extremely small initialization variance is required, making it challenging to learn a rank-1 solution. Yet, with an exceedingly small initialization variance of $10^{-83}$, we can still observe the second singular value plummeting to zero. The origin is a saddle point. The smaller the initialization, the longer it will stay at the origin. If initialization continues to decrease, training will stagnate. Therefore, if the magnitude difference is even greater, such as replacing 20 with 100 in $\boldsymbol{M}_5$, then with the small initialization allowed by machine precision, it is nearly impossible to learn the value 200 completely.

For the matrix factorization model $\boldsymbol{f}_\theta = \boldsymbol{AB}$, the Hessian matrix at 0 has strictly negative eigenvalues, making the origin a strict saddle point. Under small random initialization, gradient descent escapes this saddle at an exponential rate. Our Theorem 1 ensures that only strict saddle points and global minima exist as critical points in the loss landscape. Subsequent saddle points on invariant manifolds are also strict, with exponential escape speeds, maintaining acceptable optimization process.

Reaching the lowest rank solution requires parameters to escape the saddle point along the unique top eigen-direction. Fig. B6 illustrates the relationship between required initialization scale and observation magnitude differences. For lowest possible rank with large numerical magnitude differences in observations, extremely small initialization is necessary. However, for approximately low rank solutions, a relatively small initialization suffices.

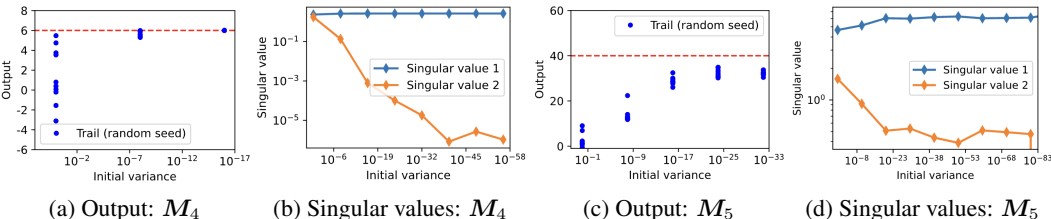

(a) Output: $\boldsymbol{M}_4$     (b) Singular values: $\boldsymbol{M}_4$     (c) Output: $\boldsymbol{M}_5$     (d) Singular values: $\boldsymbol{M}_5$

Figure B6: (a, c) The value of the $(2, 2)$ element learned by the model as the initialization decreases. (b, d) The singular values of the matrix learned by the model with decreasing initialization.

## B.5 High-dimensional Experiments

To validate the scalability of our findings, we extended our experiments to higher-dimensional matrices. We conducted tests on $20 \times 20$ matrices, employing both connected (Fig. B7) and disconnected (Fig. B8) sampling patterns, while monitoring rank evolution during training. Our results consistently corroborated the main findings:

(i) Connected observations converged to optimal low-rank solutions.

(ii) Disconnected observations yielded higher-rank solutions.

(iii) The Hierarchical Invariant Manifold Traversal (HIMT) process was observed in both connected and disconnected scenarios.

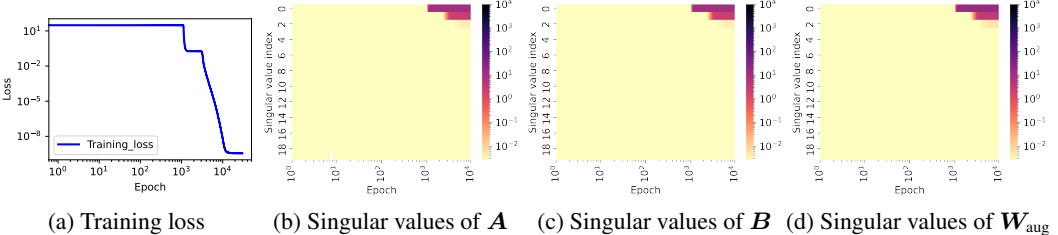

    (a) Training loss      (b) Singular values of $\boldsymbol{A}$    (c) Singular values of $\boldsymbol{B}$    (d) Singular values of $\boldsymbol{W}_{\mathrm{aug}}$

Figure B7: **Connected sampling pattern analysis for a** $20 \times 20$ **random rank-2 matrix completion problem.** (a) Training loss under small initialization. (b-d) Singular value evolution for $\boldsymbol{A}, \boldsymbol{B}, \boldsymbol{W}_{\mathrm{aug}}$.

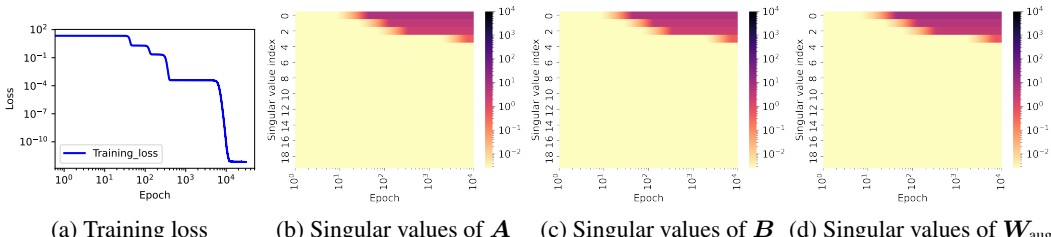

    (a) Training loss      (b) Singular values of $\boldsymbol{A}$    (c) Singular values of $\boldsymbol{B}$    (d) Singular values of $\boldsymbol{W}_{\mathrm{aug}}$

Figure B8: **Disconnected sampling pattern analysis for a** $20 \times 20$ **random rank-2 matrix completion problem.** (a) Training loss under small initialization. (b-d) Singular value evolution for $\boldsymbol{A}, \boldsymbol{B}, \boldsymbol{W}_{\mathrm{aug}}$.

## B.6 Dynamics of Deep Matrix Factorization

In the context of depth-3 matrix factorization models, we consider the functional form:

$$\boldsymbol{f_\theta} = \boldsymbol{ABC}, \quad \text{where} \quad \boldsymbol{A}, \boldsymbol{B}, \boldsymbol{C} \in \mathbb{R}^{d \times d}.$$

Figs. B9 and B10 suggest that even for a depth-3 model, the learning process exhibits a progression from low rank to high rank structures.

## B.7 Incorporating Attention Mechanisms

Within the Transformer architecture, the matrix factorization component retains its significance. The attention mechanism is formalized as follows:

$$\boldsymbol{f_\theta}(\boldsymbol{X}) = \sum_{i=1}^{h} \mathrm{softmax}_{\mathrm{row}} \left( \frac{\boldsymbol{X} \boldsymbol{W_{Q_i}} \boldsymbol{W_{K_i}^\top} \boldsymbol{X}^\top}{\sqrt{d_k}} \right) \boldsymbol{X} \boldsymbol{W_{V_i}} \boldsymbol{W_{O_i}},$$

where the row-wise softmax operation is applied to the attention scores, and the sum is over the $h$ different attention heads, with $\boldsymbol{W_{Q_i}}, \boldsymbol{W_{K_i}}, \boldsymbol{W_{V_i}}, \boldsymbol{W_{O_i}}$ representing the learnable weight matrices

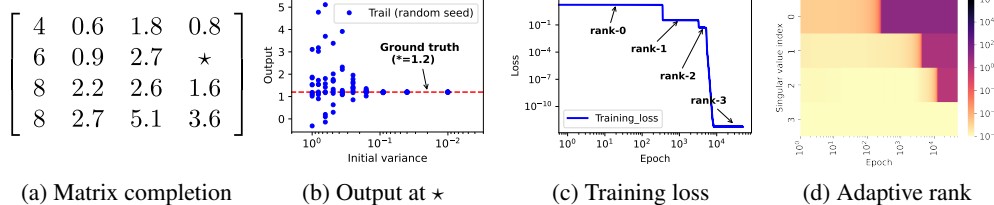

(a) Matrix completion      (b) Output at $\star$      (c) Training loss      (d) Adaptive rank

Figure B9: **Deep Matrix factorization models learn adaptively from low rank to high rank with small initialization.** (a) The matrix to be completed, the $\star$ position is unknown. (b) The value at $\star$ learned under different initialization scales (mean is zero, variance changes), 10 random experiments were done under each variance, and each blue point represents an experiment. (c) The training loss curve with an initial variance of $10^{-14}$. (d) Evolution of singular values for $f_{\theta} = ABC$ during training. The count of significantly non-zero singular values is indicative of the rank.

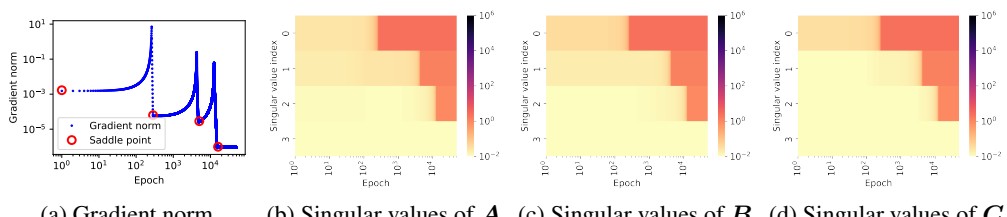

(a) Gradient norm    (b) Singular values of $A$    (c) Singular values of $B$    (d) Singular values of $C$

Figure B10: **Deep Matrix factorization models learn adaptively from low rank to high rank with small initialization.** (a) Evolution of the $l_2$-norm of the gradients for all parameters throughout the training process. (b-d) Evolution of singular values for matrices $A, B$, and $C$ during training. The count of non-zero singular values is indicative of the rank.

for queries, keys, values, and output transformations, respectively, and $d_k$ is the dimensionality of the key vectors.

The attention module's ability to capture low-rank representations is reflected in the depth-2 matrix factorization model. As illustrated in Fig. B11, the attention models consistently learns representations that evolve from lower to higher ranks.

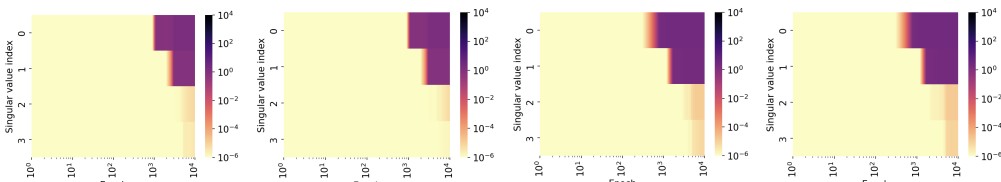

(a) Singular values of $W_Q$ (b) Singular values of $W_K$ (c) Singular values of $W_V$ (d) Singular values of $W_O$

Figure B11: **The attention modules in Transformer learn adaptively from low rank to high rank with small initialization.** (a-d) The evolution of singular values for the matrices $W_Q, W_K, W_V$, and $W_O$ throughout the training process. The number of significantly non-zero singular values suggests the effective rank of each matrix.

