# OpenReview forum: "Connectivity Shapes Implicit Regularization in Matrix Factorization Models for Matrix Completion"
_NeurIPS.cc/2024/Conference — NeurIPS 2024 poster_

### Official Review · Reviewer_1kCy · 2024-06-24

**Soundness:** 3
**Presentation:** 4
**Contribution:** 2
**Rating:** 6
**Confidence:** 3

**Summary:**

The paper studies implicit regularization in matrix completion, a problem that estimates missing entries from a partially observed matrix. In particular, in this paper, matrix completion is formulated as an optimization problem of the form: $\min_{A, B} f(A,B) = \|M - AB\|_S^2$ where $S$ is the subset of observed entries and the function $\|\cdot\|_S$ is the Frobenius norm of a matrix after masking entries outside $S$ to zero. Using this formulation, the authors study the implicit regularization of the gradient flow associated with $f$ whose initial condition is close to zero. Their main results characterize the influence of connectivity - a property defined based on $S$ - on the dynamics and solutions obtained by the gradient flow. The results are supported both theoretically and numerically.

**Strengths:**

- The results seem to be correct. Although I did not check all the details, there are certain lemmas/theorems that I checked carefully and I did not find any serious problem.
- The presentation is excellent. Many definitions are followed by examples and/or remarks to provide intuitions/insightful discussion. The experiments are quite convincing.

**Weaknesses:**

- The main concern is the interest of the problem: why should we care about data connectivity (in the sense of this paper)? I think the motivation of this study needs to be further justified.

**Questions:**

1. Major questions/suggestions:
- I wonder under which regime of $n$ (the size of the matrix) and $m$ (the number of observed entries) do we have more than/exactly one connected component? I think this question is important to justify the motivation of this work.
2. Minor questions/suggestions:
- The experiments in Section 4 need to be provided with more details: What is the algorithm used? If it is gradient descent, what is the learning rate? Or the solution is obtained by solving an ODE.
- Throughout the introduction, the term "connectivity" is repeatedly used but its description or definition is not provided (it is delayed until Section 3). The authors might consider describing/defining this term earlier to improve the readability.
- Line 612: The second inequality should be $\|w^T B\|\_F^2$ instead of $\|w^T B\|\_{S_x}^2$.

**Limitations:**

There are no particular limitations or potential negative impacts of this paper.

---

> ### Author Rebuttal · Authors · 2024-08-06
>
> * **Point 1: Why should we care about data connectivity (in the sense of this paper)? I think the motivation of this study needs to be further justified.**
>
> * Reply: We appreciate the reviewer's question regarding the importance of data connectivity in our study. As the title of our paper suggests, "Connectivity Shapes Implicit Regularization in Matrix Factorization Models for Matrix Completion", connectivity plays a crucial role in understanding the implicit regularization behavior of matrix factorization models. This concept is closely related to recent research on nuclear norm minimization and rank minimization mentioned in the Introduction and Related Works.
>
>   - From an experimental perspective, our initial motivation stemmed from extensive experiments exploring the conditions under which low-rank tendencies emerge or fail to emerge. Through a series of simple experiments and analysis of their dynamics, we discovered that the connectivity of the observed data played a pivotal role, as illustrated in the Introduction.
>
>   - Theoretically, the ongoing debate about whether the implicit regularization of matrix factorization models tends towards low nuclear norm or low rank necessitates a clear definition of the conditions that determine these outcomes. Our experiments provided the initial motivation for developing a theoretical framework. In fact, connectivity is closely related to whether the dynamic system can be decoupled into several parts (please refer to Proposition A.4 in Appendix A), which inspired our subsequent theoretical investigations.
>
>   This study thus bridges a crucial gap in our understanding of how data structure influences the behavior of matrix factorization models, providing insights that are valuable for both theoretical analysis and practical applications in matrix completion tasks.
>
> * **Point 2: I wonder under which regime of $n$ (the size of the matrix) and $m$ (the number of observed entries) do we have more than/exactly one connected component? I think this question is important to justify the motivation of this work.**
>
> * Reply: We thank the reviewer for this insightful question about the relationship between matrix size, number of observations, and connectivity. To address this, we've added the following discussion in Appendix B:
>
>   "Threshold function: For a square $n\times n$ matrix, the threshold for connectivity is $m = (n-1)^2 + 1$. Below this threshold, disconnected components are likely; above it, a single connected component is guaranteed. The intuition is that for a square $n\times n$ matrix, if the upper left $(n-1)\times (n-1)$ sub-block is fully observed, and the last $(n,n)$ position is observed, this represents the case with the maximum number of observations while still being disconnected. If one more entry is observed, the entire observation set becomes connected. However, if there are fewer than $m = (n-1)^2 + 1$ observations, both connected and disconnected situations may occur, depending on the specific sampling pattern."
>
> * **Point 3: The experiments in Section 4 need to be provided with more details: What is the algorithm used? If it is gradient descent, what is the learning rate? Or the solution is obtained by solving an ODE.**
>
> * Reply: We appreciate the reviewer's request for more experimental details. We have added the following information in Section 3 to briefly introduce the experimental setup, with further details provided in Appendix B.1:
>
>   "The training dynamics follow the gradient flow of $R_S(\boldsymbol{\theta})$:
>   $$ \frac{\mathrm{d} \boldsymbol{\theta}}{\mathrm{d} t}=-\nabla_{\boldsymbol{\theta}} R_S(\boldsymbol{\theta}), \quad \boldsymbol{\theta}(0)=\boldsymbol{\theta}_0 . $$
>   In all experiments, $\theta_0 \sim N(0, \sigma^2)$ is initialized from a Gaussian distribution with mean 0 and small variance $\sigma^2$. We use gradient descent with a small learning rate to approximate the gradient flow dynamics (please refer to Appendix B.1 for the detailed experimental setup)."
>
> * **Point 4: Throughout the introduction, the term "connectivity" is repeatedly used but its description or definition is not provided (it is delayed until Section 3). The authors might consider describing/defining this term earlier to improve the readability.**
>
> * Reply: We thank the reviewer for this suggestion to improve readability. We've added a brief description of connectivity in the Introduction:
>
>   "Data connectivity, in the context of this paper, refers to the way observed entries in the matrix are linked through shared rows or columns. A set of observations is considered connected if there's a path between any two observed entries via other observed entries in the same rows or columns. This concept plays a crucial role in determining the behavior of matrix factorization models, as we will demonstrate throughout this paper."
>
> - **Point 5: Line 612: The second inequality should be $\\|w^\top B\\|^2_F$ instead of $\\|w^\top B\\|^2_{S_x}$.**
>
> * Reply: We sincerely thank the reviewer for catching this typographical error. We have corrected the inequality to $\\|w^\top B\\|^2_F$ as suggested. This correction ensures the mathematical accuracy of our presentation.
>
> We greatly appreciate the reviewer's thorough examination of our work. Your insightful comments and suggestions have significantly contributed to improving the clarity, precision, and overall quality of our presentation.

---

> > ### Comment · Reviewer_1kCy · 2024-08-10
> >
> > Thanks the authors for their responses. I am quite satisfied with their answers, except for the second point: the threshold $(n - 1)^2 + 1$ seems too large for a graph to be connected. The classical random graph model yields a much better result, see [this link](https://mathoverflow.net/questions/60075/connectivity-of-the-erd%C5%91s-r%C3%A9nyi-random-graph). This is, however, just for information. I keep my score as it is and recommend acceptance.

---

### Official Review · Reviewer_PxmR · 2024-07-07

**Soundness:** 3
**Presentation:** 3
**Contribution:** 2
**Rating:** 6
**Confidence:** 2

**Summary:**

This study empirical reveals that the connectivity of observed data significantly influences the implicit bias and identifies a hierarchy of intrinsic invariant manifolds in the loss landscape, providing a preliminary framework for understanding the mechanisms behind implicit regularisation in matrix factorisation models.

**Strengths:**

1, The paper is clearly written.\
2, Great presented figures illustrate experimental results and findings, making them easier for readers to understand.\
3, Differences from prior similar works have been properly addressed.

**Weaknesses:**

Please refer to the "Questions".

**Questions:**

1, Have you verified your findings on high dimensional matrix with experiments?\
2, What's the intuition of the augmented matrix? See $\textit{Line 110}$?\
3, What exactly "small initialisation" you defined in the experiments? See $\textit{Line 164}$\
Minor suggestion(s):\
$\textbf{A.}$ $Eq.3$ after $\textit{Line 117}$: You have mentioned $\theta(0) = \theta_0$, which indicates the initialisation (if I understand correctly). However, the initialisation method was mentioned far from here at $\textit{Line 36-37}$. It could be easier to follow if you can optimise this.\
$\textbf{B.}$ I expect a detailed example of connectivity and disconnectivity in the appendix (see $\textit{Line 561 - 564 and Fig. A1}$), which will make it easier for a wider audience to understand.

**Limitations:**

Yes.

---

> ### Author Rebuttal · Authors · 2024-08-06
>
> * **Point 1: Have you verified your findings on high dimensional matrix with experiments?**
>
> * Reply: We sincerely appreciate the reviewer's important question regarding the scalability of our findings. While our main results focus on smaller matrices for clarity of presentation, we have indeed conducted additional experiments on higher-dimensional matrices to verify the broader applicability of our findings.
>
>   We've added a new subsection in the Appendix titled "B.7 High-Dimensional Experiments" with the following content:
>
>   "To validate the scalability of our findings, we extended our experiments to higher-dimensional matrices. We conducted tests on  randomly generated $20\times 20$ matrices, employing both connected (Fig. 2) and disconnected (Fig. 3) sampling patterns, while monitoring rank evolution during training. Our results consistently corroborated the main findings (please refer to high-dimensional experiments in the attached PDF):
>
>   - (i) Connected observations converged to optimal low-rank solutions.
>   - (ii) Disconnected observations yielded higher-rank solutions.
>   - (iii) The Hierarchical Invariant Manifold Traversal (HIMT) process was observed in both connected and disconnected scenarios."
>
> * **Point 2: What's the intuition of the augmented matrix? See Line 110?**
>
> * Reply: We appreciate the request for clarification on the augmented matrix. The augmented matrix $W_{\text{aug}}$ is indeed a key construct in our analysis, and we recognize the need for a more intuitive explanation.
>
>   We've added the following explanation after Line 110:
>
>   "The augmented matrix $W_\text{aug}$ plays a crucial role in our subsequent analysis, particularly in characterizing the intrinsic invariant manifolds $\Omega_k$ of the optimization process. Specifically, it allows us to establish the relationship $\text{rank}(A) = \text{rank}(B^\top) = \text{rank}(W_{\text{aug}})$, which is important to understanding the invariance property under gradient flow."
>
> * **Point 3: What exactly "small initialization" you defined in the experiments? See Line 164.**
>
> * Reply: We thank the reviewer for pointing out the need for a clearer definition of "small initialization". To address this, we've revised the description in Line 164 as follows:
>
>   "For a small random initialization (Gaussian distribution with mean 0 and variance $10^{-16}$), the loss curves exhibit a steady, stepwise decline"
>
> * **Point 4: You have mentioned $\theta(0) = \theta_0$, which indicates the initialization (if I understand correctly). However, the initialization method was mentioned far from here at Line 36-37. It could be easier to follow if you can optimise this.**
>
> * Reply: We appreciate the reviewer's suggestion to improve the flow and clarity of our presentation regarding initialization. We have revised the relevant section as follows:
>
>   "The training dynamics follow the gradient flow of $R_S(\boldsymbol{\theta})$:
>   $$ \frac{\mathrm{d} \boldsymbol{\theta}}{\mathrm{d} t}=-\nabla_{\boldsymbol{\theta}} R_S(\boldsymbol{\theta}), \quad \boldsymbol{\theta}(0)=\boldsymbol{\theta}_0 . $$
>   In all experiments, $\theta_0 \sim N(0, \sigma^2)$ is initialized from a Gaussian distribution with mean 0 and small variance $\sigma^2$. We use gradient descent with a small learning rate to approximate the gradient flow dynamics (please refer to Appendix B.1 for the detailed experimental setup)."
>
> * **Point 5: I expect a detailed example of connectivity and disconnectivity in the appendix (see Line 561 - 564 and Fig. A1), which will make it easier for a wider audience to understand.**
>
> * Reply: We appreciate the reviewer's suggestion to provide a more detailed example of connectivity and disconnectivity. We've expanded the examples in the appendix (Lines 561-564 and Fig. A1) with a step-by-step explanation:
>
>   "Examples of connectivity and disconnectivity.
>    - Consider three matrices to be completed, each obtained by adding one more observation to the previous matrix:
>     $$ \boldsymbol{M}_1=\left[\begin{array}{ccc} 1 & 2 & \star \\\\ 3 & \star & \star \\\\ \star & \star & 5 \end{array}\right], \boldsymbol{M}_2=\left[\begin{array}{ccc} 1 & 2 & \star \\\\ 3 & 4 & \star \\\\ \star & \star & 5 \end{array}\right], \boldsymbol{M}_3=\left[\begin{array}{ccc} 1 & 2 & \star \\\\ 3 & 4 & \star \\\\ 6 & \star & 5 \end{array}\right]. $$
>
>   - The corresponding observation matrices $\boldsymbol{P}$ are:
>   $$ \boldsymbol{P}_1=\left[\begin{array}{ccc} 1 & 1 & 0 \\\\ 1 & 0 & 0 \\\\ 0 & 0 & 1 \end{array}\right], \boldsymbol{P}_2=\left[\begin{array}{ccc} 1 & 1 & 0 \\\\ 1 & 1 & 0 \\\\ 0 & 0 & 1 \end{array}\right], \boldsymbol{P}_3=\left[\begin{array}{ccc} 1 & 1 & 0 \\\\ 1 & 1 & 0 \\\\ 1 & 0 & 1 \end{array}\right]. $$
>
>   - And by Definition 1, the corresponding adjacency matrices are:
>   $$ \boldsymbol{A}_1=\left[\begin{array}{ccc} \boldsymbol{0} & \boldsymbol{P}_1^\top \\\\ \boldsymbol{P}_1 & \boldsymbol{0} \end{array}\right], \boldsymbol{A}_2=\left[\begin{array}{ccc} \boldsymbol{0} & \boldsymbol{P}_2^\top \\\\ \boldsymbol{P}_2 & \boldsymbol{0} \end{array}\right], \boldsymbol{A}_3=\left[\begin{array}{ccc} \boldsymbol{0} & \boldsymbol{P}_3^\top \\\\ \boldsymbol{P}_3 & \boldsymbol{0} \end{array}\right]. $$
>
>   - Given the adjacency matrix $A_i$, according to Definition 1, we can obtain a bipartite graph $G_{M_i}$, which we refer to as the associated observation graph. Fig. A1 in Appendix A illustrates the associated graphs $G_{M_i}$, from which we can see that $M_1$ is disconnected, with its associated observation graph consisting of two connected components. $M_2$ is also disconnected, but each connected component of its associated observation graph forms a complete bipartite subgraph. In contrast, $M_3$ is connected, and its associated observation graph consists of a single connected component."
>
> We sincerely appreciate the reviewer's thorough examination of our work, which has significantly contributed to improving the clarity and precision of our presentation.

---

> > ### Comment · Reviewer_PxmR · 2024-08-11
> >
> > I appreciate the authors' thorough response. After careful consideration, I have decided to raise my score.

---

### Official Review · Reviewer_AjkV · 2024-07-08

**Soundness:** 4
**Presentation:** 4
**Contribution:** 4
**Rating:** 9
**Confidence:** 5

**Summary:**

This paper studies the training dynamics of matrix factorisation for matrix completion, optimising the vanilla mean-squared-error loss function via gradient descent with small initialisation. The authors characterise the observation pattern of the underlying matrix via the connectivity of its associated bipartite graph, and show both empirically and theoretically that this plays an important role in the implicit regularisation of the learned solution.

The paper provides empirical evidence that in both the connected and disconnected cases, the row/column spaces of the factor matrices remain aligned at all times, and that the loss curves exhibit steady step-wise decline as the solutions traverse solutions of increasing rank. In the connected case, it is observed that the learned solutions of each rank are near optimal, however in the disconnected case, suboptimal solutions are learned.

These empirical observations and explained by theoretical results which examine the training dynamics of the continuous dynamical system following the gradient flow of the loss function with infinitesimal initialisation. The authors begin by defining a Hierarchical Inrinsic Invariant Manifold and Proposition 1 describes how the gradient flow behaves on these manifolds and that they form a hierarchy. Proposition 2 shows that in the special case that when the observation graph is disconnected, the gradient flow dynamics are restricted to sub-manifolds. Theorem 1 (which extends an analgous result for symmetric matrix factorization models in Li et al. (2020)) shows that the critical points of the gradient flow are either strict saddle points or global minima and Theorem 2 shows that the gradient flow traverses the hierarchy of Intrinsic Invariant Manifolds (or sub-manifolds in the disconnected case), a phenomenon coined "Hierarchical Invariant Manifold Traversal". The flagship results of the paper are Theorems 3 and 4 which show that in the connected case, the gradient flow converges to the minimum rank solution, and in the a special disconnected case (disconnected with complete bipartite components), the gradient flow converges to the minimum nuclear norm solution.

**Strengths:**

This paper makes a significant and original contribution to the understanding of implicit bias of gradient descent for matrix factorisation. The key strength of this paper is showing, both empirically and theoretically, how the connectivity of the observation graph impacts the training dynamics, and how this affects the final solution. Distinguishing the connected and disconnected cases is novel and enlightening, and the detail with which the training dynamics are theoretically characterised is eye-opening.

The paper is beautifully written: the problem is set out very clearly, the relevant existing literature is summarises comprehensively and concisely. The reader is guided through simple toy experiments which  precisely demonstrate the phenomenon which is described. The theoretical results then provide a satisfying explanation for the empirical observations. I really felt like I had gained a deep understanding of the training dynamics for this problem by the time I had finished reading this paper. While the theoretical results are technical, the intuition I gained in the first half of the paper really helped me to understand the details. In addition, the results and the proofs are written very clearly and accurately.

**Weaknesses:**

I find this work hard to fault. While the theoretical results could be criticised for being fairly complex, having spend some time understanding them, the payoff makes it absolutely worthwhile. I don't see that they could be made simpler.

**Questions:**

- What the three bars for each matrix in Fig 2b? What is the thick vertical line?
- In Fig 3d, is there reason to expect the norms of the gradients to be on the same scale as the difference between the matrix at the saddle points and the optimal approximation for the corresponding rank? They seem to be close to the gradient norms at the saddle points. Is this a coincidence or is there a reason to expect this?
- In Fig 3c and 4e, loss trajectories for different sampling patterns are plotted. If I understand correctly, in Fig 3a for example, each line corresponds to a different entry of the matrix being unobserved. In this case is $\star = 1.2$, so the underlying rank is three, or does it take a different value? Is it the case that the matrix has rank three if and only if the unobserved entry takes a specific value?
- In Fig 4f, are you counting position indexes from zero? In Section 3 (lines 97 and 99), you count position indexes from one.
- The red dot in Fig 4e would perhaps be best a little larger and plotted on top of the other lines. At present it is quite hard to see.
- Assumption 1 is equivalent to the top singular value of $-\delta \mathbf{M}$​ being unique.  This seems like a more intuitive way to write this assumption. Is there a reason you chose to write the assumption in terms of the eigenvalues of the symmetric dilation matrix?
- In 231, it is claimed that "in the connected case, at each level [the] model reaches an optimal solution". Is this implied by one of the theorems, by previous work, or is this an empirical observation?

**Limitations:**

The authors are upfront about the limitations of their work in the conclusions, which present exciting directions for future research.

---

> ### Author Rebuttal · Authors · 2024-08-06
>
> * **Point 1: I find this work hard to fault. While the theoretical results could be criticised for being fairly complex, having spend some time understanding them, the payoff makes it absolutely worthwhile.**
>
> * Reply: We are deeply grateful for the reviewer's careful examination and understanding of our theoretical results. Your recognition of the value and payoff of our complex theoretical framework is greatly appreciated.
>
> * **Point 2: What the three bars for each matrix in Fig 2b? What is the thick vertical line?**
>
> * Reply: We appreciate the reviewer's attention to detail regarding Figure 2b. To clarify:
>
>   - The three bars for each matrix represent the singular values of the learned matrix.
>   - The thick vertical line partitions significantly nonzero singular values, which serves as the empirical rank.
>
>   To enhance clarity, we have updated the caption of Figure 2b as follows:
>
>   "Figure 2(b): Singular values of the learned matrices for $M_1, M_2, M_3$. Each set of three bars represents the singular values of a matrix. The thick vertical lines partition significantly nonzero singular values, which serves as the empirical rank."
>
> * **Point 3: In Fig 3d, is there reason to expect the norms of the gradients to be on the same scale as the difference between the matrix at the saddle points and the optimal approximation for the corresponding rank?**
>
> * Reply: We appreciate the reviewer's insightful observation about the similarity between gradient norms and matrix differences at saddle points. This relationship indeed stems from the structure of optimization and can be more precisely characterized.
>
>   As $R_S(\theta)$ approaches $R_S(\theta^*)$, we observe that $\nabla R_S(\theta)$ also approaches zero. This relationship can be formalized under certain conditions:
>
>   1. For $L$-smooth functions, we have the inequality: $\\|\nabla R_S(\theta)\\|^2 \leq 2L(R_S(\theta) - R_S(\theta^*))$, where $L$ is the Lipschitz constant of the gradient.
>
>   2. For $\mu$-strongly convex functions, we also have the inequality (the Polyak-Łojasiewicz condition): $\\|\nabla R_S(\theta)\\|^2 \geq 2\mu(R_S(\theta) - R_S(\theta^*))$.
>
>   In our matrix factorization setting, while the problem isn't globally strongly convex, it may exhibit local strong convexity restricted to invariant manifold $\Omega_k$ near optimal solutions. This local behavior possibly contributes to the observed similarity.
>
> * **Point 4: Clarification on Figures 3c and 4e. For Figure 3c, is it the case that the matrix has rank three if and only if the unobserved entry takes a specific value?**
>
> * Reply: We appreciate the reviewer's attention to detail regarding Figures 3c and 4e. To clarify:
>
>   1. The dashed lines in both figures correspond to different sampling patterns. For Figure 3(c), the matrix indeed has rank three if and only if the unobserved entry takes a specific value. This is because, given 15 observations, the rank-3 matrix is uniquely determined.
>
>   2. To improve clarity, we have updated the captions of Figures 3(c) and 4(e):
>
>   "Figure 3(c): Training loss for 16 connected sampling patterns in a $4\times 4$ matrix, each covering 1 element and observing the remaining 15 in a fixed rank-3 matrix."
>
>   "Figure 4(e): Training loss for 9 disconnected sampling patterns in a $3\times 3$ matrix, each covering 4 elements and observing the remaining 5 in a fixed rank-1 matrix."
>
> * **Point 5: In Fig 4f, are you counting position indexes from zero? In Section 3 (lines 97 and 99), you count position indexes from one.**
>
> * Reply: We sincerely thank the reviewer for pointing out this inconsistency. Indeed, in Figure 4f, we counted position indexes from zero, while in Section 3 we counted from one. To maintain consistency throughout the paper, we have updated Figure 4f to count from one.
>
>   We've revised the caption of Figure 4f as follows:
>
>   "Figure 4f: Learned values at symmetric positions $(1, 2)$ and $(2, 1)$ under varying initialization scales (zero mean, varying variance)."
>
> * **Point 6: The red dot in Fig 4e would perhaps be best a little larger and plotted on top of the other lines.**
>
> * Reply: We appreciate this suggestion for improving the figure's clarity. We have implemented this change by increasing the size of the red dot and ensuring it's plotted on top of the other lines in Figure 4e. This modification enhances the visibility of this key data point in the figure.
>
> * **Point 7: Assumption 1 is equivalent to the top singular value of $-\delta M$ being unique. This seems like a more intuitive way to write this assumption.**
>
> * Reply: We thank the reviewer for this insightful suggestion. We agree that reformulating Assumption 1 in terms of the top singular value of $-\delta M$ is indeed more intuitive. We have revised Assumption 1 accordingly:
>
>   "Assumption 1 (Unique Top Singular Value): Let $\delta M=\left(A_c B_c-M\right)_{S_x}$ be the residual matrix at the critical point $\theta_c=\left(A_c, B_c\right)$. Assume that the largest singular value of $\delta M$ is unique."
>
> * **Point 8: In 231, it is claimed that "in the connected case, at each level [the] model reaches an optimal solution". Is this implied by one of the theorems, by previous work, or is this an empirical observation?**
>
> * Reply:  The statement "in the connected case, at each level the model reaches an optimal solution" is based on our empirical observations. Our experiments consistently show this behavior in connected cases, as illustrated in Figure 3.
>
>   To improve precision, we have revised the statement in line 231 to:
>
>   "In the connected case, at each level we observe that the model reaches an optimal solution (Figure 3)."
>
>    This revision clarifies that our claim is based on empirical evidence rather than a theoretical guarantee.
>
> We sincerely appreciate the reviewer's thorough examination of our work, which has significantly contributed to improving the clarity and precision of our presentation.

---

> > ### Comment · Reviewer_AjkV · 2024-08-09
> >
> > I thank the authors for their clarifications and I continue to strongly support this paper.

---

### Official Review · Reviewer_FMau · 2024-07-12

**Soundness:** 3
**Presentation:** 3
**Contribution:** 3
**Rating:** 6
**Confidence:** 3

**Summary:**

This paper attempts to present a unified understanding of when and how matrix factorization models have different implicit regularization effects. Their key finding is that connectivity of the observed data plays an important role: (i) in the connected cases, the model learns the lowest-ranked solution and (ii) in the disconnected case, it seems to sort of depend, but generally does not find the minimum nuclear norm solution. They identify invariant manifolds in the loss landscape that guides the training trajectory from these lower ranked to higher ranked solutions. They support a few of their findings with theoretical results.

**Strengths:**

- Overall, I think this paper presents some neat findings while addressing an interesting question.
- The paper is well-written and supported with examples to help elucidate the definitions / settings.
- The paper fits well into the literature of implicit regularization, and helps unify some of the observations in the literature as well, such as greedy low-rank learning.

**Weaknesses:**

- I am unsure of how restrictive the assumptions used for the theoretical results are which are used to prove Theorem 2. The authors do make a remark about the assumptions, but it is still not immediately clear to me that these are fair assumptions to make.
- As the authors mention, it would be interesting (and important) future work to show how the connectivity of the observed data holds attainment across the invariant manifolds.

**Questions:**

- I am trying to understand Figure 4(e): do the different dashed lines correspond to different variances of the initialization? In that case, I am assuming that the figure is trying to show that despite the initialization scales, the model always learns a sub-optimal solution in this disconnected case.

**Limitations:**

I have addressed a few limitations in the weaknesses section.

---

> ### Author Rebuttal · Authors · 2024-08-06
>
> * **Point 1: I am unsure of how restrictive the assumptions used for the theoretical results are which are used to prove Theorem 2.**
>
> * Reply: We appreciate the reviewer's concern regarding the restrictiveness of our assumptions. We have taken steps to clarify and justify these assumptions:
>
>   1. Regarding Assumption 1, as other reviewers have noted, it can be more concisely stated as the uniqueness of the largest singular value of the residual matrix. We have revised it as follows:
>
>     "Assumption 1 (Unique Top Singular Value): Let $\delta M=\left(A_c B_c-M\right)_{S_x}$ be the residual matrix at the critical point $\theta_c=\left(A_c, B_c\right)$. Assume that the largest singular value of $\delta M$ is unique."
>
>   2. We have also updated the accompanying remark:
>
>     "Remark: Assumption 1 ensures that upon departing from a critical point $\theta_c$, the trajectory is constrained to escape along a single dominant eigendirection corresponding to the largest singular value. This assumption holds for randomly generated matrix with probability 1, making it a reasonable condition in most practical scenarios."
>
>   3. To illustrate the implications when Assumption 1 does not hold, we've added an example in the appendix:
>
>     "Consider the $2\times 2$ matrix completion problem: $M = \begin{bmatrix}2 & \star\\\\ \star & 2\end{bmatrix}$. In this case, the two numbers on the diagonal are identical, which causes the maximum singular value of the residual matrix at the origin to be non-unique. Consequently, the training process will jump directly from the rank 0 to the rank 2 invariant manifold, thereby missing the lowest rank solution of rank 1. This behavior is demonstrated in the attached PDF (Figure 1), which shows experimental results for this scenario."
>
>   4. To enhance understanding of Theorem 2, we've included a detailed proof sketch:
>
>    "Proof sketch: We analyze the local dynamics in the vicinity of the critical point $\theta_c$. The nonlinear dynamics can be approximated linearly near $\theta_c$: $\frac{d\theta}{dt} \approx H(\theta_0 - \theta_c)$, where $H = -\nabla^2R_S(\theta_c)$ is the negative Hessian matrix. For exact linear approximation, the solution is: $\theta(t) = e^{tH}(\theta_0 - \theta_c) + \theta_c$. Let $\lambda_1 > \lambda_2 > ... > \lambda_s$ be the eigenvalues of $H$, with corresponding eigenvectors $q_{ij}$. We can express $\theta(t)$ as: $\theta(t) = \sum_{i=1}^s \sum_{j=1}^{l_i} e^{\lambda_i t}\langle\theta_0 - \theta_c, q_{ij}\rangle q_{ij} + \theta_c$. For sufficiently large $t_0$, the dynamics follows a dominant eigenvalue trajectory: $\theta(t_0) = \sum_{j=1}^{l_1} e^{\lambda_1 t_0}\langle\theta_0 - \theta_c, q_{1j}\rangle q_{1j} + O(e^{\lambda_2t_0}).$ Through detailed analysis of the eigenvalues and eigenvectors of the Hessian matrix (Lemmas A.2-A.4), we demonstrate that if the largest singular value of residual matrix $\delta M$ at $\theta_c$ is unique and $\theta_c$ is a second-order stationary point within $\Omega$, the first principal component $\sum_{j=1}^{l_1} e^{\lambda_1 t_0}\langle\theta_0 - \theta_c, q_{1j}\rangle q_{1j}$ will correspond to an $\Omega_1$ invariant manifold. Consequently, escaping $\theta_c$ increases the rank by 1, entering $\Omega_{k+1}$."
>
>   5. Regarding Assumption 2, we've revised the remark to better explain its role:
>
>   "To ensure the escape direction falls within the $\Omega_{k+1}$ invariant manifold, the Hessian's top eigenvectors must satisfy $\text{rank}(A) = \text{rank}(B^\top) = \text{rank}(W_{\text{aug}})$. The condition that $\theta_c$ is a second-order stationary point within $\Omega$ in Assumption 2 guarantees this Hessian structure. Our Assumption 2 is more general than conditions proposed by Li et al. (2020), as it remains valid across both connected and disconnected configurations. Empirical findings (Fig. 3 and Fig. 4) indicate that this assumption consistently holds in practical scenarios."
>
> * **Point 2: It would be interesting (and important) future work to show how the connectivity of the observed data holds attainment across the invariant manifolds.**
>
> * Reply: We concur that investigating the relationship between data connectivity and attainment across invariant manifolds is a crucial direction for future research. Our planned approach includes a comprehensive analysis of the loss landscape structure at each invariant manifold level, with a particular focus on the characteristics of critical points. This investigation will provide valuable insights into how data connectivity influences the optimization trajectory and the ultimate attainment of low-rank solutions.
>
> * **Point 3: Do the different dashed lines correspond to different variances of the initialization?**
>
> * Reply: We appreciate the reviewer's attention to detail regarding Figure 4(e). To clarify:
>
>   1. The dashed lines in Figure 4(e) represent different disconnected sampling patterns, each covering 4 elements and observing the remaining 5 elements in a fixed rank-1 3x3 matrix. We comprehensively explored all nine possible disconnected sampling patterns with 5 observations. To enhance clarity, we have updated the caption of Figure 4(e):
>
>   "Figure 4(e): Training loss for all nine disconnected sampling patterns in a 3x3 matrix, each covering 4 elements and observing the remaining 5 in a fixed rank-1 matrix."
>
>   2. Furthermore, we have added a brief discussion in Section 5.2, Line 199:
>
>   "In Figure 4(e), we fixed a rank-1 matrix and explored all nine disconnected sampling patterns with 5 observations. For each pattern, we conducted experiments with small initializations. The loss curves consistently indicate that in disconnected cases, the model learns a sub-optimal solution in the rank-1 manifold, ultimately resulting in a rank-2 solution. This demonstrates that regardless of the specific disconnected sampling pattern, the model fails to achieve the optimal low-rank solution."
>
> We sincerely appreciate the reviewer's thorough examination of our work.

---

> > ### Comment · Reviewer_FMau · 2024-08-12
> > **Thank you for your detailed response**
> >
> > Thank you for clearing up my confusion on the figure and the detailed response to the assumptions. I believe that my current assessment of the paper is correct and will keep my score, which recommends acceptance.

---

### Official Review · Reviewer_CGEg · 2024-07-12

**Soundness:** 3
**Presentation:** 3
**Contribution:** 3
**Rating:** 6
**Confidence:** 3

**Summary:**

This paper systematically investigates the implicit regularization of matrix factorization for solving matrix completion problems. The authors find by empirical results that the connectivity of observed data plays a crucial role in the implicit bias, with a transition from low nuclear norm to low rank as data becomes more connected with increased observations. They identify a hierarchy of intrinsic invariant manifolds in the loss landscape that guide the training trajectory to evolve from low-rank to higher-rank solutions. The optimization trajectory follows a Hierarchical Invariant Manifold Traversal (HIMT) process, generalizing the characterization of Li et al. (2020), whose proposed Greedy Low-Rank Learning (GLRL) algorithm equivalence only corresponding to the connected case. Regarding the minimum nuclear norm regularization, the authors establish conditions that provide guarantees closely aligned with the empirical findings, and they present a dynamic characterization condition that assures the attainment of the minimum rank solution.

**Strengths:**

- The paper systematically studies the training dynamics and implicit regularization of matrix factorization for matrix completion, considering both connected and disconnected cases, which provides a more comprehensive understanding of the model.
- The paper identifies the hierarchical invariant manifolds in the loss landscape and characterizes the training trajectory, providing a theoretical basis for understanding the model's behavior.
- The authors conduct extensive experiments to support their findings, demonstrating the influence of data connectivity on the implicit regularization and the training dynamics of the matrix factorization model.

**Weaknesses:**

- In some cases, an extremely small initialization is required, which may potentially impact the training speed.
- The proofs of the theoretical results are quite complex and may be difficult to follow for readers who are not familiar with the mathematical background. A more intuitive explanation or proof sketch could help readers better understand the key ideas.
- It is suggested to provide justifications to demonstrate the reasonableness of Assumption 1.

**Questions:**

No

---

> ### Author Rebuttal · Authors · 2024-08-01
>
> * **Point 1: In some cases, an extremely small initialization is required, which may potentially impact the training speed.**
>
> * Reply: Our empirical analysis in Appendix B.4 (Fig. B6) demonstrates the relationship between observation magnitude differences and the required initialization scale. To elucidate the trade-off between initialization scale and training speed, we have expanded Appendix B.4 with the following details:
>
>   1. Theoretical optimization insights: "For the matrix factorization model $f_\theta = AB$, the Hessian matrix at 0 has strictly negative eigenvalues, making the origin a strict saddle point. Under small random initialization, gradient descent escapes this saddle at an exponential rate. Our Theorem 1 ensures that only strict saddle points and global minima exist as critical points in the loss landscape. Subsequent saddle points on invariant manifolds are also strict, facilitating exponential escape speeds and maintaining an efficient optimization process."
>
>   2. Practical low-rank attainment: "Achieving the lowest rank solution requires parameters to escape the saddle point along the unique top eigen-direction and keep evolving near the invariant manifold $\Omega_k$. Fig. B6 in Appendix B.4 illustrates the relationship between required initialization scale and observation magnitude differences. For the lowest possible rank solution with large numerical magnitude differences in observations, an extremely small initialization is necessary. However, for approximately low rank solutions (some singular values ​​are relatively small), which may be sufficient in practice, a relatively small initialization suffices without significantly impacting training speed."
>
> * **Point 2: The proofs of the theoretical results are quite complex and may be difficult to follow for readers who are not familiar with the mathematical background. A more intuitive explanation or proof sketch could help readers better understand the key ideas.**
>
> * Reply: We acknowledge that Theorem 2 is a comprehensive result with a relatively complex proof. To enhance understanding, we have added a more detailed proof sketch for Theorem 2:
>
>   "Proof sketch: We analyze the local dynamics in the vicinity of the critical point $\theta_c$. The nonlinear dynamics can be approximated linearly near $\theta_c$: $\frac{d\theta}{dt} \approx H(\theta_0 - \theta_c)$, where $H = -\nabla^2R_S(\theta_c)$ is the negative Hessian matrix. For exact linear approximation, the solution is: $\theta(t) = e^{tH}(\theta_0 - \theta_c) + \theta_c$. Let $\lambda_1 > \lambda_2 > ... > \lambda_s$ be the eigenvalues of $H$, with corresponding eigenvectors $q_{ij}$. We can express $\theta(t)$ as: $\theta(t) = \sum_{i=1}^s \sum_{j=1}^{l_i} e^{\lambda_i t}\langle\theta_0 - \theta_c, q_{ij}\rangle q_{ij} + \theta_c$. For sufficiently large $t_0$, the dynamics follows a dominant eigenvalue trajectory: $\theta(t_0) = \sum_{j=1}^{l_1} e^{\lambda_1 t_0}\langle\theta_0 - \theta_c, q_{1j}\rangle q_{1j} + O(e^{\lambda_2t_0}).$ Through detailed analysis of the eigenvalues and eigenvectors of the Hessian matrix (Lemmas A.2-A.4), we demonstrate that if the largest singular value of residual matrix $\delta M$ at $\theta_c$ is unique and $\theta_c$ is a second-order stationary point within $\Omega$, the first principal component $\sum_{j=1}^{l_1} e^{\lambda_1 t_0}\langle\theta_0 - \theta_c, q_{1j}\rangle q_{1j}$ will correspond to an $\Omega_1$ invariant manifold. Consequently, escaping $\theta_c$ increases the rank by 1, entering $\Omega_{k+1}$. We defer the details to Appendix A."
>
> * **Point 3: It is suggested to provide justifications to demonstrate the reasonableness of Assumption 1.**
>
> * Reply: 1. We appreciate this suggestion and have revised Assumption 1 and its accompanying remark to provide better justification:
>
>   "Assumption 1 (Unique Top Singular Value): Let $\delta M=\left(A_c B_c-M\right)_{S_x}$ be the residual matrix at the critical point $\theta_c=\left(A_c, B_c\right)$. Assume that the largest singular value of $\delta M$ is unique."
>
>   "Remark: Assumption 1 ensures that upon departing from a critical point $\theta_c$, the trajectory is constrained to escape along a single dominant eigendirection corresponding to the largest singular value. This assumption holds for randomly generated matrix with probability 1, making it a reasonable condition in most practical scenarios."
>
>   2. To further illustrate the implications when this assumption does not hold, we have added a simple example in the appendix:
>
>   "Consider the $2\times 2$ matrix completion problem: $M = \begin{bmatrix}2 & \star\\\\ \star & 2\end{bmatrix}$. In this case, the two numbers on the diagonal are identical, which causes the maximum singular value of the residual matrix at the origin to be non-unique. Consequently, the training process will jump directly from the rank 0 to the rank 2 invariant manifold, thereby missing the lowest rank solution of rank 1. This behavior is demonstrated in the attached PDF (Figure 1), which shows experimental results for this scenario."
>
> We sincerely appreciate the reviewer's thorough examination of our work, which has significantly contributed to improving the clarity and precision of our presentation.

---

> > ### Comment · Reviewer_CGEg · 2024-08-11
> > **Response to rebuttals**
> >
> > Thank you for the rebuttals. The authors' responses to my main concerns are generally satisfactory to me. Alongside the reviews of other reviewers, I am inclined to maintain my positive assessment of this submission.

---

### Author Rebuttal · Authors · 2024-08-06

Dear Reviewers,

We sincerely thank all reviewers for their thoughtful and insightful comments. We have carefully addressed every comment, and we believe that the reviewers' collective feedback has significantly improved the manuscript. To address the common concerns raised, we have made the following key improvements:

1. **Clarified Definitions:** We have refined the definition of connectivity and provided a more intuitive explanation of the augmented matrix, enhancing the overall readability of the paper.

2. **Enhanced Experimental Details:** We have added comprehensive information about our experimental setup, including initialization methods and training dynamics, in both the main text and Appendix B.1.

3. **Expanded Theoretical Justifications:** We have provided more detailed proof sketches and intuitive explanations for our theoretical results, particularly for Theorem 2.

4. **Improved Figures and Captions:** We have updated several figures (e.g., Fig. 2b, 3c, 4e, 4f) and their captions to provide clearer visualizations and explanations of our results.

5. **Added High-Dimensional Experiments:** To address concerns about scalability, we have included new experiments on 20x20 matrices, demonstrating the applicability of our findings to higher-dimensional problems.

6. **Strengthened Motivation:** We have elaborated on the importance of data connectivity in shaping implicit regularization, providing a stronger justification for our study.

7. **Corrected Mathematical Notations:** We have fixed minor typographical errors in mathematical expressions to ensure accuracy.

8. **Extended Appendix:** We have added more detailed examples and explanations in the appendix, particularly regarding connectivity and disconnectivity.

Additionally, we have included a one-page PDF in the attachment presenting key experimental results, including the example of coincident top eigenvalues and the high-dimensional experiments.

We believe these revisions comprehensively address the reviewers' concerns while maintaining the core contributions of our work. The revised manuscript now offers a clearer, more rigorous, and more insightful exploration of how connectivity shapes implicit regularization in matrix factorization models for matrix completion.

We sincerely hope that the revised manuscript now satisfies the reviewers' requirements, and we hereby submit it for publication consideration. We are grateful for the opportunity to improve our work based on such valuable feedback.

Best regards,

The Authors

---

### Decision · Program_Chairs · 2024-09-25

**Decision:**

Accept (poster)

**Comment:**

The paper investigates the implicit regularization of matrix factorization for solving matrix completion problems, revealing that the connectivity of observed data has a strong influence on the impact of implicit regularization. It received a positive response from the reviewers, who unanimously found it novel and original, with sound numerical and theoretical analysis and a high-quality presentation.